# Bootstrapping bulk locality
# Part II: Interacting functionals

**Nat Levine**[a,b]   and   **Miguel F. Paulos**[a]

[a]*Laboratoire de Physique,*       [b]*Institut Philippe Meyer,*
*École Normale Supérieure,*
*Université PSL, CNRS, Sorbonne Université, Université Paris Cité,*
*24 rue Lhomond, F-75005 Paris, France*

nat.levine@phys.ens.fr ,  miguel.paulos@ens.fr

## Abstract

Locality of bulk operators in AdS imposes stringent constraints on their description in terms of the boundary CFT. These constraints are encoded as sum rules on the bulk-to-boundary expansion coefficients. In this paper, we construct families of sum rules that are (i) complete and (ii) 'dual' to sparse CFT spectra. The sum rules trivialise the reconstruction of bulk operators in strongly interacting QFTs in AdS space and allow us to write down explicit, exact, interacting solutions to the locality problem. Technically, we characterise 'completeness' of a set of sum rules by constructing Schauder bases for a certain space of real-analytic functions. In turn, this allows us to prove a Paley-Weiner type theorem characterising the space of sum rules. Remarkably, with control over this space, it is possible to write down closed-form 'designer sum rules', dual to a chosen spectrum of CFT operators meeting certain criteria. We discuss the consequences of our results for both analytic and numerical bootstrap applications.

# 1 Introduction

A special class of $d$-dimensional conformal field theories (CFTs) admit a dual description as **local** theories in a $(d + 1)$-dimensional bulk Anti de Sitter (AdS) spacetime. Although the main case of interest is when this theory has a dynamical metric, much can be learned by considering the simpler case of UV-complete QFTs in a non-fluctuating AdS space. Such theories support well-defined, exactly localised operators which are mutually local, i.e. commuting at spacelike separations (unlike the gravitational case). Expressing these local bulk fields in terms of the boundary CFT data is an old problem studied by several authors [1–14].

In our previous work [15], we considered this problem from a modern bootstrap perspective. Our starting point was the state–operator correspondence in AdS, which guarantees that these operators admit a representation in terms of boundary CFT operators, schematically

$$\Psi^{\mathrm{AdS}} \sim \sum_{\Delta} \mu_\Delta \, \mathcal{O}_\Delta^{\mathrm{CFT}} \, . \tag{1.1}$$

The task of reconstructing a bulk field $\Psi^{\mathrm{AdS}}$ means producing an admissible choice of the dynamical information in this *boundary operator expansion* (BOE): the coefficients $\mu_\Delta$. To

see that most choices of $\mu_\Delta$ would not be admissible, one may inspect the correlator of the putative bulk field against two boundary operators $\mathcal{O}_1, \mathcal{O}_2$. After stripping off universal kinematic prefactors, the resulting 2-point *AdS form factor* $F(z)$ depends on a single conformal cross-ratio, denoted $z$, and admits a BOE expansion:

$$\langle \Psi \,|\, \mathcal{O}_1 \,\mathcal{O}_2 \rangle \sim F(z) = \sum_\Delta \mu_\Delta \lambda_\Delta^{12}\, G_\Delta(z)\,, \tag{1.2}$$

with further details given below.

Microcausality in the bulk implies that correlation functions can only become singular when local insertions enter each other's lightcones. In the present setup, this only happens for $z \leq 0$, so the only singularity of $F$ should be there. On the other hand, the BOE allows the existence of another, unphysical branch cut at $z \geq 1$ (at Lorentzian but spacelike-seperated kinematics), which is present in each summand appearing above. Our locality constraint is the vanishing of this acausal branch cut [12]:

$$\mathcal{I}_{z \geq 1} F(z) = 0 \,. \tag{1.3}$$

Since each block $G_\Delta$ has a discontinuity at $z \geq 1$, this clearly places constraints on the coefficients $\mu_\Delta, \lambda_\Delta$. However, characterizing these constraints is not immediately easy since the discontinuity appearing in this equation does not commute with the BOE expansion.

We resolved this issue in [15] by smearing the form factor against specially designed **analytic functional kernels**, along a contour that wraps the hypothetical $z \geq 1$ cut. After this smearing, one can commute the locality condition with the BOE to produce sum rules on the BOE data. Concretely, we constructed a basis of kernels $\{f_n\}$ that lead to sum rules satisfying two desirable properties:

---

**Property 1): Completeness**

$$\mathcal{I}_{z \geq 1} F(z) = 0 \qquad \Longleftrightarrow \qquad \sum_\Delta \mu_\Delta \lambda_\Delta^{12}\, \theta_n(\Delta) = 0 \quad \text{for} \ \ n \geq 1 \ \ .$$

**Property 2): Duality**

For some $\{\Delta_n\}_{n=1}^\infty$, $\quad \theta_n(\Delta_m) = \delta_{nm} \quad \text{for} \ \ m, n \geq 1 \,.$

---

Here $\theta_n(\Delta)$ denotes the action of a functional $f_n$ on the block $G_\Delta$. Let us emphasise that these conditions draw up a duality between a basis of functionals and some distinguished spectrum,

$$\{f_n\} \quad \longleftrightarrow \quad \{\Delta_n\}\,. \tag{1.4}$$

In [15], we constructed functionals satisfying these properties, dual to the specific choice of a Generalized Free Field-type spectrum $\{\Delta_n = 2\tilde{\alpha} + 2n\}$. In this paper, we will do it for a very general class of interacting spectra.

Before detailing our new results, let us explain the significance of the above properties. Completeness is clearly desirable, guaranteeing that the sum rules fully capture the locality constraint. However, it is still a difficult task to actually solve the infinite set of sum rules for the BOE data. Duality strikes at the core of this issue: it means the sum rules are trivialised when the BOE spectrum $\{\Delta\}$ is taken to contain the particular distinguished one $\{\Delta_n\}$ that is 'dual' to the basis of functionals. As a result, the sum rules, and hence the locality problem, admit certain canonical solutions. For example, making the ansatz for the BOE spectrum $\{\Delta\} = \{\Delta_n\} \cup \{\Delta_0\}$ consisting of the dual spectrum, plus one extra block of dimension $\Delta_0$, the sum rules are **diagonalised** and admit an exact solution for the BOE data $c_n \equiv \mu_{\Delta_n}\lambda_{\Delta_n}^{12}$:

$$
\mathcal{I}_{z\geq 1}\left[G_{\Delta_0} + \sum_{n=1}^{\infty} c_n\, G_{\Delta_n}\right] = 0 \qquad \underset{\theta_n(\Delta_m)=\delta_{nm}}{\overset{\text{Duality}}{\Longleftrightarrow}} \qquad c_n = -\theta_n(\Delta_0)\ . \tag{1.5}
$$

For the choice $\Delta_n = 2\tilde{\alpha} + 2n$ of [15], the form factors corresponding to these solutions were called 'local blocks', and have a simple interpretation in terms of weakly coupled QFTs in AdS space. For instance, with $\tilde{\alpha} = \frac{\Delta_0}{2} = \Delta_\phi$, they correspond to the form factors $\langle \Phi^2 | \phi\phi \rangle$ in the theory of a free scalar $\Phi$ in AdS with mass $m^2 = \Delta_\phi(\Delta_\phi - d)$.

We will refer to these special solutions as **extremal solutions**. These solutions share many features with the familiar extremal solutions that saturate unitarity bounds in other problems, such as the crossing equation [16–20]. Namely, (i) they are 'minimal' solutions, in the sense that no blocks can be removed, (ii) they possess dual bases of functionals and (iii) they can be explicitly bootstrapped using those functionals. Indeed, understanding extremal solutions of bootstrap problems is an important motivation for us: it is equivalent to understanding the bootstrap bounds that they saturate, and would open the door for their analytic understanding. In the case of the 1d crossing equation for 4-point correlators, free extremal solutions have been studied analytically [17–19], while interacting ones have only been studied numerically [20]. For this simpler locality problem, we can do better: for a very large set of choices of interacting spectra $\{\Delta_n\}$, we will analytically construct sets of sum rules satisfying properties 1) and 2) and the corresponding extremal solutions (1.5).

Our main results concern the locality problem for $d < 4$ and are stated as follows:

**Assumptions.** Take a '**sparse**' spectrum, $\Delta_n = 2\tilde{\alpha} + 2n + \gamma_n$ $(n \in \mathbb{N}_{>0})$, meaning:

- $\gamma_n \underset{n \to \infty}{=} O(n^{-\epsilon})$, $\quad \epsilon > 0$,

- $\gamma_n$ is an analytic function at large $n \in \mathbb{C}$ (for small $\arg(n)$).

**Results.**

- Then there exist functionals $\theta_n$ satisfying properties 1) and 2),

- The functionals form a **Schauder basis** for a certain topological vector space,

- The functionals' actions on blocks, $\theta_n(\Delta)$, can be written down explicitly:

$$\theta_n(\Delta) = \prod_{\substack{m=1 \\ m \neq n}}^{\infty} \left( \frac{\lambda_\Delta - \lambda_{\Delta_m}}{\lambda_{\Delta_n} - \lambda_{\Delta_m}} \right), \qquad \lambda_\Delta = \Delta(\Delta - d).$$

Let us now comment on the relevance of these results.

Firstly, as discussed above, they immediately produce huge classes of interacting, highly non-trivial solutions to the locality problem. The most elementary such solution has the BOE coefficients given by equation (1.5). Using the third result above, the resulting local form factor takes the explicit form

$$F_{\Delta_0}^{\{\Delta_n\}} = G_{\Delta_0} - \sum_{n=1}^{\infty} \Big[ \prod_{\substack{m=1 \\ m \neq n}}^{\infty} \left( \frac{\lambda_{\Delta_0} - \lambda_{\Delta_m}}{\lambda_{\Delta_n} - \lambda_{\Delta_m}} \right) \Big] G_{\Delta_n}. \tag{1.6}$$

These non-perturbative, interacting solutions are valid for any desired choice of $\{\Delta_n\}$ satisfying the assumptions above. The products and sums are absolutely convergent and are readily evaluated numerically: for an example of this technology in action, the reader may look ahead to plots of these form factors in Figure 6.

Secondly, for general $d$, these results can be taken as another piece of evidence that Generalized Free Fields and their deformations arise from local AdS Lagrangians [21]. Indeed, for a CFT correlator whose (scalar) spectrum is close to GFF, and without additional operators, our results allow us to readily construct associated solutions to the locality problem in AdS. Moreover, for the $d = 1$ case (corresponding to $AdS_2/CFT_1$), our results indicate that arbitrary 'extremal' solutions $\langle \phi\phi\phi\phi \rangle$ of the 1d crossing equation admit local AdS fields $\Psi$, such that the form factors $\langle \Psi | \phi\phi \rangle$ are manifestly local and computable. This is because those extremal solutions (saturating unitarity bounds) have sparse spectra of the type assumed above [20].[1] This means that, in order to improve numerical bootstrap bounds, one must do more than simply imposing locality constraints on top of the standard

---

[1] At least with respect to sparseness. Analyticity properties also hold in perturbation theory and in certain (flat space) limits, but have not be established in full generality.

semi-definite programming for the crossing equation. We will discuss this matter further in Section 6.

Finally, while the second result in the box above sounds abstract and technical, we would like to emphasise the conceptual importance of the functionals forming a complete basis. For the first time, we will make precise the notion of completeness of functionals for the bootstrap; to which space they belong; and the deep connection between complete bases of functionals and extremal solutions of the bootstrap (i.e. solutions uniquely determined by bootstrap equations, with sparsest possible OPEs, and saturating bounds). Our results make concrete a correspondence:

$$
\boxed{\text{Extremal solutions} \quad \longleftrightarrow \quad \text{Bases of a topological vector space}}
$$

Establishing the existence of the interacting bases will lead us through some unfamiliar but rich mathematics. While our construction will be detailed for the locality problem, we will argue that the same logic is also likely to apply to the crossing problem for CFTs. More generally, we believe that the formalism and constructions of this work are the right mathematical language for thinking about analytic functionals — which have found important applications not only in physical, but also mathematical, problems (see [22] and refs. therein).

The remainder of this introduction is a bird's eye view of the ideas leading to the above results and an outline of the paper's structure.

## Key ideas of the paper

Our first task is to understand carefully what it means for a set of functionals to be complete. A familiar notion of completeness arises in Hilbert spaces, which possess orthonormal bases, with respect to which arbitrary elements can be decomposed. Given such an orthonormal basis $|e_n\rangle$, an element is shown to be zero by checking that its expansion coefficients in that basis are all zero,

$$
|F\rangle = 0 \qquad \Leftrightarrow \qquad \theta_{e_n}[F] := \langle e_n | F \rangle = 0 \quad \forall n \ . \tag{1.7}
$$

We emphasise that $e_n$ has the interpretation as an element both of the Hilbert space and of its dual, i.e. as a bounded linear functional acting on the Hilbert space. Can we apply the same logic to our problem? We need to check whether a certain discontinuity of an analytic function vanishes (eq. (1.3)). As mentioned above, we will obtain our sum rules by integrating the unphysical cut of the form factor against a functional kernel:

$$
\theta_n[F] = \int_1^\infty \mathrm{d}z \, f_n(z) \, \mathcal{I}_z F \ . \tag{1.8}
$$

This expression would suggest defining some suitable Hilbert space of functions on $[1, \infty]$ for which $f_n$ would form a basis. Making this idea concrete will require overcoming several

difficulties and the introducing some new concepts.

The first such difficulty deals with understanding what kind of object is $\mathcal{I}_z F$ and in what space it lives. We propose that it is both natural and convenient to view this object as a *hyperfunction*: the discontinuity of an analytic function across a branch cut.[2] Hyperfunctions are a very large class of distributions: they contain derivatives of delta-functions of arbitrary orders, but also non-Schwartzian distributions such as the discontinuity of an essential singularity,

$$\mathcal{I}_z \frac{1}{z^n} \sim \partial_z^n \delta(z) \;, \qquad \mathcal{I}_z \, e^{1/z} \; \sim \; \text{``} \partial_z^\infty \delta(z) \text{''} \tag{1.9}$$

The locality condition is then simply the requirement that it $\mathcal{I}_{z \geq 1} F$ is the zero hyperfunction. Choosing to work in this space means that we can rely on elegant tools of complex analysis to understand the constraints of locality. It also makes only minimal assumptions about the properties of form factors (essentially absolute convergence of the BOE), meaning the scope of our work extends to cases where we may in future want to allow non-trivial discontinuities (say when reconstructing non local bulk operators).[3]

We are thus led to consider a space of hyperfunctions $\mathcal{HF}$ with a branch cut along an interval $V$. This is a topological vector space, so it has a notion of convergence, but does not have an inner-product or norm, and in particular it is not a Hilbert space. However, it does have a **dual space**, which is the space $\mathcal{A}$ of **real-analytic functions** on $V$. There is then a dual pairing between $f \in \mathcal{A}$ and $F \in \mathcal{HF}$:

$$\theta_f[F] = (f, F) := \oint_{\Gamma_V} \frac{\mathrm{d}z}{2\pi i} \; f(z) \, F(z) \;, \tag{1.10}$$

where $\Gamma_V$ is a contour encircling the interval $V$, inside the shared domain of analyticity of $f$ and $F$. As we will discuss, defining the functionals in this way makes the *swapping* property of [24] automatic, since the pairing $(\bullet, \bullet)$ is continuous.

The space $\mathcal{A}$ is again not a Hilbert space, but it still makes sense to ask for **complete** sets of analytic functions $f_n$. This is a set such that any $f \in \mathcal{A}$ is a limit of linear combinations of the $f_n$. Integrating against such a complete set then provides a characterisation of whether a hyperfunction vanishes (generalising (1.7)):

$$\mathcal{I}_z F = 0 \;\; \in \mathcal{HF} \qquad \Leftrightarrow \qquad (f, F) = 0 \quad \forall f \in \mathcal{A} \qquad \Leftrightarrow \qquad (f_n, F) = 0 \quad \forall n \;. \tag{1.11}$$

Having understood the proper meaning of completeness, i.e. property 1), we are still left with the task of constructing not only complete sets of $f_n$, but ones that are dual to particular

---

[2]Formally, hyperfunctions on an interval $V$ are defined as functions analytic on a complex neighbourhood of $V$ but singular at $V$, up to equivalence by adding functions that are non-singular at $V$. In this sense, the hyperfunction "only cares" about the discontinuity of $F$.

[3]For CFT 4-point correlators, it was argued in [23] that the discontinuities are tempered distributions, with the dual space of linear functionals being the Schwartz space of test functions. We comment further on the relation of their work with ours in footnote 12 and Section 3.4.1.

spectra of blocks, i.e. satisfying property 2). We have found a natural way to obtain both these properties. The key idea is that, while $\mathcal{I}_z F$ is taken to be a hyperfunction, it may decomposed into the discontinuities $\mathcal{I}_z G_\Delta$ of individual blocks thanks to the BOE (1.1). Each of these is actually a function, and they are naturally associated to a Hilbert space: $L^2$ functions on $[1, \infty)$. This $L^2$ space has a continuous embedding as a dense subspace of $\mathcal{HF}$. In turn, the analytic functional kernels are also naturally identified with a dense subspace of $L^2$. Thus we naturally arrive at a hierarchy of spaces called a *Gelfand triple* (or *rigged Hilbert space*)

$$\mathcal{A} = \left\{ \substack{\text{real-analytic} \\ \text{functional kernels}} \right\} \subset L^2 \subset \mathcal{HF} = \{\text{hyperfunctions}\} . \tag{1.12}$$

The $L^2$ space will buy us technical power: it is a Hilbert space with well-behaved notions of basis and decomposition. In particular, one can already guess that a useful approach may be first to construct bases of $L^2$ consisting of analytic functions, and then argue that they form the desired complete sets in $\mathcal{A}$, leading to property 1).

To achieve property 2), we have to introduce one last idea: we should work not with orthonormal bases for $L^2$, but with the more general *Riesz bases* [25]. A Riesz basis can be seen as a structure-preserving "skewing" of an orthonormal basis: it is the image of an orthonormal basis under a bounded, invertible, linear map. While Riesz basis elements $\{p_n\}$ are not necessarily orthonormal, they automatically possess another, dual Riesz basis $\{h_n\}$ such that

$$\langle h_n \,|\, p_m \rangle = \delta_{nm} \tag{1.13}$$

This suggests choosing sets $\Delta_n$ such that the discontinuities $p_n = \mathcal{I}_z G_{\Delta_n}$ form a Riesz basis, and defining the functional kernels to be the dual Riesz basis, $f_n = h_n$. In this way we have the desired duality:

$$\theta_{f_n}(\Delta_m) = \langle f_n \,|\, \mathcal{I}_z G_{\Delta_m} \rangle = \delta_{nm} \tag{1.14}$$

and the completion of our program.

As well as understanding which choices of $\Delta_m$ form Riesz bases for $L^2$, we must establish which of those can be mapped back to the spaces $\mathcal{A}$ and $\mathcal{HF}$. In general this is a complicated mathematical problem. Our strategy will be first to reformulate the functionals dual to the "free" spectrum $\Delta_n = 2\tilde{\alpha} + 2n$ (constructed in our previous work [15]) in this language, showing that they satisfy all the required properties to give a complete set of functional kernels in $\mathcal{A}$. After that, we will show that the deformation $\Delta_n \to 2\tilde{\alpha} + 2n + \gamma_n$, with $\gamma_n$ satisfying the assumptions above, does not modify this conclusion. In particular the decay of $\gamma_n$ is necessary to ensure that we still have a Riesz basis, while the fact that the $\gamma_n$ admits an analytic continuation for complex $n$ is required to establish that the dual Riesz basis elements $h_n$ (which are *a priori* just $L^2$ functions) are actually analytic.

Finally, we will study the space $\mathcal{PW}$ of functional actions $\theta(\Delta)$, defined [4] as

$$\mathcal{PW} = \left\{ \begin{array}{l} \theta : \mathbb{C} \longrightarrow \mathbb{C} \\ \theta(\Delta) = \langle f \,|\, \mathcal{I}_z G_\Delta \rangle \;\; \text{for } f \in \mathcal{A} \end{array} \right\} . \tag{1.15}$$

These functions of the complex variable $\Delta$ are physically very significant: they encode all of the valid sum rules $\sum_\Delta \mu_\Delta \lambda_\Delta^{12} \theta(\Delta) = 0$ characterising the locality problem. By studying the transform $\langle f \,|\, \mathcal{I}_z G_\Delta \rangle$, we will show that $\mathcal{PW}$ is also a dense subspace of a Hilbert space, for which the $\theta_{f_n}(\Delta)$ provide a (Schauder) basis. Most importantly, we provide an independent definition of $\mathcal{PW}$ as a space of entire functions with prescribed behaviour near infinity. Such entire functions given by an explicit product of their zeros, leading to the third result above.

**Outline.**   The contents of the paper are organised as follows.

- In Section 2, we set out the problem of finding complete sets of sum rules for locality in terms of the space of hyperfunctions and its dual space of analytic functionals.

- Section 3 explains how it is possible to construct Schauder bases for these spaces by using an auxiliary Hilbert space that sits between them. The outcome is a large class of complete sets of interacting functionals and associated sum rules.

- In Section 4, we consider the space of functional actions, and show that it inherits the same Hilbert space structure of the space of functional kernels. We give an independent characterization of this space as a set of entire functions with certain exponential type. With this definition, we write down explicit expressions for the functional actions forming our interacting bases.

- In Section 5, we verify our construction in an explicit example. First we set out the general relationship between the large-$\Delta$ behaviour of the functional actions and the asymptotics of the functional kernels. Then we start from a known family of extremal 4-point correlators solving the 1d crossing equation. By constructing a basis of interacting functionals dual to the spectrum appearing in these correlators' OPEs, we reconstruct local form factors for bulk operators consistent with these correlators.

- In Section 6, we discuss the implications of our results and future directions of study.

- Several technical appendices complement the paper. Appendix A reviews the results of our paper [15] and some useful asymptotic formulae. The most non-trivial technical work is: proving in Appendix B that our sparse sets of discontinuities of blocks form Riesz bases; and proving in Appendix C that the dual functional kernels are real-analytic functions of the appropriate class. Appendix D addresses the case of the most general values of $\tilde{\alpha}$ and the finite subtraction procedure relevant for that case. Appendix E is a technical complement to Section 5.

This subject being rather mathematical and technical, we have attempted to tread a fine line (barring inevitable missteps) between rigour and accessibility — with a physicist's penchant for the latter.

---

[4]The definition given here is, for simplicity, somewhat schematic. See Section 4 for precise definitions.

# 2   Locality bootstrap and hyperfunctions

## 2.1   Form factors as hyperfunctions

Let us review the properties of form factors and show there is a natural mapping of these objects into a certain space of hyperfunctions. Crucially, this mapping will be defined in a manner compatible with the BOE decomposition (1.1). As in [15], we are considering form factors $\langle \Psi \,|\, \mathcal{O}_1 \, \mathcal{O}_2 \rangle$ for bulk scalar fields $\Psi$, so that all the primary operators $\mathcal{O}_\Delta$ appearing in the BOE are scalars. In this paper, we will set:

$$\Delta_1 = \Delta_2 \qquad \text{and} \qquad d < 4 \ , \tag{2.1}$$

with $d$ the spacetime dimension of the boundary CFT. These assumptions simplify some technicalities, roughly corresponding to avoiding needing certain subtractions; but we do not expect them to be decisive for our general approach.

We will call *form factor* any function $F(z)$ satisfying certain properties described below, which imply in particular that $F(z)$ is analytic on the cut plane $\mathbb{C} \setminus ((-\infty, 0] \cup [1, \infty))$. A *local* form factor is one that is actually analytic on $\mathbb{C} \backslash (-\infty, 0]$, i.e. with no discontinuity at $[1, \infty)$ — solving the locality constraint (1.3).

The first property is that $F(z)$ should admit a BOE expansion,[5]

$$F(z) = \sum_\Delta \mu_\Delta \lambda_\Delta^{12} \, G_\Delta(z) = \sum_\Delta c_\Delta \, g_\Delta(z) \ , \tag{2.2}$$

where, for convenience, we have changed normalisation by setting

$$c_\Delta = \mathcal{N}_\Delta^{-1} \, \mu_\Delta \, \lambda_\Delta^{12} \ , \qquad g_\Delta(z) = \mathcal{N}_\Delta \, G_\Delta(z) \ , \tag{2.3}$$

with

$$\mathcal{N}_\Delta := \frac{\Gamma(\frac{\Delta}{2})^2}{\pi \, \Gamma(\Delta + 1 - \frac{d}{2})} \ , \qquad G_\Delta(z) = z^{\frac{\Delta}{2}} \, {}_2F_1(\tfrac{\Delta}{2}, \tfrac{\Delta}{2}; \Delta + 1 - \tfrac{d}{2}; z) \ . \tag{2.4}$$

This is an important physical consequence of the state-operator correspondence in AdS (see discussion in [15] and refs. there).

The second property is to do with the sense in which the BOE converges, and it implies that $F(z)$ satisfies certain boundedness properties near the branch points at $z = 1$ and $\infty$. Concretely, we will ask that

$$|F(z)| \le \sum_\Delta |c_\Delta| |g_\Delta(z)| \sim \begin{cases} |z|^{\alpha_F} & \text{as } z \to \infty \\ |z - 1|^{-\alpha_F} & \text{as } z \to 1 \end{cases} \tag{2.5}$$

---

[5]The sum over states is implicitly restricted to unitary $\Delta \in \{0\} \cup [\frac{d-2}{2}, \infty)$.

for some fixed number $\alpha_F$.[6] This specific polynomial bound is physically motivated by considering the bulk QFT's UV completion: see [26]. Note that these constraints imply that $F(z)$ is analytic on $\mathbb{C} \setminus \big((-\infty, 0] \cup [1, \infty)\big)$.

Let us now examine the discontinuities of these functions on the cut $[1, \infty)$. It is natural to think of these discontinuities as elements of a space of *hyperfunctions*. A hyperfunction on an interval is defined to be an equivalence class of functions which are analytic in a complex neighbourhood of the interval and which share the same discontinuity (see, e.g., the textbook [27]). Clearly, given one representative in the equivalence class, an equally good one is obtained by adding any analytic function. Choosing once and for all a representative, let us define the *subtracted form factor*,

$$ F(z) \quad \rightarrow \quad F^{(\tilde{\alpha})}(z) = \oint_\Gamma \frac{dw}{2\pi i} \frac{F(w)\, w^{-1-\tilde{\alpha}}}{w - z} \ , \qquad z \in \mathbb{C} \setminus [1, \infty) \tag{2.6} $$

where $\Gamma$ is a contour wrapping the unphysical cut $[1, \infty)$, as shown in Figure 1. $F^{(\tilde{\alpha})}$ is an analytic function on $\mathbb{C} \setminus [1, \infty)$ that vanishes if and only if the original form factor is local

$$ F^{(\tilde{\alpha})} \equiv 0 \quad \Longleftrightarrow \quad \mathcal{I}_{z \geq 1} F = 0 \ . \tag{2.7} $$

This is because its only singularities are at $[1, \infty)$, given by

$$ \mathcal{I}_z F^{(\tilde{\alpha})} = z^{-1-\tilde{\alpha}} \mathcal{I}_z F \ , \qquad 1 \leq z < \infty \ . \tag{2.8} $$

In order to produce this hyperfunction capturing only the unphysical cut, we had to integrate in between the two cuts $[1, \infty)$ and $(-\infty, 0]$ that touch at $\infty$ (see Figure 1). To make the integral (2.6) well-defined, we required a subtraction by a large enough power of $w$:

$$ \tilde{\alpha} > \alpha_F - 1 \ . \tag{2.9} $$

We note that the large-$z$ behaviour of the subtracted form factor is now:

$$ |F^{(\tilde{\alpha})}(z)| \stackrel{z \to \infty}{\sim} |z|^{-1-\tilde{\alpha}+\alpha_F} = |z|^{-\epsilon} \ , \qquad \epsilon > 0 \ , , \tag{2.10} $$

whereas behaviour near $z = 1$ is the same as $F$:

$$ |F^{(\tilde{\alpha})}(z)| \stackrel{z \to 1}{\sim} |z - 1|^{-\alpha_F} = |z - 1|^{-\tilde{\alpha}-1+\epsilon} \ , \qquad \epsilon > 0 \ . \tag{2.11} $$

These considerations motivate the following definition of a space $\mathcal{H}F$ of hyperfunctions

---

[6]In fact for our purposes a weaker assumption would be sufficient: see equation (3.39) and the discussion in Section 3.4.1.

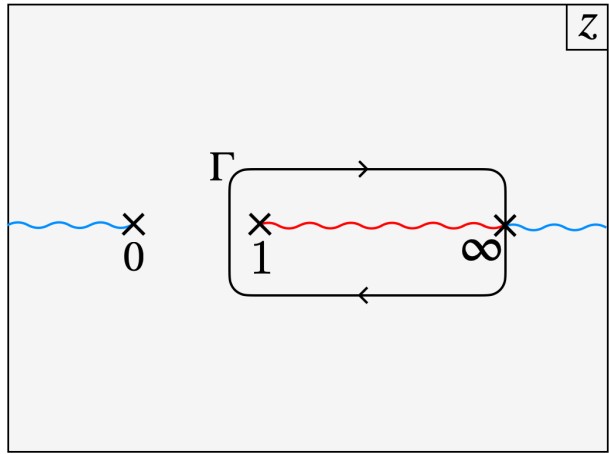

**Figure 1:** The integration contour $\Gamma$ used to define the subtracted form factor $F^{(\tilde{\alpha})}$ in equation (2.6). The contour goes through $\infty$, passing between the two cuts meeting there (the physical one in blue and the unphysical one in red).

with properties similar to those of $F^{(\tilde{\alpha})}$:[7]

$$\mathcal{HF} = \left\{ H(z) \text{ analytic on } \mathbb{C} \setminus [1, \infty) \,\middle|\, \begin{array}{l} |H(z)| \overset{z \to \infty}{\sim} |z|^{-\epsilon}, \; |H(z)| \overset{z \to 1}{\sim} |z-1|^{-\tilde{\alpha}-1+\epsilon} \\ \text{for some } \epsilon > 0 \end{array} \right\}. \quad (2.12)$$

We have seen that for any form factor $F$, its subtraction $F^{(\tilde{\alpha})}$ is an element of this space (provided $\tilde{\alpha} > \alpha_F - 1$). Our task will be to understand under which circumstances $F^{(\tilde{\alpha})}$ coincides with the zero element of $\mathcal{HF}$, since this is precisely the locality condition (2.7) for the form factor.

An important property of the map from form factors to $\mathcal{HF}$ is that it is nicely compatible with the BOE. In particular, the BOE descends to a convergent decomposition of the subtracted form factor in $\mathcal{HF}$:

$$F^{(\tilde{\alpha})} = \sum_{\Delta} c_{\Delta} \, g_{\Delta}^{(\tilde{\alpha})} \,, \quad (2.13)$$

where $g_{\Delta}^{(\tilde{\alpha})}$ are the hyperfunctions obtained by applying the subtraction (2.6) to $g_{\Delta}$. The above sum converges in the space of hyperfunctions with its natural *compact-open topology* — meaning it **converges uniformly on compact regions** away from the cut $[1, \infty)$.[8]

---

[7]It may seem odd that the space $\mathcal{HF}$ depends on the parameter $\tilde{\alpha}$, which may be chosen freely as long as $\tilde{\alpha} > \alpha_F - 1$. This subtlety is addressed below in Section 3.4.1.

[8]Formally, because of the bounds at the endpoints of the cut, we are using a slightly finer topology than the standard compact-open one: we add some extra open sets so that $H_n \to 0$ in $\mathcal{HF}$ also implies $\int_{\infty} |dz||z|^{-1}|H_n(z)| \underset{n \to \infty}{\to} 0$ and $\int_1 |dz||z-1|^{\tilde{\alpha}}|H_n(z)| \underset{n \to \infty}{\to} 0$ on arbitrary integration contours touching the endpoints $\infty$ and $1$. In the same vein, one may understand the definition of the space (2.12) more formally as requiring $\int_{\infty} |dz||z|^{-1}|H(z)| < \infty$ and $\int_1 |dz||z-1|^{\tilde{\alpha}}|H_n(z)| < \infty$. One can check that the space defined in this way is complete. The dual space $\mathcal{A}$ defined in (2.17) may also be formalised in a compatible way.

To see that (2.13) converges in this way, we will use the fact that a **pointwise** limit of analytic functions converges **uniformly** on compact sets inside the shared domain of analyticity of the functions and their limit.[9] Hence it is sufficient to show that (2.13) converges pointwise away from $[1, \infty)$. To prove this, let us apply the subtraction transform (2.6) to the BOE (2.2): if we can commute the integral with the sum over $\Delta$, then we are done. This is indeed possible by splitting the integral into two regions: one near $\infty$ and one elsewhere covered by a compact domain. For the region near $\infty$, the bound in (2.5) allows to swap the sum and integral by Dominated Convergence. On the other region covered by a compact domain, the BOE (2.2) converges uniformly by the fact mentioned above, so the sum and integral can be swapped.

Before moving on let us make a quick observation. The formulation (2.7) of locality as the vanishing of $F^{(\tilde{\alpha})}$ is closely related to the dispersion relation of [15] since we can write

$$F^{(\tilde{\alpha})}(z) = z^{-1-\tilde{\alpha}} \left( F(z) - \oint_{\Gamma_{[-\infty,0]}} \frac{\mathrm{d}w}{2\pi i} \frac{(z/w)^{1+\tilde{\alpha}}}{w - z} F(w) \right) .$$ (2.14)

where the dispersion relation is the vanishing of the RHS. Applying this relation to a single block gives a relation between subtracted blocks $g_\Delta^{(\tilde{\alpha})}$ in (2.13) and the 'local blocks' of [15]:

$$g_\Delta^{(\tilde{\alpha})}(z) = \mathcal{N}_\Delta \, z^{-1-\tilde{\alpha}} \left( G_\Delta(z) - \mathcal{L}_\Delta^{\tilde{\alpha}}(z) \right) .$$ (2.15)

## 2.2 Functionals and locality conditions

Now that we have mapped form factors into hyperfunctions, we will describe how such objects can be probed by the action of linear functionals. As discussed in the introduction, $\mathcal{HF}$ is a very broad class of distributions. Like for other classes of distributions, they may be defined as the continuous linear functionals acting on a space of test functions:[10]

$$\mathcal{HF} = \mathcal{A}' ,$$ (2.16)

where the test functions $\mathcal{A}$ are the **real-analytic functions** on the interval. In our case, the hyperfunctions have particular bounded behaviour at the endpoints 1 and $\infty$ (cf. (2.10)

---

[9]A precise statement (a result of Osgood [28]; see the review in [29]) is that, if $g_\Delta$ are holomorphic on a certain domain and the series $F = \sum_\Delta c_\Delta g_\Delta$ converges pointwise, then $F$ is holomorphic on a dense, open subset $\Omega$ of the domain, and the convergence is uniform in compact subsets of $\Omega$. If $F$ is holomorphic on (at least) the same domain as $g_\Delta$, then $\Omega$ is the whole domain.

[10]The identification of hyperfunctions as the dual to real-analytic functions on a compact real domain is due to the Grothendieck-Köthe-Silva duality. The compact-open topology that we use on $\mathcal{HF}$ is then the natural *strong-dual topology* inherited from this duality.

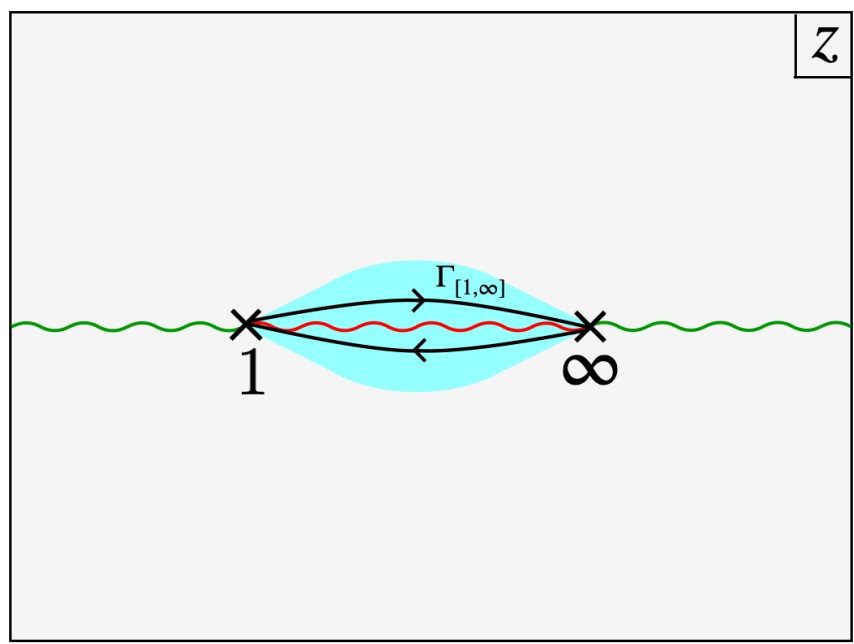

**Figure 2:** The integration contour (in black) defining the action of functionals on subtracted form factors. The shared domain of analyticity is shown in blue, the branch cut of the form factor in red and possible non-analyticities of the functional kernel in green.

and (2.11)). Hence the test function space can be taken to be:

$$\mathcal{A} = \left\{ f(z) \text{ real-analytic on } (1,\infty) \left| \begin{array}{c} \text{analytic on finite angle at endpoints} \\ |f(z)| \underset{z\to\infty}{\lesssim} |z|^{-1} \\ f(z) \underset{z\to 1}{\sim} (\text{analytic}) + |z-1|^{\tilde{\alpha}} \end{array} \right. \right\} . \qquad (2.17)$$

A typical domain of analyticity for an element of $\mathcal{A}$ is shown in blue in Figure 2. For example, $f(z)$ may have branch cuts running along $(-\infty, 1)$, as well as other singularities away from, but arbitrarily close to, the interval $[1, \infty]$.

The definition (2.16) means there is a continuous dual pairing between $\mathcal{A}$ and $\mathcal{HF}$, defined by

$$(f, G) := \oint_{\Gamma_{[1,\infty]}} \frac{\mathrm{d}z}{2\pi i} f(z) G(z) , \qquad f \in \mathcal{A} , \quad G \in \mathcal{HF} , \qquad (2.18)$$

where $\Gamma_{[1,\infty]}$ is a contour wrapping the cut $[1, \infty]$, inside the shared domain of analyticity of $F$ and $f$. Since $f$ is permitted non-analyticity at the end points, the contour must in general be 'pinched' to go through the points 1 and $\infty$, approaching these endpoints at a finite angle from the cut $[1, \infty]$ (see Figure 2).[11]

---

[11]The analytic term allowed in the functionals at $z = 1$ is understood as acting on a contour pushed to the left at $z = 1$ to avoid the singularity of the hyperfunction.

We thus identify $\mathcal{A}$ as a natural class of functional kernels acting on $\mathcal{HF}$ as[12]

$$\theta_f[H] := (f, H) \qquad f \in \mathcal{A} , \quad H \in \mathcal{HF} . \tag{2.19}$$

Since the pairing $(\bullet, \bullet)$ is continuous, then these functionals are automatically continuous. Formally, what is happening here is that there is a canonical embedding $\mathcal{A} \to \mathcal{A}'' = \mathcal{HF}'$, which maps $f \mapsto \theta_f$. In this natural mathematical language, the *swapping* property of [24] becomes automatic: the functionals (2.19) are compatible with the BOE:

$$\theta_f \left[ F^{(\tilde{\alpha})} \right] = \sum_\Delta c_\Delta \, \theta_f \left[ g_\Delta^{(\tilde{\alpha})} \right] . \tag{2.20}$$

This follows because the functionals are continuous and the BOE (2.13) converges in the topology of $\mathcal{HF}$.

Let us summarise what we have found so far. Firstly, we described a mapping of form factors into the space $\mathcal{HF}$, such that

$$F = \sum_\Delta c_\Delta \, g_\Delta \text{ local } \quad \Longleftrightarrow \quad F^{(\tilde{\alpha})} = \sum_\Delta c_\Delta \, g_\Delta^{(\tilde{\alpha})} = 0 \quad \in \mathcal{HF} \tag{2.21}$$

Secondly, we showed there is a dual space of objects which act on hyperfunctions continuously. These objects are analytic functions on the interval $(1, \infty)$ with certain boundedness properties at the endpoints. This means in particular that

$$H = 0 \quad \in \mathcal{HF} \quad \Longleftrightarrow \quad \theta_f[H] = 0 \quad \text{for all } f \in \mathcal{A} . \tag{2.22}$$

Combining both of these, we see that:

$$F = \sum_\Delta c_\Delta \, g_\Delta \text{ local } \quad \Longleftrightarrow \quad \sum_\Delta c_\Delta \, \theta_f \left[ g_\Delta^{(\tilde{\alpha})} \right] = 0 \quad \text{for all } f \in \mathcal{A} \tag{2.23}$$

At this point let us recall our two initial goals defined in the introduction: to find a complete set of sum rules for locality satisfying duality with respect to a particular spectrum. In the language of hyperfunctions, this becomes the following task:

---

**Task:** Produce a set of functional kernels $\{f_n\}_{n=1}^\infty$ in $\mathcal{A}$ such that:

1)  $\{f_n\}$ are **complete** in $\mathcal{A}$.

2)  $f_n$ are **dual** to a set of blocks, $(f_m, g_{\Delta_n}^{(\tilde{\alpha})}) = \delta_{mn}$, for some $\{\Delta_n\}$. .

---

[12]The analytic functionals of the type used in [17,19,30] clearly fall into the present class of functionals $\mathcal{A}$. Hence they are also in the larger class of smooth functional kernels considered in [23]. Note that the standard derivative functionals are also in our class: $\partial_z^n H(\frac{1}{2}) = D_n[H] := (n!(z - \frac{1}{2})^{-n-1}, H)$.

The word 'complete' now has precise meaning: $\{f_n\}$ is a *complete* set in the topological vector space $\mathcal{A}$ if any element of $\mathcal{A}$ is a limit of linear combinations of the $f_n$.[13] Together with (2.23), this implies that the associated sum rules are complete in the sense of Property 1) from the introduction:

$$F = \sum_\Delta c_\Delta \, g_\Delta \text{ local} \quad \Longleftrightarrow \quad \sum_\Delta c_\Delta \, \theta_n(\Delta) = 0 \quad \text{for all } n \geq 1 \;, \tag{2.24}$$

where we have defined

$$\theta_n(\Delta) := \theta_{f_n}\big[g_\Delta^{(\tilde{\alpha})}\big] = (f_n, \, g_\Delta^{(\tilde{\alpha})}) \;. \tag{2.25}$$

In particular, the second element of this task allows us then to achieve Property 2) as stated in the introduciton.

## 2.3   Generalized Free Field functionals

Let us now rephrase the results of our previous work [15] in the language of the preceding discussion. In that work we produced a set of analytic functions $\hat{f}_n(z)$, given explicitly in Appendix A. They are simple polynomials of degree $n$ in $1/z$ without constant term, and thus clearly belong to $\mathcal{A}$. They were found by demanding that they satisfy the duality property with respect to the sparse spectrum

$$\hat{\Delta}_n = 2\tilde{\alpha} + 2n \;, \tag{2.26}$$

i.e. a Generalized Free Field (GFF) spectrum of double-traces (with a hat denoting GFF quantities). To see that $\{\hat{f}_n(z)\}$ are complete is, in this case, rather simple, since one can just appeal to the fact that, for any value of $\tilde{\alpha}$, $\{z\hat{f}_n(z)\}$ are a complete set of polynomials (one of each order). It is a mathematical fact that such functions are dense in the space of real analyic functions, and it then follows that $\{f_n\}$ are dense in $\mathcal{A}$.

Suppose however that we were not aware of this mathematical fact. How can we convince ourselves that the GFF functional kernels are complete? We will now set out a proof of this property that will generalise to the case of functionals dual to interacting spectra $\Delta_n$ — whose kernels will no longer be polynomials. The idea is to trade the completeness condition in our task for something equivalent:

---

[13]Let us make a technical comment about the topology of $\mathcal{A}$. There are several natural topologies on this space, but since we are only using $\mathcal{A}$ as functionals acting on $\mathcal{HF}$, it suffices for us to equip it with the *weak topology*. This means that convergence $f_n \to f$ in $\mathcal{A}$ just means that the action on any hyperfunction converges: $(f_n, H) \to (f, H)$ for any $H \in \mathcal{HF}$.

**Task':** Produce a set of functionals $\{f_n\}_{n=1}^\infty$ in $\mathcal{A}$ such that:

1') Any $H \in \mathcal{HF}$ **decomposes** as $H = \sum_n (f_n, H)\, g_{\Delta_n}^{(\tilde{\alpha})}$,

2) $f_n$ are **dual** to a set of blocks, $(f_m, g_{\Delta_n}^{(\tilde{\alpha})}) = \delta_{mn}$, for some $\{\Delta_n\}$.

This immediately implies the completeness condition 1) above, as follows: for any $f \in \mathcal{A}$, acting with it on the equation in 1'), we have

$$(f, H) = \sum_n (f_n, H)\,(f, g_n^{(\tilde{\alpha})}) = \Big( \sum_n (f, g_n^{(\tilde{\alpha})}) f_n,\ H \Big) \quad \forall H \in \mathcal{HF},\ f \in \mathcal{A}\ . \tag{2.27}$$

This means that (in the weak topology)

$$f = \sum_n (f, g_n^{(\tilde{\alpha})})\, f_n \qquad \forall f \in \mathcal{A}\ . \tag{2.28}$$

Hence $f_n$ are dense in $\mathcal{A}$. In fact, we learn something even more than this: $\{f_n\}$ and $\{g_{\Delta_n}^{(\tilde{\alpha})}\}$ are **Schauder bases** for $\mathcal{A}$ and $\mathcal{HF}$ respectively. This means that all elements have unique decompositions in terms of the basis elements.

**Proof of Property 1') for GFF functionals.** To show that the GFF functionals (A.7) indeed satisfy property 1'), consider the 'master functional' of [15],

$$f_w(z) = \frac{1}{z - w}\ . \tag{2.29}$$

This is a 1-parameter family of functionals that completely characterises locality:

$$F\ \text{local} \quad \Longleftrightarrow \quad \theta_{f_w}[F^{(\tilde{\alpha})}] = 0\ . \tag{2.30}$$

That is a simple consequence of Cauchy integral formula

$$\theta_{f_w}[H] = \oint_\gamma \frac{dz}{2\pi i} \frac{1}{z - w} H(z) = H(w)\ , \qquad H \in \mathcal{HF}\ , \tag{2.31}$$

where $\gamma$ can be taken as any contour wrapping $[1, \infty]$, and with $w$ on its outside.

Now we recall [15] that the master functional admits an expansion into the GFF functionals (A.7):

$$\frac{1}{z - w} = w^{-\tilde{\alpha}-1} \sum_{n=1}^\infty \hat{f}_n(z)\, g_{\hat{\Delta}_n}(w)\ . \tag{2.32}$$

One can check that this is true, at least as a formal Taylor series for to each order $O(w^N)$. Moreover, as we shall now establish, it actually converges as an analytic function of $z$ and as a hyperfunction of $w$: i.e. for $z$ in some neighbourhood of $[1, \infty]$ and for $w$ away from $[1, \infty]$. This follows by the estimate given in Appendix A:

$$\hat{f}_n(z)\, g_{\hat{\Delta}_n}(w) \overset{n\to\infty}{\sim} \left(\frac{r(w)}{r(z)}\right)^{\tilde{\alpha}+n} \times (\ldots) , \tag{2.33}$$

where we only show the dependence on $n$ and where $r(z)$ maps $\mathbb{C}\backslash[1, \infty]$ to the disk $|r| < 1$,

$$r(z) = \frac{1 - \sqrt{1-z}}{1 + \sqrt{1-z}} . \tag{2.34}$$

It follows that the sum converges absolutely for $|r(w)| < |r(z)| \le 1$, as required.

We then observe that

$$w^{-\tilde{\alpha}-1}\, g_{\hat{\Delta}_n}(w) = g^{(\tilde{\alpha})}_{\hat{\Delta}_n}(w) . \tag{2.35}$$

This follows, for example by acting with equation (2.32) on $g^{(\tilde{\alpha})}_{\hat{\Delta}_n}(z)$ and using the duality $(f_m, g_n^{(\tilde{\alpha})}) = \delta_{mn}$. Hence we may re-write (2.32) as

$$\frac{1}{z-w} = \sum_{n=1}^{\infty} \hat{f}_n(z)\, g^{(\tilde{\alpha})}_{\hat{\Delta}_n}(w) . \tag{2.36}$$

Now substituting this decomposition into (2.31), one finds

$$H(w) = \sum_{n=1}^{\infty} (\hat{f}_n, H)\, g^{(\tilde{\alpha})}_{\hat{\Delta}_n}(w) \qquad \text{for all} \quad H \in \mathcal{HF} , \tag{2.37}$$

where commuting the series with the integral was possible because the sum converges as an analytic function of $w$ away from $[1, \infty]$. Note that our results imply that (2.37) converges in $\mathcal{HF}$, proving Property 1').

## 3 A general construction: interacting functionals

In this section we will formally establish the existence of a large set of solutions of the conditions 1') and 2), leading to a large set of Schauder bases. A practical method to construct the corresponding kernels will later be given in Section 5, and an explicit expression for the functionals' actions on blocks will be given in Section 4.

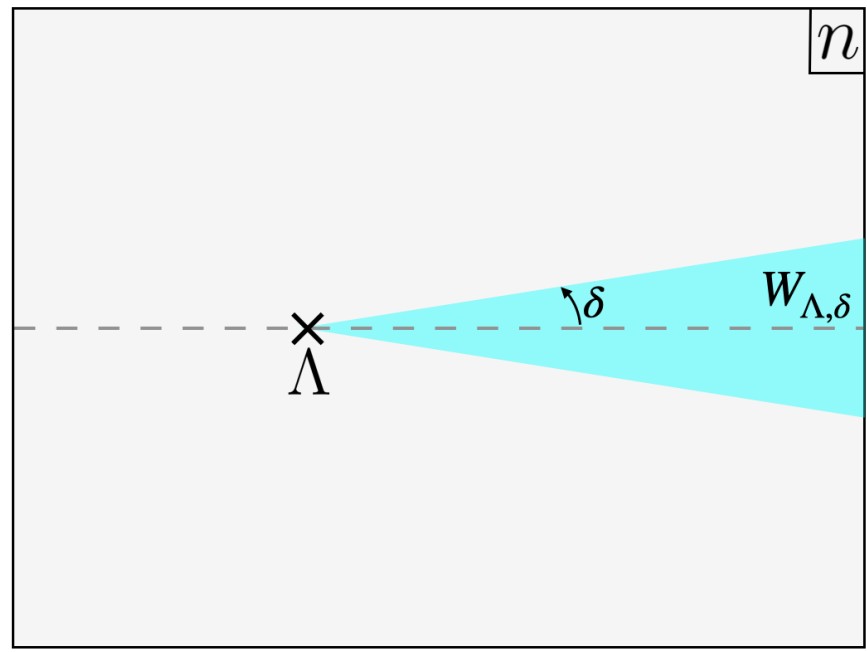

**Figure 3:** The 'wedge' domain of analyticity $W_{\Lambda,\epsilon}$ (in blue) assumed for the functions $\alpha(n)$ and $\beta(n)$ that interpolate the anomalous dimensions $\gamma_n$ according to equation (3.2). The dashed line is the real axis.

Concretely, we will produce functionals dual to a large class of possible $\{\Delta_n\}$,

$$\Delta_n = \hat{\Delta}_n + \gamma_n \, , \qquad (\hat{\Delta}_n = 2\tilde{\alpha} + 2n) \tag{3.1}$$

which may be seen as 'small' deformations of GFF spectra. In detail, we assume that $\gamma_n$ takes the form

$$\gamma_n = \alpha(n) + (-1)^n \beta(n) \, , \qquad \alpha(n), \beta(n) \underset{n\to\infty}{=} O(n^{-\epsilon}) \, , \quad \epsilon > 0 \, , \tag{3.2}$$

with $\alpha(n)$ and $\beta(n)$ analytic in $n$ in some complex 'wedge' (see Figure 3),[14]

$$W_{\Lambda,\delta} := \left\{ z \in \mathbb{C} \mid |\arg(n - \Lambda)| < \delta \right\} \, , \qquad \Lambda > 0 \, , \delta > 0 \, . \tag{3.3}$$

Note that it is sufficient to have $\Lambda$ quite large (and $\delta$ quite small). We will call such spectra '**sparse**' and, for any sparse spectrum, we will argue the existence of a dual basis of functionals satisfying Properties 1') and 2) above.

---

[14]For the case $d = 1$, we need to assume that the oscillating term is slightly more suppressed, $\beta(n) \sim n^{-\frac{1}{2}-\epsilon}$ where $\epsilon \geq 0$.

## 3.1 A useful $L^2$ space

### 3.1.1 Embeddings

In order to make functionals dual to interacting spectra (3.1), one may think of 'rotating' from the GFF functionals (A.7) by a 'small' amount. Since $\mathcal{A}$ and $\mathcal{HF}$ are topological vector spaces without norms, it is not clear how to meaningfully measure such a rotation. To do this, we will use the trick of introducing an intermediate $L^2$ space:

$$\mathcal{A} \quad \to \quad L^2_\mu \quad \to \quad \mathcal{HF} \; , \tag{3.4}$$

for an appropriate choice of inner product defined by:

$$\langle f | g \rangle_\mu := \int_1^\infty \mathrm{d}z \, \mu(z) \, f(z) \, g(z) \; , \qquad f, g \in L^2_\mu \; . \tag{3.5}$$

Such a hierarchy of spaces (a Hilbert space in between a space $\mathcal{A}$ and its dual) is called a *Gelfand triple* or *rigged Hilbert space*. As we will see below, in practice, the $L^2$ space will buy us an elegant and user-friendly shortcut for constructing Schauder bases of functionals in $\mathcal{A}$ satisfying 1') and 2). First, we will construct bases for $L^2$ itself — a simpler and more familiar task — and the desired Schauder bases will follow under general conditions.

We need to choose the $L^2$ measure $\mu$ in (3.5) and describe the embeddings in (3.4). A natural choice for $\mu$ follows from the observation that the blocks $g_\Delta$ are eigenfunctions of a Casimir operator (see, e.g., [31]):

$$(C_z - \lambda_\Delta) \, g_\Delta = 0 \; , \qquad \lambda_\Delta = \Delta(\Delta - d) \; , \tag{3.6}$$

with

$$C_z := 4(1 - z)^{1-d/2} z^{1+d/2} \partial_z [(1 - z)^{d/2} z^{1-d/2} \partial_z] \; . \tag{3.7}$$

The Casimir operator is self-adjoint (up to boundary terms) on $L^2$ if we choose the particular measure

$$\mu(z) = \frac{1}{z^2} \Big( \frac{z}{z - 1} \Big)^{1-d/2} \; , \tag{3.8}$$

so let us choose this one. With respect to this measure, the discontinuities of blocks are square-integrable: $\mathcal{I}_z g_\Delta \in L^2_\mu$. Henceforth, we will omit the subscript $\mu$, but let us emphasise that there is only one $L^2$ norm used in this paper, defined with the measure (3.8).

Next, we need to describe the arrows appearing in the diagram (3.4), i.e. the embeddings of $\mathcal{A}$ into $L^2$ and of $L^2$ in $\mathcal{HF}$. We would like the $L^2$ functions $\mathcal{I}_z g_\Delta$ to correspond to the subtracted blocks $g_\Delta^{(\tilde{\alpha})}$ in the hyperfunction space, so let us define the following $\tilde{\alpha}$-dependent

embeddings:[15]

$$i^{(\tilde{\alpha})} : \mathcal{A} \to L^2 \, , \qquad\qquad\qquad j^{(\tilde{\alpha})} : L^2 \to \mathcal{HF} \, , \qquad\qquad (3.9)$$

$$f(z) \mapsto \pi^{-1}\mu(z)^{-1}z^{-\tilde{\alpha}-1}f(z) \qquad \mathcal{I}_z[j^{(\tilde{\alpha})}(G)](z) = z^{-\tilde{\alpha}-1}G(z) \, . \qquad (3.10)$$

When unambiguous, we will use the streamlined notation

$$f^{(\tilde{\alpha})} = i^{(\tilde{\alpha})}(f) \, , \quad f \in \mathcal{A} \, , \qquad G^{(\tilde{\alpha})} = j^{(\tilde{\alpha})}(G) \, , \quad G \in L^2 \, . \qquad (3.11)$$

The hyperfunction defining $j^{(\tilde{\alpha})}(G)$ may be written down explicitly as

$$j^{(\tilde{\alpha})}(G)(z) = \int_1^\infty \frac{\mathrm{d}w}{\pi} \frac{G(w)}{w^{\tilde{\alpha}+1}(w-z)} = \left\langle \frac{1}{\pi\mu(w)w^{\tilde{\alpha}+1}(w-z)} \middle| G(w) \right\rangle . \qquad (3.12)$$

Importantly, we have the desired property $j^{(\tilde{\alpha})}(\mathcal{I}_z g_\Delta) = g_\Delta^{(\tilde{\alpha})}$.

The mappings $i^{(\tilde{\alpha})}$ and $j^{(\tilde{\alpha})}$ have several important properties:

- The maps are **compatible** in the sense

$$(f, j^{(\tilde{\alpha})}(G)) = \langle i^{(\tilde{\alpha})}(f) \, | \, G \rangle \, , \qquad\qquad f \in \mathcal{A} \, , \ \ G \in L^2 \, . \qquad (3.13)$$

- Both $i^{(\tilde{\alpha})}$ and $j^{(\tilde{\alpha})}$ are **continuous**, i.e. they preserve convergence of sequences of functions. For $i^{(\tilde{\alpha})}$ this is trivial while for $j^{(\tilde{\alpha})}$ it follows from (3.12) and the Cauchy-Schwarz inequality as long as $\tilde{\alpha} > -\frac{1}{2}$.

- Both $i^{(\tilde{\alpha})}$ and $j^{(\tilde{\alpha})}$ are **injective** (i.e. one-to-one).

- Both $i^{(\tilde{\alpha})}$ and $j^{(\tilde{\alpha})}$ have **dense image**.[16]

Altogether, these properties allow us to re-express the dual pairing of arbitrary elements of $\mathcal{A}$ and $\mathcal{HF}$ in terms of $L^2$ inner products, over which we will have better technical control.

### 3.1.2 Riesz bases

Now that we have set up the hierarchy of spaces, we would like to profit from the fact that $L^2$ is a Hilbert space by introducing a basis. Given the requirement 1') of our task, it seems natural to look for bases $\{p_n\}$ made out of discontinuities of blocks (i.e. the $L^2$ members

---

[15]To make these maps well-defined (with good enough behaviour at $z = 1$), we require that $\tilde{\alpha} > \frac{d}{4} - 1$. Note that one can formally construct functionals for other values of $\tilde{\alpha}$ but the statements of completeness in the spaces $\mathcal{A}$ and $\mathcal{HF}$ as defined here apply only in this case.

[16]For $i^{(\tilde{\alpha})}$, this is related to the statement that the subset of $L^2$ functions suppressed by an extra power at $\infty$ is still dense. For $j^{(\tilde{\alpha})}$, it follows from the statement (2.37) that any hyperfunction can be represented as a sum of hyperfunctions whose discontinuity is in $L^2$.

which map to $g_\Delta^{(\tilde\alpha)}$ under $j^{(\tilde\alpha)}$). A clue that this is the right track is that these discontinuities are given by:

$$\mathcal{I}_{z\geq 1}\, g_\Delta(z) = \frac{\Gamma\left(\frac{\Delta}{2}\right)}{\Gamma\left(\frac{\Delta}{2}+1-\frac{d}{2}\right)}\, \left(\tfrac{z-1}{z}\right)^{1-\frac{d}{2}}\, J_{\frac{\Delta-2}{2}}^{(1-\frac{d}{2},0)}\left(\tfrac{2-z}{z}\right). \tag{3.14}$$

For the particular case of $\Delta_n = 2n$ even integers (i.e. $\Delta_n = 2\tilde\alpha + 2n$ with $\tilde\alpha = 0$), the Jacobi functions reduce to Jacobi polynomials and the discontinuities form an orthonormal basis up to an overall normalisation factor. More generally, we will need to relax the requirement of orthonormality and ask instead that these discontinuities form instead what is known as a **Riesz basis** (see, e.g., the textbook [25]).

A Riesz basis is defined as the image $\{p_n\}$ of an orthonormal basis $\{e_n\}$ under a bounded, invertible map $T$:

$$p_n = T(e_n)\,, \qquad T : L^2 \to L^2 \text{ bounded, invertible .} \tag{3.15}$$

It follows that any Riesz basis automatically comes with a **dual Riesz basis**, consisting of elements $h_n := T^{-1}(e_n)$. These dual bases form a biorthogonal system:

$$\langle h_n | p_m \rangle = \delta_{mn}\,. \tag{3.16}$$

Crucially, any element in the Hilbert space has a unique, unconditionally convergent decomposition with respect to a Riesz basis:

$$f = \sum_n \langle f | h_n \rangle\, p_n = \sum_n \langle f | p_n \rangle\, h_n\,, \qquad \forall f \in L^2\,. \tag{3.17}$$

Riesz bases are exactly the tool we need to achieve our task. On the one hand, by acting with $j^{(\tilde\alpha)}$, we may hope to lift completeness of the $p_n \propto \mathcal{I}_z\, g_{\Delta_n}$ in $L^2$ into completeness in the space of hyperfunctions, i.e. property 1). On the other hand, the dual Riesz basis $\{h_n\}$ are prime candidates for the functional kernels, satisfying the duality property 2) by construction.

### 3.1.3 How to use $L^2$

Now that we have set up this $L^2$ space, let us explain how Riesz bases of $L^2$ can be used to construct complete bases of functionals solving our task. Suppose then that:

(a) $\{p_n := \mathcal{I}_z g_{\Delta_n}\}$ forms a Riesz basis, and denote its dual Riesz basis $\{h_n\}$.[17]

---

[17] In fact what we call $p_n$, and their duals $h_n$, are non-normalised Riesz bases — i.e. they become Riesz bases when normalised instead as $p_n \to n^{\frac{3}{2}-\frac{d}{2}} p_n$ and $h_n \to n^{\frac{d}{2}-\frac{3}{2}} h_n$ (making their norms bounded) — see eq. (B.3) of Appendix B. In a slight abuse of language, we will still call them Riesz basis even when not appropriately normalised. This makes no difference to the properties that we will need.

(b) $h_n = i^{(\tilde{\alpha})}(f_n)$ for some $f_n \in \mathcal{A}$ .

*A priori*, from (a), the $h_n$ are only guaranteed to be $L^2$ functions so that (b) is a non-trivial requirement. It means that $h_n$ is real-analytic on $(1, \infty)$, bounded as $|z|^{-\tilde{\alpha}}$ at $\infty$, and that $h_n(z) \underset{z \to 1}{\sim} (z-1)^{1-d/2}[(\text{analytic}) + O(z-1)^{\tilde{\alpha}}]$. In what follows, one of the major subtleties will be proving this assumption (b) to be satisfied for our class of 'sparse' spectra $\Delta_n$. It will be satisfied exactly as stated here for $\tilde{\alpha} < \frac{1}{2}$; for higher values of $\tilde{\alpha}$, we will instead need to consider particular finite linear combinations of the $h_n$, but the conclusions will be unmodified (see Section 3.3 below).

Let us show that the resulting functionals $f_n \in \mathcal{A}$ then solve the task set out above. Firstly, the duality property (3.16) of Riesz bases give

$$(f_m, g^{(\tilde{\alpha})}_{\Delta_n}) = \langle h_m | p_n \rangle = \delta_{mn} . \tag{3.18}$$

so that the functional kernels $f_n$ satisfy property 2). For property 1'), we may think of using the dual pair of Riesz bases to write down a partition of unity for the Hilbert space $L^2$,

$$\mathbb{1} = |p_n\rangle\langle h_n| \qquad \Leftrightarrow \qquad \delta(z-w) = \mu(z) \sum_{n=1}^{\infty} p_n(w) \, h_n(z) . \tag{3.19}$$

Of course such identities, familiar in quantum mechanics, only make rigorous sense when understood in the language of rigged Hilbert spaces. Since $h_n = i^{(\tilde{\alpha})}(f_n)$ for analytic functionals $f_n \in \mathcal{A}$ in this special class, we can expect such a partition of unity to be valid when acting on hyperfunctions of the $z$ variable. Thus, we may more rigorously re-write the above distributional identity as the discontinuity of the following hyperfunction of $z$ (cf. equation (2.36) in the GFF case):

$$\frac{1}{z-w} = \sum_{n=1}^{\infty} g^{(\tilde{\alpha})}_{\Delta_n}(w) \, f_n(z) . \tag{3.20}$$

This is indeed a 'partition of unity', since the LHS acts as the identity operator on hyperfunctions: $(\frac{1}{z-w}, H(z))_z = H(w)$. As we will explain below, this decomposition is always correct, at least converging in some sense (its image under $i^{(\tilde{\alpha})}_z$ converges in $L^2$). But for it to be useful, we must make an additional assumption that:

(c) The sum (3.20) converges as an analytic function of $z$ and a hyperfunction of $w$.

Then, copying the logic from the GFF case above, Property 1') follows by integrating (3.20) against $H(z)$:

$$H(w) = \left( \frac{1}{z-w}, H(z) \right)_z = \sum_{n=1}^{\infty} (f_n, H) \, g^{(\tilde{\alpha})}_{\Delta_n}(w) . \tag{3.21}$$

Let us come back to derive a version of equation (3.20) as an identity in $L^2$. We would like to decompose the analytic function $\frac{1}{z-w}$ of $z$ as a sum of the functions $f_n(z) \in \mathcal{A}$ — but we do not know how to do that. Instead let us consider the $L^2$ element $i_z^{(\tilde{\alpha})}\left(\frac{1}{z-w}\right)$, which we can decompose into the Riesz basis $h_n(z) = i^{(\tilde{\alpha})}(f_n)(z)$:

$$i^{(\tilde{\alpha})}\left(\frac{1}{z-w}\right) = \sum_n g_{\Delta_n}^{(\tilde{\alpha})}(w)\, i^{(\tilde{\alpha})}(f_n)(z) \tag{3.22}$$

Indeed these are the correct $w$-dependent expansion coefficients: $\langle i_z^{(\tilde{\alpha})}(\frac{1}{z-w})|p_n\rangle = g_{\Delta_n}^{(\tilde{\alpha})}(w)$. The assumption (c) implies that one can act on this equation with $(i_z^{(\tilde{\alpha})})^{-1}$, obtaining (3.20) converging as an analytic function of $z$.

Gathering all of the assumptions we used, we have established (for the range $\tilde{\alpha} < \frac{1}{2}$) the following sufficient conditions in $L^2$ language to produce a basis of functionals $f_n \in \mathcal{A}$ satisfying Properties 1') and 2):

---

**Criteria in $L^2$ language:**    $(\tilde{\alpha} < \frac{1}{2})$

(a) A set of discontinuities of blocks $\{p_n = \mathcal{I}_z\, g_{\Delta_n}\}$ forms a Riesz basis for $L^2$.

(b) The dual Riesz basis $\{h_n\}$ are given by $h_n = i^{(\tilde{\alpha})}(f_n)$ for some $f_n \in \mathcal{A}$ .

(c) The decomposition (3.20) converges as an analytic function of $z$ and a hyperfunction of $w$.

---

## 3.2   Completeness of functionals using $L^2$

We have derived a set of sufficient conditions (a)–(c) for the existence of a complete basis of analytic functionals dual to a given spectrum $\Delta_n$. In this section, we will first show that these conditions are met by the GFF spectrum $\Delta_n = 2\tilde{\alpha} + 2n$, giving an alternate derivation of the completeness of the GFF functionals. Then we will turn to interacting sparse spectra $\Delta_n = 2\tilde{\alpha} + 2n + \gamma_n$ and demonstrate the existence of a dual basis of analytic functionals for any $\gamma_n$ satisfying the assumptions (3.2). The analysis in this section focuses on the case $\tilde{\alpha} < \frac{1}{2}$, but the conclusions are also valid in the general case, which is treated below in Section 3.3.

### 3.2.1   GFF functionals

We begin by showing that the conditions (a)–(c) are satisfied for the spectrum $\hat{\Delta}_n = 2\tilde{\alpha} + 2n$. This will be instructive for proving the same thing for interacting functionals. As above, we use hats to denote GFF quantities (as opposed to interacting ones).

**(a) For any $\tilde{\alpha} \in (-\frac{1}{2}, \frac{1}{2})$, the GFF double trace blocks $\hat{p}_n := \mathcal{I}_z \, g_{2\tilde{\alpha}+2n}$ form a Riesz basis for $L^2$.**

As mentioned before, this is natural because we know that, for the value $\tilde{\alpha} = 0$, the $\hat{p}_n$ are related to an orthonormal basis of Jacobi polynomials (see (3.14)). Therefore, we expect that, for close enough values of $\tilde{\alpha}$, the basis property is preserved (although not orthonormality).

Proving this mathematical statement is rather technical, and is relegated to Appendix B.1. Here will only give a sketch of the argument. The task is to construct a bounded, invertible operator

$$T : L^2 \to L^2$$
$$e_n \mapsto \hat{p}_n \ , \tag{3.23}$$

where $e_n$ is some orthonormal basis (there is no need to specify which, since any is equivalent). In this case, we already explicitly know of a dual family of $L^2$ functions, $\hat{h}_n := i^{(\tilde{\alpha})}(\hat{f}_n)$, obtained by embedding the GFF functional kernels (A.7) into $L^2$, which satisfy

$$\langle \hat{h}_n | \hat{p}_m \rangle = \delta_{mn} \ . \tag{3.24}$$

This allows us to construct an inverse map by defining an operator

$$S : L^2 \to L^2$$
$$e_n \mapsto \hat{h}_n \ . \tag{3.25}$$

Assuming that $T$ and $S$ extend to well-defined, bounded operators, (3.24) gives

$$S^\dagger T = 1 \ . \tag{3.26}$$

This is almost enough to conclude that $S^\dagger = T^{-1}$, but formally for a Hilbert space it only implies that $T$ is left-invertible. However, the right-invertibility of $T$ (or the left-invertibility of $S^*$) may be argued using the fact that the $\hat{f}_n$ are polynomials, so that $\hat{f}_n^{(\tilde{\alpha})}$ can be written as **finite** combinations of some known orthonormal basis (see Appendix B.1).

The only remaining step is to show that (3.23) and (3.25) give well-defined, bounded linear operators $T$ and $S$. This is shown in Appendix B.1 by computing and estimating the matrix elements $\langle \hat{p}_n | \hat{p}_m \rangle$ and $\langle \hat{h}_n | \hat{h}_m \rangle$ for large $n, m$. We then apply **Schur's test** — a classic mathematical tool giving a sufficient condition for such operators to be bounded.

**(b) Analyticity and bounds on functionals.**

For the GFF case, we already know the dual Riesz basis $\hat{h}_n$: by construction, they must match the GFF functionals (A.7) from our previous work: $\hat{h}_n = i^{(\tilde{\alpha})}(\hat{f}_n)$. Hence property

(b) is immediate.

## (c) Decomposition of the Cauchy kernel.

The decomposition (3.20) was already derived in the GFF case in eq. (2.32). As we explained above, it can also be derived as an $L^2$ decomposition with respect to the Riesz basis $\hat{h}_n$. The estimate (2.33) then applies, implying that this decomposition converges as an analytic function of $z$ and a hyperfunction of $w$.

### 3.2.2 Interacting functionals using $L^2$

Now we are ready to do the same thing for interacting spectra, $\Delta_n = 2\tilde{\alpha} + 2n + \gamma_n$, i.e. prove that the properties (a)–(c) are satisfied.

**(a) For any $\tilde{\alpha} \in (-\frac{1}{2}, \frac{1}{2})$, the interacting set of blocks $p_n := \mathcal{I}_z \, g_{2\tilde{\alpha}+2n+\gamma_n}$ form a Riesz basis for $L^2_\mu$.**

Our strategy to establish this is to study the mapping between the GFF blocks and the interacting ones:

$$M : \quad \hat{p}_n \mapsto p_n \ . \tag{3.27}$$

In particular, if we can show that $M$ defines a bounded, invertible operator, then so is the composition $MT$ which now relates $p_n$ to an orthonormal basis (where $T$ is the map in (3.23)).

Our approach is first to show that $(1 - M)$ is a *compact* operator so that, in particular, $M$ is bounded. Essentially this follows from the fact that the $\gamma_n$ decay for large $n$ so that the inner products of $p_n$ are sufficiently close to those of $\hat{p}_n$. Then, using the fact that the $p_n$ are distinct eigenfunctions of a Casimir operator, this implies that $M$ is injective, and since $(1 - M)$ is compact this implies it is also invertible as desired. The detailed argument may be found in Appendix B.2.

## (b) Analyticity and bounds on functionals.

Unlike the GFF case, we do not have a stand-alone definition of the interacting functionals kernels $h_n$: they are simply defined as the Riesz basis to dual $p_n$. However, they may be expanded as an $L^2$-convergent sum of the kernels $\hat{h}_m$ (the latter being a Riesz basis):

$$h_n = \sum_m \langle h_n | \hat{p}_m \rangle \, \hat{h}_m \ . \tag{3.28}$$

We would like to prove that $h_n = i^{(\tilde{\alpha})}(f_n)$ for some $f_n \in \mathcal{A}$, i.e. that $h_n$ are real-analytic

and suitably bounded at $\infty$ and 1. This is carefully argued in Appendix C, making crucial use of the assumptions (3.2) of polynomial boundedness and analyticity of the anomalous dimensions. The argument consists of three steps:

1. The expansion coefficients above can be written in terms of the putative interacting functional actions:

$$b_m := \langle h_n | \hat{p}_m \rangle = (f_n, g_{\hat{\Delta}_m}^{(\tilde{\alpha})}) = \theta_n(\hat{\Delta}_m) \tag{3.29}$$

   We show that these coefficients $b_m$ admit a suitable analytic continuation into the complex $m$ plane.

2. Next, using this analytic continuation, we prove that the interacting functional kernels given by sums $\sum_m b_m \hat{h}_m$, are real-analytic functions. We do this via a Sommerfeld-Watson type trick.

3. Finally we argue that the decaying anomalous dimensions lead to the right boundedness properties at $z = 1$ and $\infty$. Roughly this is because, at large $\Delta$, interacting and GFF functionals behave similarly, and that limit is controlled by the $z = 1$ and $z = \infty$ regions (however, the interacting functionals are non-analytic at the endpoints 1 and $\infty$, and may have a branch cut along $(-\infty, 1)$).

Let us elaborate on the first step. We will be able to construct this analytic continuation explicitly thanks to the explicit formula derived in Section 4 which expresses the interacting functional actions $\theta_n(\Delta)$ in terms of the sparse spectrum $\{\Delta_n\}$. Essentially the reason why this is possible is analyticity of $\theta_n(\hat{\Delta}_m)$ only has to hold for sufficiently large $m$. In this regime we have $\theta_n(\hat{\Delta}_m) \sim -\gamma_m \hat{\theta}_n'(\hat{\Delta}_m)$, and the result now follows from our explicit knowledge of the $\hat{\theta}_n$ and the assumptions made on the analytic properties of the $\gamma_m$, cf. (3.2) and (3.3).

### (c) Decomposition of the Cauchy kernel.

The decomposition (3.20) can again be argued to converge as an analytic function of $z$ and a hyperfunction of $w$, using the same estimate (2.33) as in the GFF case. Since the anomalous dimensions are assumed to decay and $g_\Delta(w)$ has exponential dependence $r(w)^\Delta$ we have

$$g_{\Delta_n}(w) \underset{n \to \infty}{\sim} g_{\hat{\Delta}_n}(w) \left(1 + O(n^{-\epsilon})\right) . \tag{3.30}$$

Similarly, one can show

$$f_n(z) = \hat{f}_n(z) \left(1 + O(n^{-\epsilon})\right) . \tag{3.31}$$

Essentially this is because for large $n$ we have $p_n \sim \hat{p}_n$ and hence $h_n \sim \hat{h}_n$. (More rigorously, this follows from eq. (5.12), or by substituting (C.38) into (C.40)). Hence the estimate

(2.33) carries over to the interacting case, thereby guaranteeing the absolute convergence of (3.20) for $|r(w)| < |r(z)| < 1$, as required.

## 3.3 On general values of $\tilde{\alpha}$ and 'subtractions'

The criteria (a)–(c) above focus on the particular case $\tilde{\alpha} \in (-\frac{1}{2}, \frac{1}{2})$. However, the special role of $\tilde{\alpha} = \frac{1}{2}$ is simply an artefact of our use of $L^2$, which is, after all, only a convenient mathematical trick. In terms of the physically meaningful spaces $\mathcal{A}$ and $\mathcal{HF}$, all of the conclusions above still hold for general values of $\tilde{\alpha}$. Moreover, they can still be derived using $L^2$, but through a slightly more intricate relation interpreted as a 'subtraction' procedure.

In the following, we shall generalise the argument above, underlining the necessary assumptions as we use them. Casual readers may wish to skip this section. On the other hand, interested readers should note that there is a remaining boundary case $\tilde{\alpha} \in \frac{1}{2} + \mathbb{Z}$, which is treated in Appendix D.3.

For $\tilde{\alpha} > \frac{1}{2}$, clearly the blocks $\mathcal{I}_z\, g_{\Delta_n}$ with $\Delta_n \sim 2\tilde{\alpha} + 2n$ cannot be a Riesz basis for $L^2$, since we know that (a) they are a proper subset of a Riesz basis (by adding a few additional blocks):

$$\{p_n = \mathcal{I}_z\, g_{\Delta_n^\beta}\}\,, \qquad \Delta_n^\beta := \begin{cases} \Delta_n^{\mathrm{add}}\,, & 1 \le n \le M\,, \\ \Delta_{n-M}^{\tilde{\alpha}}\,, & n > M\,, \end{cases} \tag{3.32}$$

where the additional scaling dimensions $\Delta_1^{\mathrm{add}}, \dots, \Delta_n^{\mathrm{add}}$ can be chosen arbitrarily. This new spectrum then has the large $n$ behaviour $\Delta_n^\beta \sim 2\beta + 2n$, where

$$\beta = \tilde{\alpha} - M \in (-\tfrac{1}{2}, \tfrac{1}{2})\,, \qquad M \in \mathbb{N}\,. \tag{3.33}$$

Applying the argument of Section 3.1.3 with this Riesz basis, we reach the following equation before encountering any issues:

$$i_z^{(\tilde{\alpha})}\Big(\frac{1}{z-w}\Big) = \sum_n g_{\Delta_n^\beta}^{(\tilde{\alpha})}(w)\, h_n(z)\,. \tag{3.34}$$

In the $\tilde{\alpha} < \frac{1}{2}$ case, we had assumed $h_n = i^{(\tilde{\alpha})}(f_n)$, but for $\tilde{\alpha} > \frac{1}{2}$ that assumption will not be true. Instead, (b) certain 'subtracted' combinations of $h_n$ will be in the image of $i^{(\tilde{\alpha})}$:

$$h_{n+M} - \sum_{k=1}^{M} c_{n+M,k}^{\tilde{\alpha}}\, h_k = i^{(\tilde{\alpha})}(f_n)\,, \qquad f_n \in \mathcal{A}\,, \tag{3.35}$$

at the price of 'losing' $M$ functionals $h_1, \dots, h_M$. Due to the duality of the spaces $\mathcal{A}$ and $\mathcal{HF}$, the loss of these functionals is reflected in the fact that the hyperfunctions $g_{\Delta_n^\beta}^{(\tilde{\alpha})}(w)$ on the RHS of (3.34) are not independent when viewed as elements of $\mathcal{HF}$. Intuitively, this is

because they are 'oversubtracted' by $\tilde{\alpha}$, which is $M$ units greater than $\beta$. As a result, there are $M$ relations between them (see Appendix D.1) involving precisely the same coefficients,

$$g_{\Delta_k^\beta}^{(\tilde{\alpha})}(w) = -\sum_{n>M} c_{nk}^{\tilde{\alpha}} \, g_{\Delta_n^\beta}^{(\tilde{\alpha})}(w) \,, \qquad k = 1, \ldots, M \,. \tag{3.36}$$

Using these relations to eliminate the lowest $M$ blocks, the remaining expression only contains the original blocks (not the added ones) and the subtracted combinations of functionals (3.35):

$$i_z^{(\tilde{\alpha})}\left(\frac{1}{z-w}\right) = \sum_n g_{\Delta_n}^{(\tilde{\alpha})}(w) \, i^{(\tilde{\alpha})}(f_n)(z) \,. \tag{3.37}$$

Now we would like to apply $(i^{(\tilde{\alpha})})^{-1}$ and pull it through the sum we obtain the same result (3.20) as above:

$$\frac{1}{z-w} = \sum_{n=1}^\infty g_{\Delta_n}^{(\tilde{\alpha})}(w) \, f_n(z) \,. \tag{3.38}$$

We now need to assume that (c) this decomposition converges as an analytic function of $z$ and a hyperfunction of $w$. As in the unsubtraced case, this implies that the functionals $f_n$ solve our task and form a complete Schauder basis for $\mathcal{A}$.

Let us comment that the arbitrary choice of the $M$ additional scaling dimensions $\Delta_n^{\text{add}}$ drops out in the eventual functionals $f_n \in \mathcal{A}$. This must be the case, since $f_n$ are dual to the Schauder basis $g_{\Delta_n}^{(\tilde{\alpha})}$, so the difference between functionals generated by making different choices is orthogonal to all basis elements, and hence vanishes.

Gathering up the assumptions, a set of sufficient conditions for a complete basis of analytic functionals for general values of $\tilde{\alpha}$ is:

---

**Criteria in $L^2$ language:**     **(general $\tilde{\alpha}$)**

(a) A set of discontinuities of blocks $\{p_n = \mathcal{I}_z \, g_{\Delta_n}\}$ can be completed to a Riesz basis for $L^2$ by adding a finite number $M$ of blocks.

(b) Finite combinations of $M$ of the dual basis Riesz basis $\{h_n\}$ take the form $i^{(\tilde{\alpha})}(f_n)$ for some $f_n \in \mathcal{A}$.

(c) The decomposition (3.38) converges as an analytic function of $z$ and a hyperfunction of $w$.

---

In Appendix D.1, we show that any 'sparse' spectrum $\Delta_n = 2\tilde{\alpha} + 2n + \gamma_n$ (with $\gamma_n$ satisfying the assumptions (3.2)) satisfies these conditions — generalising the arguments from the $\tilde{\alpha} < \frac{1}{2}$ case above in Section 3.2.

## 3.4 Completeness and extremality

### 3.4.1 Completeness

The results of the previous sections show that certain spectra $\Delta_n = 2\tilde{\alpha} + 2n + \gamma_n$ possess complete bases of functionals dual to them. In this section we will discuss in more detail the meaning of completeness and what exactly can be learned from the sum rules associated to these functionals.

We have established the following statement:

---

**Locality of form factors:**

Suppose $\sum_\Delta |c_\Delta||g_\Delta(z)|$ satisfies the assumed polynomial bound (2.5) for some $\alpha_F$. Then:

$$F(z) = \sum_\Delta c_\Delta\, g_\Delta(z) \quad \text{local} \qquad \Longleftrightarrow \qquad \sum_\Delta c_\Delta\, \theta_n(\Delta) = 0 \quad \text{for } n \geq 1 \ ,$$

where the basis of functionals on the RHS can have any value $\tilde{\alpha} > \alpha_F - 1$.

---

It may seem puzzling that the choice of $\tilde{\alpha}$ above is almost arbitrary. For instance, when the choice of basis has $\gamma_n = 0$, then increasing $\tilde{\alpha}$ by one unit seems to remove a constraint, as the resulting functionals are now dual to a proper subset of the basis elements. How can this be?

The answer lies in the assumptions of polynomial boundedness. While it is true that, as we increase $\tilde{\alpha}$, we get a less restrictive set of sum rules, this just reflects the fact that the sum rules progressively allow for local solutions with harder and harder polynomial behaviour. Since the correct behaviour is being imposed as an external assumption on the allowed $c_\Delta$, then there is no contradiction.

Relaxing this *a priori* requirement on the $c_\Delta$ can be at least partially achieved by working with the largest possible set of functionals. To see this, note that the polynomial boundedness of the form factor $F$ roughly correlates with the growth of the $c_\Delta$ for large $\Delta$. Since $\theta_n(\Delta) \sim \Delta^{-2\tilde{\alpha}+\frac{d}{2}-3}$ (cf. Appendix A), we see that lower values of $\tilde{\alpha}$ automatically rule out unwanted kinds of polynomial growth of $F$. To avoid as many extraneous solutions as possible from the start, one ought to choose $\tilde{\alpha}$ not too far from $\alpha_F$. In practice, a natural sweet spot is to choose $\tilde{\alpha} = \alpha_F$. The reason is that, if we believe some bulk operator to have a local form factor with polynomial behaviour given by $\alpha_F$, then there are certainly local form factors of the same operator acted on by the bulk AdS Laplacian, whose growth will now be $\alpha_F + 1$. To pick up the first but not the second in our solutions to locality, we should choose $\tilde{\alpha} > \alpha_F - 1$ and $\tilde{\alpha} \leq \alpha_F$. To speed up convergence of the sum rules, the maximum value $\tilde{\alpha} = \alpha_F$ is a good choice.

**Hyperfunctions vs. tempered distributions.** This perspective of choosing the largest possible set of functionals suggests the following question. Given that we know discontinuities of form factors actually live in the smaller space of tempered distributions[18] rather than hyperfunctions, we could have chosen this space from the start and worked with functionals in its dual space of Schwartz test functions, which is larger than $\mathcal{A}$. This means that there exist additional sum rules, distinct from ours, that could have been written down. What do such sum rules buy us? Clearly they do not change anything regarding locality: the space $\mathcal{A}$ is dense in the Schwartz space, so our bases of functionals are still complete in that larger space. The new sum rules would simply be more wild combinations of our functionals, converging only in a generalised sense, but not in $\mathcal{A}$.

Instead, such exotic sum rules should be again understood as replacing some *a priori* requirements on the $c_\Delta$. In particular, they would constrain the $c_\Delta$ such that $\sum_\Delta |c_\Delta||g_\Delta|$ is polynomially bounded — and otherwise the new sum rules would not converge. Note that this is **not** the case for our own sum rules: strictly speaking, our sum rules require only polynomial boundedness of $F$ near the points $z = 1$ and $z = \infty$. This is evident from the fact that the functional actions are defined by contour integrals touching the cut $z \in (1, \infty)$ only at these endpoints. A sufficient but not necessary condition for achieving this is indeed to request polynomial boundedness of $\sum_\Delta |c_\Delta||g_\Delta|$ as we have in (2.5). But this can be relaxed, e.g., to

$$\left| \sum_{\Delta > \Delta^*} c_\Delta \, g_\Delta(z) \right| \lesssim \left\{ \begin{array}{ll} |z|^{\alpha_F} & \text{as } z \to \infty \\ |z - 1|^{-\alpha_F} & \text{as } z \to 1 \end{array} \right. \qquad \text{for all } \Delta^* \, . \qquad (3.39)$$

Thus our sum rules are actually compatible with a wilder set of $c_\Delta$, e.g. those which *a priori* could have led to form factors with essential singularities on the cut $z \in (1, \infty)$. The extra functionals gained by working in the space of Schwartz functions would rule out such esoteric possibilities. By working with hyperfunctions with their duals, we instead choose to rule these out by directly constraining the allowed $c_\Delta$.

### 3.4.2 Extremal solutions

Moving on, let us now note that our locality sum rules have particularly simple solutions which we call '**extremal**'. For each complete basis of functionals, such solutions can be labelled by a choice of fixed BOE data which acts as inputs. These are then completed by the sparse set $\{g_{\Delta_n}\}$ of blocks dual to our functionals. In the simplest case of a single input block $g_\Delta$, we write the ansatz for a tentative local form factor.

$$\mathcal{L}_\Delta^{\{\Delta_n\}} = g_\Delta + \sum_{n=1}^{\infty} c_n \, g_{\Delta_n} \, . \qquad (3.40)$$

---

[18]This follows from the analysis of [23] for the crossing equation, since our assumed polynomial bound implies the 'slow-growth' condition there.

Applying the functionals $\theta_n$ to the above construction, the sum rules are 'diagonalised' by duality, leading to an immediate solution for the coefficients:

$$c_n = -\theta_n(\Delta) \ . \tag{3.41}$$

The resulting extremal solutions are thus interacting generalisations of the 'local block' solutions of our previous work [15], to which they reduce for the special case $\Delta_n = \hat{\Delta}_n$. In Section 4 below we will give an explicit expression for $\theta_n(\Delta)$ in terms of $\{\Delta_n\}$ — see equation (4.19).

We note that the above local blocks should correspond to an alternate formulation of the locality constraint of the form

$$F \text{ is local} \qquad \Leftrightarrow \qquad \sum_\Delta c_\Delta \left[ g_\Delta - \mathcal{L}_\Delta^{\{\Delta_n\}} \right] = 0 \tag{3.42}$$

That is, for local form factors the usual conformal block expansion may be swapped by a manifestly local one. Again, this was established in [15] for the special GFF case, but we expect it to be generally valid for any set $\{\Delta_n\}$ satisfying our assumptions. To prove it we would have to show that the 'master functional' [32] for these bases is a valid functional.

# 4   The space of functional actions

Until now, we have been concerned with the space $\mathcal{A}$ of functional kernels, and have constructed bases for it. This section proposes to consider instead the space of *functional actions*,

$$\theta(\Delta) = \langle h_\theta \, | \, \mathcal{I}_z g_\Delta \rangle = \int_1^\infty \mathrm{d}z \, \mu(z) \, h_\theta(z) \, \mathcal{I}_z g_\Delta(z) \ . \tag{4.1}$$

These are the functions of the complex variable $\Delta$ defined by integrating functional kernels $h_\theta$ against the blocks $g_\Delta$. If one can understand the space of such functional actions $\theta(\Delta)$, then one understands all sum rules, $\sum_\Delta c_\Delta \theta(\Delta) = 0$, characterising the locality constraints.

In this section it will be convenient to define:

$$P_{\lambda_\Delta}(z) := \mathcal{I}_z \, g_\Delta(z) = \frac{\Gamma(\frac{\Delta}{2})}{\Gamma(\frac{\Delta+2-d}{2})} \left(\tfrac{z-1}{z}\right)^{1-\frac{d}{2}} J_{\frac{\Delta-2}{2}}^{(1-\frac{d}{2},0)}\left(\tfrac{2-z}{z}\right) \ . \tag{4.2}$$

This is indeed a function of the Casimir eigenvalue $\lambda_\Delta = \Delta(\Delta - d)$, since it is invariant under $\Delta \leftrightarrow d - \Delta$. In the remainder of this section we will abuse notation and equate $\theta(\Delta) \equiv \theta(\lambda_\Delta)$.

Next, we introduce an operator that formalises the mapping between kernels and

functional actions. We set:

$$\theta := K[h_\theta] \; , \qquad K[h](\lambda) := \int_1^\infty \mathrm{d}z\, \mu(z)\, h(z)\, P_\lambda(z) \; . \tag{4.3}$$

The image of this transform is typically called a *Paley-Wiener* space:

$$\mathcal{PW}^{(\tilde{\alpha})} := K[i^{(\tilde{\alpha})}(\mathcal{A})] \quad \subset \quad \mathcal{PW} := K[L^2] \; . \tag{4.4}$$

We recall that that $i^{(\tilde{\alpha})}(\mathcal{A})$ is the subspace of $L^2$-functions that are real-analytic and sufficiently bounded at the endpoints — so as to have finite action on hyperfunctions and produce valid sum rules. Its image $\mathcal{PW}^{(\tilde{\alpha})}$ is therefore the space of actual functional actions. But, as above in Section 3, it will be useful to consider the larger space $\mathcal{PW}$.

The operator $K : L^2 \to \mathcal{PW}$ is a bijection: i.e. to each functional action $\theta(\lambda) \in \mathcal{PW}$, there corresponds precisely one functional kernel $h_\theta \in L^2$. Surjectivity follows by the definition of $\mathcal{PW}$. For injectivity, we note that

$$K[h](\lambda) := \langle h \,|\, P_\lambda \rangle = 0 \quad \forall \lambda \quad \implies \quad \langle h \,|\, P_{\lambda_n} \rangle = 0 \quad \forall n \geq 1 \quad \implies \quad h = 0 \; , \tag{4.5}$$

for any set $P_{\lambda_n}$ forming a Riesz basis of $L^2$. (We have shown that there are many.) The Hilbert space structure on $L^2$ then defines a Hilbert space structure on $\mathcal{PW}$ with inner product

$$\langle \theta | \theta' \rangle := \langle h_\theta | h_{\theta'} \rangle \; , \tag{4.6}$$

so that $K$ is a Hilbert space isomorphism. With this definition, since our analytic functionals $i^{(\tilde{\alpha})}(\mathcal{A})$ are dense in $L^2$, then their image $\mathcal{PW}^{(\tilde{\alpha})}$ is clearly a dense subspace of $\mathcal{PW}$.

## 4.1 An intrinsic definition

We would like to give an intrinsic definition of the space $\mathcal{PW}$, i.e. one that does not rely on the mapping $K$. To see how this may be achieved, let us examine more closely the properties of the functions $\theta(\lambda)$. First, we note that $\theta(\lambda)$ are entire functions (analytic functions with no singularities in $\mathbb{C}$): this follows since the integral in (4.3), which is controlled by $L^2$, converges locally uniformly over $\lambda \in \mathbb{C}$. Moreover, it follows from the asymptotic form (given in detail in (A.16))

$$P_{\lambda_\Delta}(z) \underset{|\Delta| \gg 1}{\sim} \sin\left[\frac{\Delta}{2} \log(r(z))\right] \times (\ldots) \tag{4.7}$$

that $\theta(\lambda)$ is entire of *order $1/2$ and type $\pi/2$*, i.e. it is bounded in all directions of the complex plane as

$$|\theta(\lambda)| \leq C \exp\left(\frac{\pi}{2}\sqrt{|\lambda|}\right) \qquad \text{for all} \quad \lambda \in \mathbb{C} . \tag{4.8}$$

For positive real $\lambda$, the functional actions are actually much more constrained. To see this, first let us set

$$\check{\lambda}_n := \hat{\lambda}_n\big|_{\tilde{\alpha}=0} = 2n(2n-d) . \tag{4.9}$$

It is easily checked from the relation of $P_{\check{\lambda}_n}$ to Jacobi polynomials that

$$\langle P_{\check{\lambda}_n}|P_{\check{\lambda}_m}\rangle = \delta_{n,m}||P_{\check{\lambda}_n}||^2 , \qquad ||P_{\check{\lambda}_n}||^2 = \frac{1}{\sqrt{\frac{d^2}{4} + \check{\lambda}_n}\left(\frac{\check{\Delta}_n}{2}\right)^2_{1-\frac{d}{2}}} . \tag{4.10}$$

Finiteness of the $L^2$ norm of $h_\theta$ now implies

$$\sum_{n=1}^{\infty} \frac{|\theta(\check{\lambda}_n)|^2}{||P_{\check{\lambda}_n}||^2} < \infty , \qquad , \tag{4.11}$$

which roughly translates as

$$|\theta(\lambda)| = O(\lambda^{\frac{d}{4}-1-\epsilon}) \qquad \text{for } \lambda > 0 \qquad (\text{where } \epsilon > 0) . \tag{4.12}$$

We are ready to write down a tentative definition of the space $\mathcal{PW}$: it is the space of entire functions satisfying conditions (4.8),(4.11) equipped with the inner product

$$\langle \theta|\theta'\rangle := \sum_{n=1}^{\infty} \frac{\theta(\check{\lambda}_n)\theta'(\check{\lambda}_n)}{||P_{\check{\lambda}_n}||^2} \tag{4.13}$$

To prove that this is indeed our original space we must show that to any element $\theta$ there corresponds an $L^2$ function $h_\theta$ such that $\theta = K[h_\theta]$. Starting from $\theta(\lambda)$ let us then set:

$$h_\theta(z) = \sum_{n=1}^{\infty} \theta(\check{\lambda}_n)\frac{P_{\check{\lambda}_n}(z)}{||P_{\check{\lambda}_n}||^2} . \tag{4.14}$$

This sum converges in $L^2$ thanks to condition (4.11). Furthermore, with this definition, we have $\langle \theta|\theta'\rangle = \langle h_\theta|h_{\theta'}\rangle$ as required. It remains to show that the prescription (4.14) satisfies

$K[h_\theta](\lambda) = \theta(\lambda)$. To see this, we first compute:

$$K[h_\theta](\lambda) = \sum_{n=1}^{\infty} \theta(\check{\lambda}_n) K_{\check{\lambda}_n}(\lambda) \tag{4.15}$$

with

$$K_{\check{\lambda}_n}(\lambda) := \int_1^\infty \mathrm{d}z\, \mu(z) \frac{P_{\check{\lambda}_n}(z) P_\lambda(z)}{||P_{\check{\lambda}_n}||^2} = \frac{4}{\pi} (-1)^n \frac{\sin[\frac{\pi}{2}\Delta]}{\lambda - \check{\lambda}_n} \frac{\left(\frac{\check{\Delta}_n}{2}\right)_{1-\frac{d}{2}}}{\left(\frac{\Delta}{2}\right)_{1-\frac{d}{2}}} . \tag{4.16}$$

To see that this indeed is the same as $\theta(\lambda)$, consider the identity

$$\theta(\lambda) = \oint_\lambda \frac{\mathrm{d}\lambda'}{2\pi i} \frac{\theta(\lambda')}{\lambda' - \lambda} \frac{\sin[\frac{\pi}{2}\Delta]}{\sin[\frac{\pi}{2}\Delta']} \frac{\left(\frac{\Delta'}{2}\right)_{1-\frac{d}{2}}}{\left(\frac{\Delta}{2}\right)_{1-\frac{d}{2}}} . \tag{4.17}$$

Note that the factors inside the integrand should be really thought of as functions of $\lambda, \lambda'$ by setting e.g. $\Delta = \frac{d}{2} + \sqrt{\left(\frac{d}{2}\right)^2 + \lambda}$. In particular, it can be checked that they respect invariance under $\Delta \leftrightarrow d - \Delta$. Deforming the contour, we can drop the arcs thanks to assumptions (4.8) and (4.11), leading to a sum over poles, which precisely reproduces the expression (4.15). Thus we indeed conclude $\theta = K[h_\theta]$.

This completes the construction of the space $\mathcal{PW}$. Since $\mathcal{PW}$ is isomorphic to $L^2$, Riesz bases of the latter lift into Riesz bases of the former. In particular, the Riesz bases $h_n$ that we constructed for $L^2$ (dual to a sparse set of blocks $p_{\Delta_n}$) are mapped to Riesz bases $\theta_n := K[h_n]$ for $\mathcal{PW}$, and any functional action can be decomposed into such a basis:

$$\theta(\lambda) = \sum_{n=1}^{\infty} \theta(\lambda_n)\, \theta_n(\lambda) . \tag{4.18}$$

## 4.2 Exact interacting functional actions: a product formula

The interacting functional actions live in the $\mathcal{PW}^{(\tilde{\alpha})}$ space. The classic factorisation theorem of Hadamard tells us that entire functions of order less than 1 are entirely determined by their zeros. This is in fact valid for any $\theta \in \mathcal{PW}$, and in particular the subspace $\mathcal{PW}^{(\tilde{\alpha})}$. Consider then an interacting functional basis element. Since we know the positions of its zeros, then the following remarkable result follows almost directly:

Consider a basis $\{\theta_m\}$ dual to $\{\Delta_n\}$, i.e. satisfying $\theta_m(\lambda_n) = \delta_{mn}$. They are given by

$$\theta_n(\lambda) = \prod_{\substack{m=1 \\ m \neq n}}^{\infty} \left( \frac{\lambda - \lambda_m}{\lambda_n - \lambda_m} \right) . \tag{4.19}$$

Before we continue, there is a small argument necessary to actually establish this formula, since we must prove that the set $\lambda_n$ exhausts *all* of the zeros of the functional actions. To show this, let us first check that the formula above holds for the GFF bases. Starting from expression (A.12), we can rewrite it as (with now $\hat{\lambda}_n = 2\tilde{\alpha} + 2n$):

$$\hat{\theta}_n(\lambda_\Delta) = \lim_{\Delta' \to \Delta_n} \left( \frac{\lambda_{\Delta'} - \hat{\lambda}_n}{\lambda_\Delta - \hat{\lambda}_n} \right) \frac{\Gamma(\frac{2+2\tilde{\alpha}-\Delta'}{2})\Gamma(\frac{2+2\tilde{\alpha}+d-\Delta'}{2})}{\Gamma(\frac{2+2\tilde{\alpha}-\Delta}{2})\Gamma(\frac{2+2\tilde{\alpha}+d-\Delta}{2})} \tag{4.20}$$

and (4.19) now follows from the identity $\Gamma(1 + z) = \prod_{m=1}^{\infty} \frac{(1+\frac{1}{n})^z}{1+\frac{z}{n}}$. We can now show for the interacting bases that there are still no other zeros. Firstly, for any basis dual to a spectrum with $\gamma_n = \Delta_n - \hat{\Delta}_n$ decaying, the large-$\lambda$ behaviour of $\theta_n(\lambda)$ and $\hat{\theta}_n(\lambda)$ must be the same, as argued in eq. (5.5) below. Let us then write $\theta_n(\lambda)$ as a polynomial factor $R_N(\lambda)$ capturing the possible extra zeros times the infinite product in (4.19). Since $\lambda_n \sim \hat{\lambda}_n$ for large $n$, we can write

$$\frac{\theta_n(\lambda)}{\hat{\theta}_n(\lambda)} \sim \frac{R_n^N(\lambda)}{\hat{\theta}_n(\lambda_n)} \prod_{m=1}^{N-1} \left[ \left( \frac{\lambda - \lambda_m}{\lambda - \hat{\lambda}_m} \right) \left( \frac{\lambda_n - \hat{\lambda}_m}{\lambda_n - \lambda_m} \right) \right] \tag{4.21}$$

Requiring the same large $\lambda$ behaviour now fixes $R_n^N(\lambda) \sim 1$.

Incidentally, this expression is also a useful way to approximate the infinite product in practice. One finds to subleading order in $N$:

$$R_n^N(\lambda) \sim 1 + (\lambda - \hat{\lambda}_n) \sum_{m=N}^{\infty} \frac{\lambda_m - \hat{\lambda}_m}{(\lambda - \hat{\lambda}_m)(\hat{\lambda}_n - \hat{\lambda}_m)} . \tag{4.22}$$

This makes clear that for fixed $\lambda$, $\hat{\lambda}_n$ and $\gamma_n = O(n^{-\epsilon})$, the correction term to $R_N \sim 1$ is $O(N^{-2-\epsilon})$. In applications, we know $\gamma_n$ as an analytic function of $n$ and it is possible to do better. For instance, with $\gamma_n \sim g/n^2$ we get

$$\frac{R_n^N(\lambda) - 1}{g} \sim \frac{2 \log\left(1 - \frac{\hat{\lambda}_n}{\hat{\lambda}_N}\right)}{\hat{\lambda}_n} - \frac{2 \log\left(1 - \frac{\lambda}{\hat{\lambda}_N}\right)}{\lambda} \tag{4.23}$$

More generally, one may use the exact formula (C.15) derived in Appendix C.1, which expresses $R_n^N$ essentially as the exponential of an integral.

We conclude this section with two comments.

Firstly, the explicit formula (4.19) means, as promised in Section 3.4, that we can now write down the interacting, extremal solutions (3.40) corresponding to any particular interacting basis in closed form:

$$\mathcal{L}_\Delta^{\{\Delta_n\}}(z) = g_\Delta(z) - \sum_{n=1}^{\infty} \Big[ \prod_{\substack{m=1 \\ m \neq n}}^{\infty} \left( \frac{\lambda_\Delta - \lambda_m}{\lambda_n - \lambda_m} \right) \Big] g_{\Delta_n}(z) \ . \tag{4.24}$$

Moreover, any linear combination of such solutions (possibly with different choices of basis) is also a solution to locality.

Secondly, as we mentioned in Section 3.2.2 (item (b)), having this explicit expression for the interacting functional actions $\theta_n(\Delta)$ crucially allows us to study the sequence $\theta_n(\hat{\Delta}_m)$ appearing in the decomposition $f_n = \sum_m \theta_n(\hat{\Delta}_m) \hat{f}_m$. In particular, in Appendix C.1, we use it to establish that this sequence has appropriate analyticity properties in $m$, which are then used to show that $f_n \in \mathcal{A}$.

# 5 Application: a case study in extremality

In this section we will apply our formalism to a concrete example. Specifically, we will construct a functional basis, and associated extremal solutions, which diagonalise the bulk locality constraints for a certain interacting CFT spectrum. We will do this in multiple ways, as a consistency check on our various results.

Before doing so, it is useful to have a general idea of what interacting functionals and their actions look like, which we turn to next.

## 5.1 Preamble: general expectations

We have established the existence of large families of interacting functionals, in terms of their kernels $f_n(z)$ and their actions $\theta_n(\Delta)$ on blocks. We would like to understand how the large $\Delta$ behaviour of $\theta_n(\Delta)$ correlates with properties of the functional kernel. The functionals act on blocks as

$$\theta_n(\Delta) = (f_n, g_\Delta^{(\tilde{\alpha})}) = \int_1^{\infty} \frac{\mathrm{d}z}{\pi z^{1+\tilde{\alpha}}} f_n(z) \mathcal{I}_z g_\Delta(z) \ . \tag{5.1}$$

For large $\Delta$, the integral is dominated by contributions around $z = 1$ and $\infty$. In the neighbourhood of these points we can deform the contour to write

$$\theta_n(\Delta) = \int_{-\infty} \frac{dz}{\pi} \mathcal{I}_z \left[ f_n(z) e^{i\pi \frac{\Delta - 2\tilde{\alpha}}{2}} \right] (-z)^{-1-\tilde{\alpha}} \tilde{g}_\Delta(z) - \int^1 \frac{dz}{\pi} \mathcal{I}_z[z^{-1-\tilde{\alpha}} f_n(z)] g_\Delta + \dots \; ,$$

(5.2)

where the ellipsis represents contributions away from those points and $\tilde{g}_\Delta$ is defined in equation (A.5). Note that, in the above, the contour has been deformed so that the integrals shown run along the real axis along $z < 0$ and $z < 1$ respectively in the neighbourhoods of $\infty$ and 1. Let us now assume

$$f_n(z) \underset{z \to 1}{\sim} d_1 (z - 1)^{\tilde{\alpha} + \frac{\eta_1}{2}} + \text{analytic} \; ,$$

$$f_n(z) \underset{z \to \infty}{\sim} -\frac{1}{z} + \frac{d_\infty}{z^{1 + \frac{\eta_\infty}{2}}} \; ,$$

(5.3)

where we need $\eta_{1,\infty} \geq 0$ in order that $f_n$ belong to the space $\mathcal{A}$. Using the asymptotic expressions given in appendix (A.17) we can perform the integrals and obtain an expression for the functional action at large $\Delta$. Defining the constants,

$$r_\infty := \frac{2}{\pi} (1 + \tilde{\alpha})^2_{\frac{\eta_\infty}{2}} \sin[\tfrac{\pi}{2} \eta_\infty] \; ,$$

$$r_1 := \frac{2}{\pi} \sin[\tfrac{\pi}{2}(2\tilde{\alpha} + \eta_1)](1 + \tilde{\alpha})_{\frac{\eta_1}{2}} (1 + \tilde{\alpha})_{\frac{\eta_1 + 2 - d}{2}} \; ,$$

(5.4)

then we obtain:

$$\theta_n(\Delta) \underset{\Delta \to \infty}{\sim} \frac{\Gamma(1 + \tilde{\alpha})^2}{\pi^2 (\frac{\Delta}{2})^{2\tilde{\alpha} - \frac{d}{2} + 3}} \left( 1 - \frac{d_\infty r_\infty}{(\frac{\Delta}{2})^{\eta_\infty}} \right)$$

$$\times 2 \sin[\tfrac{\pi}{4}(\Delta - 2\tilde{\alpha} - \gamma^{(+)}(\Delta)] \cos[\tfrac{\pi}{4}(\Delta - 2\tilde{\alpha} - \gamma^{(-)}(\Delta)] \; , \quad (5.5)$$

with

$$\frac{\gamma^{(+)}(\Delta) + \gamma^{(-)}(\Delta)}{2} = -\frac{d_\infty r_\infty}{(\frac{\Delta}{2})^{\eta_\infty}} \; , \qquad \frac{\gamma^{(+)}(\Delta) - \gamma^{(-)}(\Delta)}{2} = \frac{d_1 r_1}{(\frac{\Delta}{2})^{\eta_1 - \frac{d-2}{2}}} \cdot \quad (5.6)$$

This result shows that the behaviour of the functional kernel near $z = 1, \infty$ directly correlates with the positions of the zeros at large $\Delta$. In particular the zeros occur at $\Delta_n = 2\tilde{\alpha} + 2n + \gamma_n$ with

$$\gamma_n \underset{n \to \infty}{\sim} -\frac{d_\infty r_\infty}{n^{\eta_\infty}} + (-1)^n \frac{d_1 r_1}{n^{\eta_1 + 1 - \frac{d}{2}}} \cdot \quad (5.7)$$

If anomalous dimensions are to decay at large $n$ then necessarily:

$$\eta_\infty > 0 \,, \qquad \eta_1 > \tfrac{d-2}{2} \,. \tag{5.8}$$

For $d < 2$ we note that the earlier condition $\eta_1 \geq 0$ dominates so that, for any basis, the alternating-sign piece of $\gamma_n$ must decay at least as $n^{-1+d/2}$ — i.e. $n^{-1/2}$ in the $d = 1$ case. This is in agreement with the analysis of Appendix C.4, and is the reason for this slightly stronger assumption in the $d = 1$ case (see footnote 14).

We now recall that the interacting functionals admit the following decomposition in terms of the GFF basis:

$$f_n = \sum_{m=1}^{\infty} b_{nm} \, \hat{f}_m \,, \qquad \theta_n(\Delta) = \sum_{m=1}^{\infty} b_{nm} \, \hat{\theta}_m(\Delta) \,. \tag{5.9}$$

We would like to understand how this can be consistent with the results above. The coefficients $b_{nm}$ in these decompositions will be fixed in terms of the interacting functional's zeros $\Delta_n$. We will see that inputting the behaviour (5.7) of the zeros reproduces the right power-law behaviour (5.3) of the interacting functional kernels by summing up the GFF ones. In passing, we note that the direct analysis here is distinct, but morally equivalent to, that of Appendices C.3 and C.4, which instead passes through an integral representation for the functionals.

Let us set $\gamma_m$ as in (5.7). We want to check that the sum over GFF kernels produces the correct behaviour at $z = 1, \infty$ given in (5.3). Since none of the individual GFF kernels has satisfies it, this behaviour must arise from the infinite sum, and it is sufficient to focus on its tail. We now note that demanding the right duality properties for large $\Delta$ gives

$$\theta_n(\hat{\Delta}_m + \gamma_m) = 0 \quad \Rightarrow \quad b_{nm} = -\gamma_m \hat{\theta}'_n(\hat{\Delta}_m) + O(\gamma_m^2) \,, \tag{5.10}$$

with

$$\theta'_n(\hat{\Delta}_m) \underset{m \to \infty}{\sim} (-1)^m O(m^{-2\tilde{\alpha} + \frac{d}{2} - 3}) \,. \tag{5.11}$$

Zooming in on $z \to \infty$ and $z \to 1$ regions while doing the sum over large $m$ with $m/\sqrt{z}$ and $m\sqrt{z-1}$ fixed respectively, we can use the asymptotic expressions (A.18) to find

$$\begin{aligned}
\sum_m b_{nm} \, \hat{f}_m(z) &\underset{z \to \infty}{\supset} \hat{f}_n(z) \times \frac{d_\infty}{z^{\frac{\eta_\infty}{2}}} \,, \\
\sum_m b_{nm} \, \hat{f}_m(z) &\underset{z \to 1}{\supset} \hat{f}_n(1) \times 1 + d_1(z-1)^{\tilde{\alpha} + \frac{\eta_1}{2}} \,,
\end{aligned} \tag{5.12}$$

precisely as expected.

After these preliminaries, let us move on to the analysis of a concrete set of bases relevant

for an actual CFT computation.

## 5.2 An extremal family

Our example originates from considering a family of 1d CFT 4-point correlators of an elementary field $\phi$ with scaling dimension $\Delta_\phi$. In the OPE of $\phi$ with itself, after the identity operator, we denote the lowest-dimension operator as $\phi^2$ and its dimension as $\Delta_0$, which is chosen as an input. Below we set

$$g := \gamma_0 = \Delta_0 - 2\Delta_\phi \tag{5.13}$$

and consider $g \geq 0$ for definiteness. Asking that the OPE coefficient $\lambda_{\phi\phi\phi^2}$ is maximised then determines the correlator to take the form [19, 30]

$$z^{2\Delta_\phi} \langle \phi(\infty)\phi(1)\phi(z)\phi(0) \rangle = 1 + \sum_{n=0}^{\infty} a_n\, G_{\Delta_n}^{\mathtt{CFT}} , \tag{5.14}$$

where $G_\Delta^{\mathtt{CFT}} = z^\Delta\, _2F_1(\Delta, \Delta, 2\Delta, z)$ and

$$\begin{aligned}
\Delta_n &= 2\Delta_\phi + 2n + \gamma_n &\quad \text{with} \quad & \gamma_n = O(n^{-2}) , \\
a_n &= a_n^{\mathrm{gff}}(1 + \delta_n) &\quad \text{with} \quad & \delta_n = O(n^{-2}) .
\end{aligned} \tag{5.15}$$

The OPE contains only a discrete set of operators, denoted $(\phi^2)_n$, and both their OPE coefficients $a_n \equiv (\lambda_{\phi\phi(\phi^2)_n})^2$ and their dimensions $\Delta_n$ are uniquely determined by the choice of $\Delta_0$ — and known numerically to high precision [20, 33]. For $g \ll 1$, both $\delta_n$ and $\gamma_n$ are $O(g)$ and hence the correlator approaches that of a bosonic Generalized Free Field (GFF). For finite $g$, the above corresponds to a family of deformations of the GFF that preserve the density of states in the $\phi \times \phi$ OPE. This family cannot be extended beyond $g = 1$, since $\Delta_0 = 1 + 2\Delta_\phi$ represents an absolute upper bound on the dimension of the first operator in the OPE [17]. As we approach $g = 1$, the correlator approaches that of a Generalised Free Fermion, which saturates this bound. In detail, as $g \to 1$ we have $\Delta_n \to 1 + 2\Delta_\phi + 2n$ (the fermionic solution spectrum), but non-uniformly in $1 - g$, i.e. such that the asymptotics (5.15) remain valid for all $g < 1$.

For small $g$, the above CFT data matches that of a free scalar field $\Phi$ in AdS$_2$ (dual to the generalized free field $\phi$) with an additional quartic interaction $\propto g\,\Phi^4$. Thus, in that regime at least, it makes sense to say that the above CFT data arises from a local bulk dual.[19] For instance, to a few orders in perturbation theory in $g$, it is possible to consider the form factor $\langle \Phi^2 | \phi\phi \rangle$. After stripping off the universal prefactors, this form factor is

---

[19]Note that, for **finite** coupling $g$, this data is **not** described by a $\Phi^4$ theory in AdS$_2$. Specifically, the mismatch will occur at $O(g^4)$ where in $\Phi^4$ theory we would have new states $\sim \phi^4$ appearing in the $\phi \times \phi$ OPE.

written:

$$F^{\Phi^2}(z) = \sum_{n=0}^{\infty} c_n \, g_{\Delta_n}(z) \,, \qquad c_n := \lambda_{\phi\phi(\phi^2)_n} \, \mu^{\Phi^2}_{(\phi^2)_n} \, \mathcal{N}^{-1}_{\Delta_n} \,, \tag{5.16}$$

where for convenience we continue to use the normalisation (2.3) of the locality blocks. In our previous work [15] we considered this form factor, using the GFF functional bases to solve for the coefficients $c_n$ to leading order in the coupling $g$.

Given that the non-perturbative (albeit numerical) CFT data is available, a natural question is whether this local $\Phi^2$ operator can still be constructed at finite coupling $g$. Indeed, our construction of interacting functionals above answers this question affirmatively. Identifying $\Delta = \Delta_0$ as the fixed 'input' and $\Delta_{n \geq 1}$ as the 'sparse' spectrum dual to a basis of functionals, such form factors must be proportional to our extremal 'local block' soltuions:

$$F^{\Phi^2} = c_0 \, \mathcal{L}^{\{\Delta_n\}}_{\Delta_0} \,. \tag{5.17}$$

These interacting local block solutions $\mathcal{L}^{\{\Delta_n\}}_{\Delta}$ have been constructed explicitly in (3.40) and (4.24). They are guaranteed to be local as long as the dimensions $\Delta_n$ satisfy the required analyticity assumptions. There is much evidence for this: it is true in perturbation theory in the coupling $g$ and, even for $g = O(1)$, the $\gamma_n$ admit a systematic expansion in powers of $1/\hat{\lambda}_n$. In what follows we will take this as a working assumption.[20]

The local blocks, and hence the form factor above, are explicitly known, so the easiest way to evaluate them is by directly using the explicit formula (4.24). Nonetheless, we would also like to use this example to check our whole formalism. In the following subsections, we will construct the interacting functional kernels, functional actions, and extremal solutions corresponding to this example — first perturbatively for small $g$ and then numerically for finite $g$.

## 5.3 Interacting basis: perturbative computation

Let us first describe what happens in perturbation theory for $g \ll 1$. To understand how locality constrains the form factor (5.16), the simplest approach is to use the GFF sum rules with $\tilde{\alpha} = \Delta_\phi$. Writing $c_n = c_n^{(0)} + g \, c_n^{(1)}$ and expanding in $g$, we find (setting $\hat{\Delta}_n \equiv 2\Delta_\phi + 2n$ and assuming $\gamma_n = O(g)$):

$$\sum_{m=0}^{\infty} c_m \, \hat{\theta}_n(\Delta_m) = 0 \quad \Longleftrightarrow \quad \begin{cases} c_0^{(0)} \, \hat{\theta}_n(\hat{\Delta}_0) + c_n^{(0)} = 0, \\[2mm] g \, c_n^{(1)} + \sum_{m=0}^{\infty} c_m^{(0)} \, \gamma_m \, \hat{\theta}'_n(\hat{\Delta}_m) = 0 \,, \end{cases} \tag{5.18}$$

---

[20]We are also assuming that $F^{\Phi^2}$ is sufficiently polynomially bounded, so that $\mathcal{L}^{\{\Delta_n\}}_{\Delta_0}$ is the unique such solution up to proportionality. Indeed, this is physically reasonable since it is the case in perturbation theory in $g$.

which uniquely fixes the BOE data to this order.

Now let us see how to derive the same results by constructing an interacting functional basis satisfying

$$\theta_n(\Delta_m) = \delta_{nm} \ . \tag{5.19}$$

These functionals may always be expanded in the GFF basis:

$$\theta_n = \sum_{m=1}^{\infty} b_{nm}\, \hat{\theta}_m \ . \tag{5.20}$$

Imposing the duality relation (5.19) to $O(g)$ we find

$$
\begin{aligned}
\theta_n &= \hat{\theta}_n - \sum_{m=1}^{\infty} \gamma_m\, \hat{\theta}_n{}'(\hat{\Delta}_m)\, \hat{\theta}_m + \dots \\
\Longrightarrow \quad f_n &= \hat{f}_n - \sum_{m=1}^{\infty} \gamma_m\, \hat{\theta}_n{}'(\hat{\Delta}_m)\, \hat{f}_n + \dots \ .
\end{aligned}
\tag{5.21}
$$

We can also check whether this expression for the interacting functionals is consistent with our general formula (4.19). In perturbation theory that formula gives

$$
\begin{aligned}
\theta_n(\lambda) &= \hat{\theta}_n(\lambda)\left[1 - (\lambda_n - \hat{\lambda}_n)\theta_n'(\hat{\lambda}_n)\right] + \sum_{\substack{m=1 \\ m\neq n}}^{\infty} (\lambda_m - \hat{\lambda}_m)\frac{(\lambda - \hat{\lambda}_n)\hat{\theta}_n(\lambda)}{(\lambda - \hat{\lambda}_m)(\hat{\lambda}_n - \hat{\lambda}_m)} \\
&= \hat{\theta}_n(\lambda) - \sum_{m=1}^{\infty} (\lambda_m - \hat{\lambda}_m)\hat{\theta}_n'(\hat{\lambda}_m)\theta_m(\lambda)
\end{aligned}
\tag{5.22}
$$

which indeed matches the above.

Continuing, the interacting basis sum rules applied to the interacting solution are trivialized:

$$\sum_{\Delta} c_n\, \theta_n(\Delta_n) = 0 \quad \Leftrightarrow \quad c_n = -\theta_n(\Delta_0)\, c_0 \ . \tag{5.23}$$

Plugging in (5.21) and expanding in $g$ this gives:

$$c_n^{(0)} + g\, c_n^{(1)} = -c_0\, \theta_n(\hat{\Delta}_0) + c_0 \sum_{m=0}^{\infty} \gamma_m\, \hat{\theta}_n{}'(\hat{\Delta}_m)\, \hat{\theta}_m(\hat{\Delta}_0) \ , \tag{5.24}$$

which indeed agrees with the previous computation (5.18).

Further details can be worked out for special values of $\Delta_\phi$. For instance, for $\Delta_\phi = 1$ the

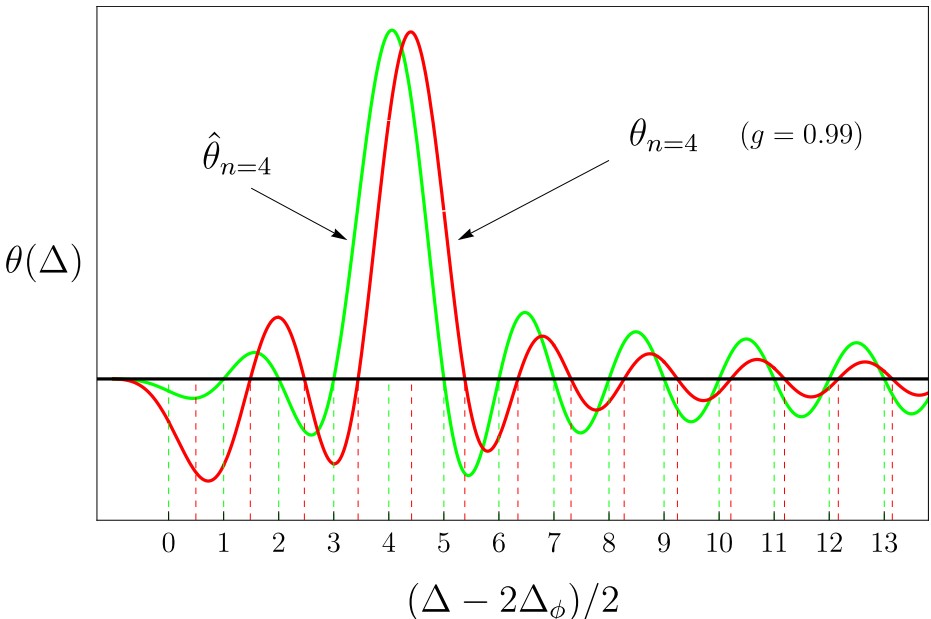

**Figure 4:** Functional actions. The figure shows a comparison between a GFF functional action and an interacting one, the latter corresponding to an extremal spectrum with $\gamma_0 = 0.99$ (rescaled by a positive function of $\Delta$ for clarity). The dashed lines at the bottom of the plot in green and red mark the functionals' zeros, i.e. the dimensions of the spectra of the corresponding solutions to crossing. For large $\Delta$, as expected, the actions become identical.

anomalous dimensions are given analytically by (see e.g. [19]):

$$\gamma_n = \frac{\gamma_0}{(n+1)(2n+1)} = \frac{2\gamma_0}{\hat{\Delta}_n(\hat{\Delta}_n - 1)} \ . \tag{5.25}$$

In this case, as explained in appendix E, it is possible to perform the sum in (5.21) explicitly, finding

$$\frac{f_1(z) - \hat{f}_1(z)}{\gamma_0} = \frac{z^{\frac{3}{2}} \sinh^{-1}(\sqrt{z-1})}{3\sqrt{z-1}} - \frac{(24z \log(2) + 6(2z+1) \log(z) + 7)}{77} \ . \tag{5.26}$$

As is manifest from this explicit expression, upon adding interactions, the functional kernels have now developed a branch cut running along $(-\infty, 1)$ — consistently with the general expectations of Section 5.1 and Appendix C.

## 5.4 Interacting bases: numerical computation

We will now discuss a method for a non-perturbative but numerical construction of the local $\Phi^2$ operator. This method provides a bottom-up consistency check for the above closed form expressions for interacting functionals. The idea is as follows. Starting from the CFT data $\Delta_n$, we will construct interacting functionals as the large-$N$ limit of combinations of GFF

functionals:

$$\theta_n^N = \sum_{m=1}^{N} b_{nm}^N \, \hat{\theta}_m \ . \tag{5.27}$$

The duality conditions:

$$\theta_n^N(\Delta_m) = \delta_{nm} \quad \Longleftrightarrow \quad \sum_{m=1}^{N} b_{nm}^N \, \hat{\theta}_m(\Delta_n) = \delta_{nm} \ . \tag{5.28}$$

provide finitely many linear constraints on the coefficients $b_{nm}^N$, which may be solved. As $N$ increases, this leads to systematically better approximations to the exact interacting functional $\theta_n$. This limit is guaranteed to converge in the sense $b_{nm}^N \to b_{nm}$.[21] This follows from the special properties of Riesz bases, and would not be true for a general functional basis (i.e. derivatives).

In fact, it is possible to improve the convergence in $N$ by subtracting off the following perturbative-like 'tail'

$$\theta_n^N \to \theta_n^N + T_n^N \ , \qquad T_n^N := \sum_{m=N+1}^{\infty} \gamma_m \, \theta_n^{N\prime}(\hat{\Delta}_m) \, \hat{\theta}_m \ . \tag{5.29}$$

The tail $T_n^N$ approximates the contributions of the operators of large dimension. The addition of this tail introduces a small error in the duality conditions that were previously solved. To fix this, one may follow an iterative procedure, writing

$$\theta_n^{N,k} = \sum_{m=1}^{N} b_{nm}^{N,k} \, \hat{\theta}_m + T_n^{N,k-1} \ , \tag{5.30}$$

and solving for $b_{nm}^{N,k}$ iteratively. In practice, two or three iterations suffice to find negligible changes in the coefficients. Once we have the coefficients $b_{nm}$, then we can obtain not only the functional actions but also the interacting kernels $f_n = \sum_m b_{nm} \, \hat{f}_m$.

The results of this procedure are shown in Figures 4 and 5. The first figure plots the interacting functional action $\theta_n(\Delta)$ while the second plots the functional kernels $f_n(z)$, for various choices of the coupling $g$ (i.e. $\Delta_0$). As an important cross-check, we find that:

$$\lim_{N \to \infty} b_{nm}^N = \theta_n(\hat{\Delta}_m) \tag{5.31}$$

---

[21]This can be seen by noting that the functional kernels $f_n^N$ corresponding to (5.27) are the first $N$ elements of a Riesz basis dual to $\{p_{n \leq N}, \hat{p}_{n > N}\}$. By the analysis of Appendix B.2, the latter are the image of $\{\hat{p}_n\}$ under a bounded, invertible operator $M_N = 1 - K_N$, with $K_N$ defined in (B.43). This operator converges in operator norm $M_N \to M$ where $M(\hat{p}_n) = p_n$, and hence so does $M_N^{-1} \to M^{-1}$. Then we find $b_{nm}^N = \langle \hat{f}_n | M_N^{-1} | \hat{p}_m \rangle \to b_{nm} = \langle \hat{f}_n | M^{-1} | \hat{p}_m \rangle$ as required.

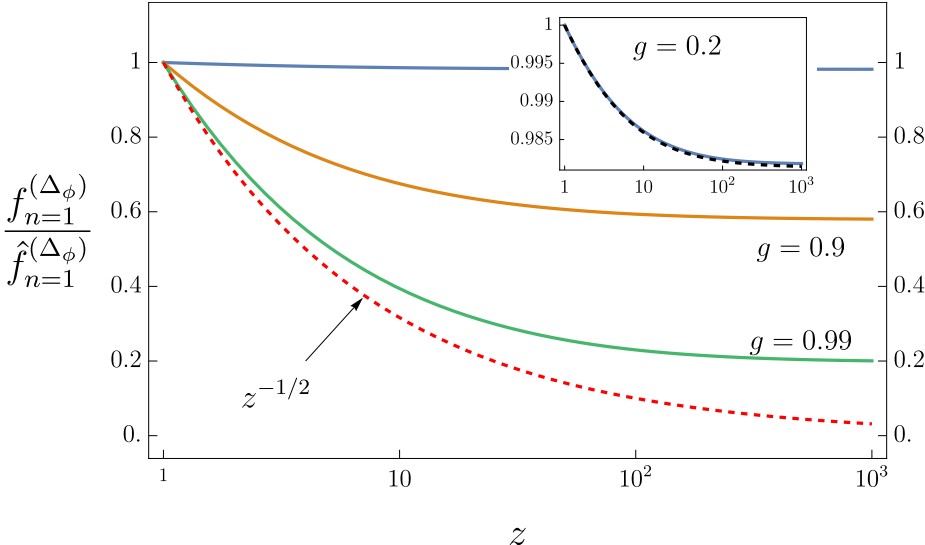

**Figure 5:** Interacting functional kernels with $n = 1$ (rescaled by the GFF functional kernel, which is just a power of $z$). The figure shows the kernels for various choices of $g$, with the inset displaying a better view of the kernel for $g = 0.2$. The black dashed curve corresponds to the analytic result given in equation (5.26). As $g \to 1$, the curves approach $z^{-\frac{1}{2}}$ (the red dashed curve), so that $f_1^{(\tilde{\alpha})} \to \hat{f}_1^{(\tilde{\alpha}+\frac{1}{2})}$ (with $\tilde{\alpha} = \Delta_\phi$). The plots were done for the particular choice $\Delta_\phi = 1$.

with the RHS computed using the exact formula (4.19) (or in practice the approximations (4.21) and (4.22), which ultimately correspond to the same approximation scheme used above). Incidentally, the plots demonstrate that, as the coupling is increased, the interacting functionals approach the GFF basis with $\tilde{\alpha} = \Delta_\phi + 1/2$ as they should, since the CFT data is approaching the Generalized Free Fermion solution with spectrum $\Delta_n = 2(\Delta_\phi + 1/2) + 2n$.

With the functional actions well under control, we can compute the BOE data. For instance, the construction of the interacting functionals above gives

$$c_n = -c_0\, \theta_n^N(\Delta_0) + O(\gamma_N^2) \qquad \text{for} \quad 1 \leq n \leq N \, , \tag{5.32}$$

where quadratic accuracy follows from our inclusion of the tail. It is also possible to predict the BOE data to arbitrarily high $n$ in terms of the lower $n$ data, simply by using the higher-$n$ GFF functional sum rules to get

$$c_n = -\sum_{m=0}^{N} \frac{\hat{\theta}_n(\Delta_m)}{\hat{\theta}_n(\Delta_m)} + O(\gamma_N) \qquad \text{for all} \quad n > N \tag{5.33}$$

This in turn can be improved to $O(\gamma_N^2)$ by making the replacement

$$\hat{\theta}_n \to \hat{\theta}_n - \sum_{m=N+1}^{\infty} \gamma_m\, \hat{\theta}_n'(\hat{\Delta}_m)\, \hat{\theta}_m \tag{5.34}$$

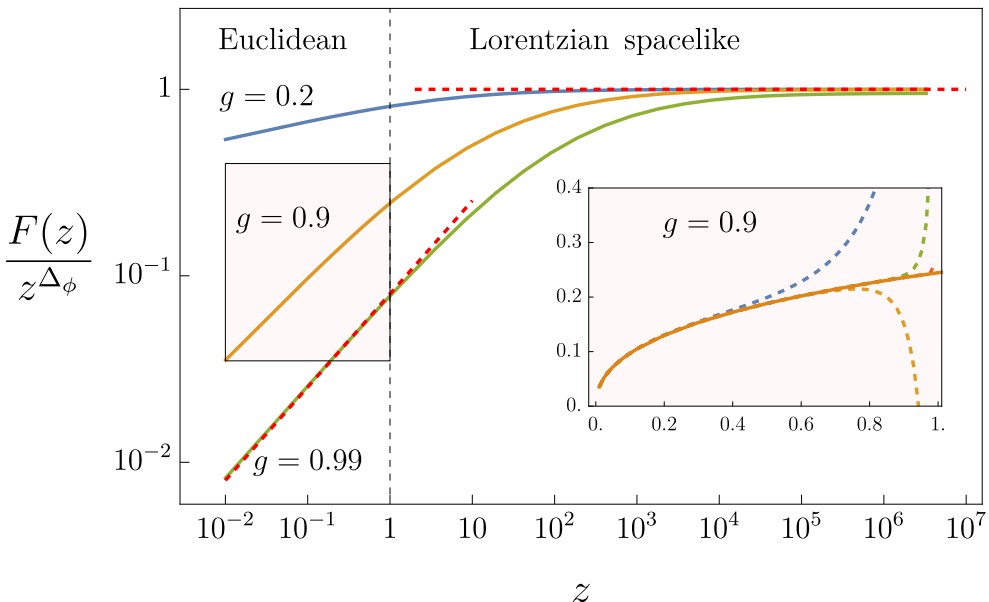

**Figure 6:** Form factors. In the figure we show the interacting form factors corresponding to various values of the coupling $g$. As the coupling increases the form factor interpolates between $z^{\Delta_\phi+\frac{1}{2}}$ (the red dashed sloping line) and $z^{\Delta_\phi}$ (the horizontal dashed curve). The form factors were computed by representing them as sums of local blocks, with the corresponding coefficients $\mu\lambda$ computed as discussed in the main text. The result is local by construction, and in particular it shows no singularity for $z \geq 1$. As a consistency check we show in the inset that in the Euclidean region ($z \leq 1$) this representation agrees with a direct compution of the form factor as sum of conformal blocks. As we include more terms in that sum (shown as successive dashed curves) the result nicely approaches the local block result in the entire Euclidean region.

everywhere in the above formula. The outcome of this procedure is to improve our estimates of the BOE data, such that the error in all quantities is now $O(\gamma_N^2)$.

With the BOE data in hand, we can reconstruct the local form factor. As noted above, this form factor can be identified with the interacting local block $\mathcal{L}_{\Delta_0}^{\{\Delta_n\}}$. We plot it in Figure 6. The result contains rather rich physics: for instance, by scaling the coupling $g \to 1$ and $z \to \infty$ at the same time, one probes the flat space RG flow for the bulk QFT (which happens well above the AdS scale), ending in a free fermion CFT in the IR [33].

# 6  Discussion and outlook

## 6.1  Extremal solutions vs bases

In this paper we described a natural language for the locality bootstrap problem. Candidate form factors can be mapped to hyperfunctions in a suitable space $\mathcal{HF}$, with dual space

$\mathcal{A}$ whose elements are functional kernels. This language makes precise the question of completeness of a set of functionals $\{f_n\}$, which had to be tackled in previous works [17, 19, 30] from a more 'bottom-up' perspective: completeness simply means that the set $\{f_n\}$ is complete in $\mathcal{A}$, i.e. that any element of $\mathcal{A}$ is a limit of linear combinations of the $f_n$. We constructed a large family of such complete sets by asking for the $f_n$ to form Schauder bases for $\mathcal{A}$ dual to distinguished bases in $\mathcal{HF}$ made up of blocks with specific sets of scaling dimensions $\{\Delta_n\}$.

Such complete bases are associated with canonical solutions to the locality problem, 'local blocks' $\mathcal{L}_\Delta^{\{\Delta_n\}}$ given in equation (4.24). These are local form factors which can be thought of as 'extremal', in the sense that their BOE contains the minimal amount of blocks compatible with locality.[22] We have thus established a link between **bases** and **extremal solutions**. We expect this link to be quite general, and hold across a variety of distinct bootstrap problems. It seems therefore that characterising bases for the relevant spaces is a useful path for an understanding of bootstrap equations.

In our construction, an intermediate step involved understanding Riesz bases for $L^2$. This is a classical problem in functional analysis. For example, for the space $L^2(-\pi, \pi)$, the exponential functions $\{e^{int}\}_{n \in \mathbb{Z}}$ form an orthonormal basis, but one may consider different 'spectra' $\{\lambda_n\}$ and ask whether $\{e^{i\lambda_n t}\}_{n \in \mathbb{Z}}$ still forms a Riesz basis. The solutions of that problem have been classified [34] (in Russian; see the recent review [35]), with a sufficient condition being $\sup_n |\lambda_n - n| < \frac{1}{4}$ (a famous result of Kadets [36]). In our case, we dealt instead with the sets $\{\mathcal{I}_z g_{\Delta_n}(z)\}$ and established sufficient conditions on the spectrum $\Delta_n$ for them to form a Riesz basis for $L_\mu^2(1, \infty)$:

$$|\Delta_n - (2\tilde\alpha + 2n)| \overset{n \to \infty}{\lesssim} n^{-\epsilon}, \qquad |\tilde\alpha| < \frac{1}{2}, \; \epsilon > 0 . \tag{6.1}$$

Importantly, in our setting, in order to for these to uplift to Schauder bases for the spaces $\mathcal{A}$ and $\mathcal{HF}$, it was also necessary that the sequence $\{\Delta_n\}$ has certain analyticity properties, adding a significant further constraint.

Let us mention an interesting set of bases that do not satisfy our assumptions on $\{\Delta_n\}$, but nevertheless lead to functionals satisfying our properties 1) and 2) in the introduction. These 'Zettl bases' [37, 38] rely on the fact that the block discontinuities $\mathcal{I}_z g_\Delta$ are eigenfunctions of the Casimir operator (3.7). This operator can be made Hermitian on $L_\mu^2$ by choosing appropriate boundary conditions at $z = 1, \infty$, in such a way that special sets $\{\mathcal{I}_z g_{\Delta_n}\}$ remain allowed eigenfunctions. The net result is a continuous family of sequences $\{\Delta_n\}$ (labeled by $\Delta_1$ and computable numerically) such that $\{\mathcal{I}_z g_{\Delta_n}\}$ form *orthonormal* bases — thus with dual functional kernels trivially given by the blocks themselves. The reason we did not find these basis in our construction is that such sequences have $\Delta_n - 2n = O(1/\log(n))$ (not satisfying our assumptions), which does not seem to be immediately physically relevant for bootstrap applications.

---

[22]For a fixed polynomial growth at $z = \infty$.

## 6.2 Implications for numerics

The results of this paper have interesting implications for the numerical bootstrap program in $d = 1$ (and probably for higher $d$ as well, but we focus the following discussion on that case). One question of interest is: in demanding that the CFT data supports the existence of local bulk operators, does one get stronger bounds? In other words, does combining crossing with locality lead to stronger constraints?

To get a handle on this question, first we note that, to understand the bootstrap bounds, it is sufficient to focus on the construction of the extremal solutions of crossing that saturate them.[23] Thus, to answer the question, one only has to consider extremal solutions of crossing, and ask whether it is possible to solve the locality constraints using them as inputs.

In Section 5, we saw for a particular family of extremal solutions that it was indeed possible to solve the locality constraints, and we explicitly reconstructed a local form factor associated to it, corresponding to some operator $\Phi^2$ in the bulk of AdS. In fact, we expect this to be possible for all extremal solutions of crossing, which always consist of finite sets of fixed 'input' CFT data (specifying which bound is to be saturated), completed by an infinite collection of 'double-trace' operators, which are dynamically determined by crossing and optimality. There are both analytical arguments and ample numerical evidence that the scaling dimensions of such operators must always asymptote to either $\Delta_n = 2\Delta_\phi + 2n$ or $\Delta_n = 1 + 2\Delta_\phi + 2n$ and meet our other assumptions on 'sparse' spectra: being analytic and decaying as some power for large $n$. The results of this paper then imply that we can always choose a basis of locality functionals that will be dual to such spectra, and hence reconstruct a local form factor, which will in practice always be a finite sum of the 'local block' solutions for that basis.

Thus we find that generic extremal solutions are compatible with the existence of local bulk operators. Intuitively what has happened is that, although bulk locality introduces infinitely many new constraints, it also introduces infinitely many new pieces of data to be fixed: the BOE coefficients. Non-trivially, these two infinities line up on the nose, thanks to the fact that the bases we were able to construct for locality are precisely adapted to those spectra that saturate CFT bounds.

The good news is that we can still learn something from locality and obtain additional bounds on top of crossing, by including some additional input. In fact, since we already have a tight matching of infinite constraints and degrees of freedom, any finite mismatch will suffice to create the required tension. The simplest thing is just to lower the parameter $\tilde{\alpha}$. In particular, lowering this value by one unit corresponds to adding one extra locality constraint. This can be motivated if, in some case, one already has a notion of what the correct value should be. In a numerical setup, bounds would thus have to become tighter. For instance, for the family of examples we considered in Section 5, choosing $\tilde{\alpha} = \Delta_\phi - 1$ would lead to the absence of a solution (contingent on fixing an overall normalisation of the

---

[23]A detailed characterization and study of such extremal solutions will be given elsewhere. For the purposes of this section, it is sufficient to think of them as the solutions of crossing saturating a bound on the CFT data.

bulk field, e.g. fixing one of the $c_n$ coefficients). The upper bound on the OPE coefficient $a_0$ in (5.14) would thus have to decrease, as the extremal solution under consideration is now ruled out. Perhaps more interestingly, special choices of bulk operators such as the stress-tensor or conserved currents already come with added constraints — e.g. those developed for the stress-tensor in [26]. This makes sense, since such operators know about the bulk Lagrangian and cannot be compatible with arbitrary boundary data.

## 6.3   Outlook

To conclude, let us briefly mention a few of the interesting research directions in this subject.

Firstly, it would be very interesting to apply our machinery to other bootstrap problems, particularly the crossing equation. In particular, one of our most significant results is the characterisation of the space of functional actions in Section 4 and the explicit product formula (4.19) expressing the functional actions in terms of their zeros. We believe that this approach may similarly give explicit formulae for the functional actions in the crossing case — perhaps yielding greater analytic control over the interacting extremal solutions there [20]. In general, working with the space of functional actions is appealing since it bypasses the difficult construction of functional kernels and the laborious computation of their action. In fact, similar logic already had success in the past, although it was not formulated in our language [39–42].

Secondly, we sketched in [15] the eventual generalisations of our work for bulk theories with gauge and gravitational symmetries. An appropriate choice of dressing will enforce specific non-localities in certain form factors, which will generally be distributions:

$$F^{(\tilde{\alpha})} = H^{\text{dressing}} . \tag{6.2}$$

Our formulation of functionals as Schauder bases immediately provides a way to convert this 'source' term to source terms in the locality sum rules. Decomposing

$$H^{\text{dressing}} = \sum_n (f_n, H^{\text{dressing}}) \, g_{\Delta_n}^{(\tilde{\alpha})} , \tag{6.3}$$

the sum rules become

$$\sum_\Delta c_\Delta \, \theta_n(\Delta) = (f_n, H^{\text{dressing}}) . \tag{6.4}$$

For instance it would be very interesting to see if this prescription reproduces bulk radial Wilson-line dressings for charged operators.

Finally, our work has been concerned with characterizing locality around the AdS vacuum. However, in gravitational theories with a large $N$ limit, the bulk Hilbert space splits into various superselection sectors occupied, e.g., by black holes states. It would be interesting to try to formulate locality constraints in this context and make contact with

the proposals of [43, 44].

## Acknowledgements

This work was co-funded by the European Union (ERC, FUNBOOTS, project number 101043588). Views and opinions expressed are however those of the author(s) only and do not necessarily reflect those of the European Union or the European Research Council. Neither the European Union nor the granting authority can be held responsible for them. NL was supported by the Institut Philippe Meyer at the École Normale Supérieure in Paris. We acknowledge discussions with K. Ghosh, P. Kravchuk, M. Meineri and S. Rychkov.

# A   Compendium of useful formulae

For completeness and convenience, we recall in this section the results of our previous work [15] as well as various useful results. The locality constraint is equivalent to the following sum rules on the BOE data:

$$\sum_\Delta c_\Delta \hat{\theta}_n(\Delta) = 0 \, , \qquad n \geq 1 \, , \tag{A.1}$$

with $c_\Delta$ defined in (2.3). The functional actions are given as

$$\hat{\theta}_n(\Delta) = (\hat{f}_n, g_\Delta^{(\tilde{\alpha})}) = \int_1^\infty \frac{\mathrm{d}z}{\pi} \, \hat{f}_n(z) \frac{\mathcal{I}_z g_\Delta(z)}{z^{1+\tilde{\alpha}}}$$
$$= \langle \hat{h}_n | \mathcal{I}_z g_\Delta \rangle = \int_1^\infty \mathrm{d}z \, \mu(z) \, \hat{h}_n(z) \, \mathcal{I}_z g_\Delta(z) \tag{A.2}$$

with the measure

$$\mu(z) = \frac{1}{z^2} \left( \frac{z}{z-1} \right)^{1-\frac{d}{2}} . \tag{A.3}$$

To derive the functional action, a useful representation is obtained by deforming the contour of integration to wrap the cut running along the negative real axis,

$$\theta_n(\Delta) = \sin[\tfrac{\pi}{2}(\Delta - 2\tilde{\alpha})] \int_{-\infty}^0 \frac{\mathrm{d}z}{\pi} f_n(z)(-z)^{-1-\tilde{\alpha}} \tilde{g}_\Delta(z) \tag{A.4}$$

with

$$\tilde{g}_\Delta(z) = \mathcal{N}_\Delta \left( \frac{z}{z-1} \right)^{\frac{\Delta}{2}} {}_2F_1 \left( \tfrac{\Delta}{2}, \tfrac{\Delta+2-d}{2}, \Delta + 1 - \tfrac{d}{2}; \tfrac{z}{z-1} \right) . \tag{A.5}$$

The $\hat{h}_n \in L^2$ are given by

$$\hat{h}_n(z) = i^{\tilde{\alpha}}(\hat{f}_n) = \frac{\hat{f}_n(z)}{\pi\mu(z)z^{1+\tilde{\alpha}}} , \qquad (A.6)$$

and the functional kernels $f_n$ are polynomials of degree $n$ in $1/z$ without constant term, given explicitly by

$$\hat{f}_n(z) = \pi\frac{(-1)^{n-1}}{(n-1)!}(\hat{\Delta}_n - \tfrac{d}{2})\Gamma(2\tilde{\alpha} + n + 1 - \tfrac{d}{2})$$

$$\times \frac{1}{z}\,{}_3\tilde{F}_2(1, 1-n, 2\tilde{\alpha} + n + 1 - \tfrac{d}{2}; 1+\tilde{\alpha}, 1+\tilde{\alpha}; \tfrac{1}{z}) . \quad (A.7)$$

Note that $\hat{h}_n$ solves an inhomogeneous Casimir equation [15],

$$\left[C_z - \lambda_{\hat{\Delta}_n}\right]\hat{h}_n(z) = \frac{A_n}{\mu(z)z^{1+\tilde{\alpha}}} , \quad A_n = -\frac{4\,\Gamma(\hat{\Delta}_n + 1 - \tfrac{d}{2})}{\Gamma(\tilde{\alpha})^2\Gamma(n)\left(1 + \tfrac{d}{2} - \hat{\Delta}_n\right)_{n-1}} , \qquad (A.8)$$

with

$$C_z := 4(1-z)^{1-d/2}z^{1+d/2}\partial_z[(1-z)^{d/2}z^{1-d/2}\partial_z] . \qquad (A.9)$$

The resulting functional actions are:

$$\hat{\theta}_n(\Delta) = \frac{2}{\pi}\frac{\left(\frac{\hat{\Delta}_n - 2\tilde{\alpha}}{2}\right)_{1-\frac{d}{2}+2\tilde{\alpha}}}{\left(\frac{\Delta - 2\tilde{\alpha}}{2}\right)_{1-\frac{d}{2}+2\tilde{\alpha}}}\frac{(2\hat{\Delta}_n - d)\sin\left[\frac{\pi}{2}(\Delta - \hat{\Delta}_n)\right]}{(\hat{\Delta}_n + \Delta - d)(\Delta - \hat{\Delta}_n)} , \qquad (A.10)$$

with $\hat{\Delta}_n = 2\tilde{\alpha} + 2n$. In particular they satisfy the duality conditions

$$\hat{\theta}_n(\hat{\Delta}_m) = \delta_{n,m} , \qquad n, m \geq 1 . \qquad (A.11)$$

They can be rewritten in a way that makes their invariance under $\Delta \leftrightarrow d - \Delta$ manifest:

$$\hat{\theta}_n(\Delta) = \frac{2\,(-1)^n\,(2\hat{\Delta}_n - d)}{(\Delta + \hat{\Delta}_n - d)(\Delta - \hat{\Delta}_n)\,\Gamma\left(\frac{2+2\tilde{\alpha}-\Delta}{2}\right)\Gamma\left(\frac{2+2\tilde{\alpha}-d+\Delta}{2}\right)}\left(\frac{\hat{\Delta}_n - 2\tilde{\alpha}}{2}\right)_{1-\frac{d}{2}+2\tilde{\alpha}} . \qquad (A.12)$$

The actions satisfy the asymptotics:

$$|\hat{\theta}_n(\Delta)| \underset{\Delta\to\infty}{=} O(\Delta^{-2\tilde{\alpha}+\frac{d}{2}-3})$$

$$|\hat{\theta}_n(\Delta)| \underset{n\to\infty}{=} O(n^{2\tilde{\alpha}-\frac{d}{2}}) \qquad (A.13)$$

The conformal blocks and the functional kernels have the following asymptotic forms for $z \in \mathbb{C}\backslash(1, \infty)$.

$$g_\Delta(z) \underset{\Delta \to \infty}{\sim} \frac{2^{\frac{3}{2}} \Delta^{\frac{d-3}{2}}}{\sqrt{\pi}} \frac{[1 - r(z)]^{1 - \frac{d}{2}}}{\sqrt{1 + r(z)}} r(z)^{\frac{\Delta}{2}} \,,$$

$$\frac{\hat{f}_n(z)}{z^{1+\tilde{\alpha}}} \underset{n \to \infty}{\sim} \frac{\mathrm{d}r}{\mathrm{d}z} \frac{\sqrt{\pi}}{2^{\frac{3}{2}} (\hat{\Delta}_n)^{\frac{d-3}{2}}} \frac{\sqrt{1 + r(z)}}{[1 - r(z)]^{1 - \frac{d}{2}}} r(z)^{-1 - \tilde{\alpha} - n} \,,$$

(A.14)

where

$$r(z) = \frac{1 - \sqrt{1 - z}}{1 + \sqrt{1 - z}} \,. \tag{A.15}$$

We also have with $\theta = \frac{1}{i} \log\left(\frac{1 + i\sqrt{z-1}}{1 - i\sqrt{z-1}}\right)$:

$$\mathcal{I}_{z \geq 1} \, g_\Delta(z) \underset{\Delta \to \infty}{\sim} \frac{\left(\frac{\Delta}{2}\right)^{\frac{d-3}{2}}}{\sqrt{\pi}} \frac{[\sin(\theta/2)]^{\frac{1-d}{2}}}{\sqrt{\cos(\theta/2)}} \sin\left[\frac{\Delta}{2}\theta + \frac{1}{4}[\pi(d-1) - d\theta]\right] \,. \tag{A.16}$$

Other useful asymptotic expressions are given by:

$$g_\Delta(z) \underset{\substack{z \to 1 \\ \Delta \to \infty}}{\sim} \frac{2}{\pi} \frac{\Gamma(\frac{\Delta}{2})}{\Gamma(\frac{\Delta+2-d}{2})} (1 - z)^{\frac{2-d}{4}} K_{\frac{d-2}{2}}(\Delta\sqrt{1-z}) \,,$$

$$\tilde{g}_\Delta(z) \underset{\substack{z \to -\infty \\ \Delta \to \infty}}{\sim} \frac{2}{\pi} \frac{\Gamma(\frac{\Delta}{2})}{\Gamma(\frac{\Delta+2-d}{2})} K_0(\Delta/\sqrt{-z}) \,,$$

(A.17)

and similarly for the kernels

$$\hat{f}_n(z) \underset{\substack{n \to \infty \\ z \to 1}}{\sim} \frac{2\pi(-1)^{n-1} n^{2\tilde{\alpha} - \frac{d}{2}}}{\Gamma(\tilde{\alpha})^2} + 2\pi \, n^{2 - \frac{d}{2}} (z - 1)^{\frac{d-2}{4}} J_{1 - \frac{d}{2}}(2n\sqrt{z-1}) \,,$$

$$\hat{f}_n(z) \underset{\substack{z,n \to \infty \\ z \sim n^2}}{\sim} \frac{2\pi}{z} (-1)^{n-1} n^{2\tilde{\alpha} + 2 - \frac{d}{2}} \, {}_1\tilde{F}_2(1, 1 + \tilde{\alpha}, 1 + \tilde{\alpha}, -n^2/z)$$

(A.18)

All asymptotic expressions can be derived from the fact that the blocks and functionals satisfy a Casimir equation, cf. (3.6) and (A.8).

# B  Riesz bases

This appendix contains the proofs that the GFF functionals and the interacting functionals form Riesz bases for $L^2$. Before beginning, we point out that a sequence of functions being a Riesz basis depends on the choice of normalisation: their norms need to be bounded. Thus

in this section it will be convenient to use the normalisation

$$p_\Delta(z) := \sqrt{\Delta - \frac{d}{2}} \; \left(\frac{\Delta}{2}\right)_{1-\frac{d}{2}} \mathcal{I}_z \, g_\Delta(z) \tag{B.1}$$

and define

$$p_n := p_{\Delta_n}\,, \qquad \hat{p}_n := p_{\hat{\Delta}_n}\,, \qquad \hat{\Delta}_n = 2\tilde{\alpha} + 2n. \tag{B.2}$$

where for $p_n$ the particular implied sequence of $\Delta_n$ will be clear from context. Similarly we have to modify the normalisation of the dual elements $h_n$. Overall there is a mismatch in normalisation conventions with the rest of the paper, which can be summarised as:

$$p_n = \nu_n \, p_n^{\text{above}}\,, \qquad h_n = \nu_n^{-1} \, h_n^{\text{above}} \qquad \text{with} \quad \nu_n = \sqrt{\Delta_n - \frac{d}{2}} \; \left(\frac{\Delta_n}{2}\right)_{1-\frac{d}{2}}\,. \tag{B.3}$$

## B.1   GFF Riesz bases

In this section we will prove property (a) for the GFF 'double trace' spectrum in Section 3.2, i.e. that the set $\{\hat{p}_n\}$ forms a Riesz basis for $L^2$. To achieve this, we just need to show there is a bounded, invertible, linear map:

$$T : L^2 \to L^2 \tag{B.4}$$
$$e_n \mapsto \hat{p}_n\,,$$

where $e_n$ is some orthonormal basis (any choice is equivalent). We remind the reader that our $L^2$ space is always defined with the non-trivial measure

$$\langle f|g\rangle := \int_1^\infty \frac{\mathrm{d}z}{z^2} \left(\frac{z}{z-1}\right)^{1-d/2} f(z)\, g(z)\,. \tag{B.5}$$

**Invertibility.**   Our approach for demonstrating invertibility of the putative map (B.4) is to explicitly construct an inverse. This is easy since we already explicitly know that the elements $\hat{h}_n$ that are dual to the GFF blocks $\hat{p}_n$ (as reviewed in Section A):

$$\langle \hat{h}_m|\hat{p}_n\rangle = \delta_{mn}\,, \qquad m, n \geq 1\,. \tag{B.6}$$

Hence we may define a new map $S$ as

$$S : L^2 \to L^2 \tag{B.7}$$
$$e_n \mapsto \hat{h}_n\,. \tag{B.8}$$

Assuming we can prove that $T$ and $S$ define bounded operators, then $S^\dagger = T^{-1}$, since the duality relation (B.6) gives

$$S^\dagger T e_n = e_n \ , \qquad n \geq 1 \ . \tag{B.9}$$

More precisely, this only shows that $S^\dagger$ is a left-inverse of $T$. To conclude the proof of invertibility we need to show that $S^\dagger$ is one-to-one (or equivalently that $T$ is surjective). The trick will be that the GFF kernels $\hat{h}_n$ are (up to some measure factors) polynomials in $1/z$, so can be related to known bases by taking **finite** linear combinations.

Let us see this in detail. We must show that if $S^\dagger g = 0$ for $g \in L^2$, then $g = 0$. For any $n \geq 1$, we have

$$\langle \hat{h}_n | g \rangle = \langle e_n | S^\dagger g \rangle = 0 \ . \tag{B.10}$$

But the GFF functional kernels take the special form

$$\hat{h}_n(z) = \nu_n^{-1} \, z^{-\tilde{\alpha}} \left( \tfrac{z}{z-1} \right)^{\frac{d}{2}-1} z \hat{f}_n(z) \tag{B.11}$$

where $z \hat{f}_n(z)$ are polynomials of degree $(n-1)$ in $1/z$. Hence it follows that

$$\langle z^{-\tilde{\alpha}} \left( \tfrac{z}{z-1} \right)^{\frac{d}{2}-1} j_n(1/z) | g \rangle = 0 \tag{B.12}$$

for any polynomials $j_n$. Now let us take $j_n(1/z) \propto J_{n-1}^{(1-d/2,2\tilde{\alpha})}(\tfrac{2}{z}-1)$ Jacobi polynomials. These form a complete orthonormal basis for a different space, $\tilde{L}^2$:

$$\delta_{n,m} = \langle j_n | j_m \rangle_{\tilde{\mu}} = \int_1^\infty dz \, \tilde{\mu}(z) \, j_n(1/z) j_m(1/z) \ , \qquad \tilde{\mu}(z) = \frac{1}{z^{2+2\tilde{\alpha}}} \left( \frac{z}{z-1} \right)^{\frac{d}{2}-1} \ . \tag{B.13}$$

We now reinterpret equation (B.12) as

$$\langle j_n | \tilde{g} \rangle_{\tilde{\mu}} = 0 \ , \qquad \tilde{g}(z) = g(z) \, z^{\tilde{\alpha}} \left( \frac{z}{z-1} \right)^{1-\frac{d}{2}} \ . \tag{B.14}$$

Since $j_n$ are complete in $\tilde{L}^2$ then $g = \tilde{g} = 0$ as long as $\tilde{g} \in \tilde{L}^2$, but this is true since $\|\tilde{g}\|_{\tilde{\mu}}^2 = \|g\|^2$ as is easily checked. We conclude that $S^*$ is 1-1.

**Schur's test.** We want to show that the operators $S$ and $T$ defined above are well-defined, bounded operators. Note that in general if $M : e_n \mapsto q_n$ for $e_n$ an orthonormal basis and $q_n$ some elements of $L^2$, then $M$, $M^*$ and $M^*M$ are of course well-defined on finite linear combinations of the $e_n$, and they satisfy

$$\|Mx\|^2 = \langle x | M^\dagger M x \rangle \leq \|x\| \, \|M^\dagger M x\| \leq \|M^\dagger M\| \, \|x\|^2 \ , \tag{B.15}$$

where we have used the Cauchy-Schwarz inequality and defined $\|M^\dagger M\| = \sup_x \|M^\dagger M x\|$ as usual (with $x$ running over finite combinations of $e_n$).

We wish to extend $M$ to a linear operator on the full $L^2$ space by the definition $M(\lim x_n) = \lim M(x_n)$, and similarly for $M^\dagger M$. To show this is possible, it is sufficient to apply Schur's test to the matrix elements $M^2_{mn} := \langle e_m | M^\dagger M | e_n \rangle = \langle q_m | q_n \rangle$. This classic result states that, if there exist positive sequences $\beta_n$, $\alpha_n$ and numbers $\lambda$, $\mu$ such that

$$\sum_n |M^2_{mn}| \, \beta_n \leq \lambda \, \alpha_m \ , \quad \sum_m |M^2_{mn}| \, \alpha_m \leq \mu \, \beta_n \ , \tag{B.16}$$

then $M^\dagger M$ is bounded with $\|M^\dagger M\| \leq \sqrt{\lambda\mu}$, and hence is a well-defined map on all of $L^2$. It follows from (B.15) that both $M$ and $M^\dagger$ are also well-defined and bounded.

Below we will apply this strategy to prove first boundedness of $S$ and then $T$.

**Boundedness of $S$.** In this case we have

$$\langle e_n | S^* S | e_m \rangle = \langle \hat{p}_n | \hat{p}_m \rangle \tag{B.17}$$

so we must study the latter matrix of inner products. In doing so it will become clear why we chose the particular $L^2$ measure (B.5). The functions $\hat{p}_n$ are eigenfunctions of the conformal quadratic Casimir operator defined by (A.9),

$$(C_z - \lambda_{\hat{\Delta}_n}) \, \hat{p}_n(z) = 0 \ . \tag{B.18}$$

This $L^2$ measure (B.5) is precisely the one for which $C_z$ is 'almost self-adjoint', i.e. self-adjoint up to boundary terms:

$$\langle f \, | \, C_z g \rangle - \langle C_z f \, | \, g \rangle = -4 \big[ (z-1)^{d/2} z^{1-d/2} \, (f \, \partial_z g - g \, \partial_z f) \big]_1^\infty \ , \tag{B.19}$$

by simply integrating by parts. Taking $(f, g) = (\hat{p}_n, \hat{p}_m)$ and using (B.18) leads to a standard trick for evaluating the desired matrix elements:

$$\langle \hat{p}_n | \hat{p}_m \rangle = \begin{cases} \dfrac{[\psi_1(1 - \frac{\hat{\Delta}_n}{2}) - \psi_1(1 - \frac{d - \hat{\Delta}_n}{2})] \, \sin^2[\frac{\pi}{2}\hat{\Delta}_n]}{\pi^2} \sim 1 + O(\frac{1}{n}) & n = m \ , \\[4mm] \dfrac{\sqrt{\hat{\Delta}_n - d/2}\sqrt{\hat{\Delta}_m - d/2}}{\hat{\Delta}_m + \hat{\Delta}_n - d} \dfrac{4 \, \sin[\frac{\pi}{2}\hat{\Delta}_n] \sin[\frac{\pi}{2}\hat{\Delta}_m]}{\pi^2} & n \neq m \ , \\[1mm] \quad \times \dfrac{[\psi_0(1 - \frac{\hat{\Delta}_n}{2}) + \psi_0(1 - \frac{d - \hat{\Delta}_n}{2}) - \psi_0(1 - \frac{\hat{\Delta}_m}{2}) - \psi_0(1 - \frac{d - \hat{\Delta}_m}{2})]}{\hat{\Delta}_m - \hat{\Delta}_n} & \end{cases} \tag{B.20}$$

where we recall that that $\hat{\Delta}_n = 2\tilde{\alpha} + 2n$, and $\psi_p$ denotes the polygamma function of order $p$.

Let us apply Schur's test (B.16) with $\beta_n = \alpha_n = 1$. Approximating $\sum_m \to n \int_{1/n}^\infty dx$

with $m = xn$, we find

$$\sum_m |\langle \hat{p}_n | \hat{p}_m \rangle| \sim \int_{1/n}^\infty dx \frac{\sqrt{x} \log x}{1 - x^2} \lesssim 1 . \tag{B.21}$$

Hence the test is satisfied and $S$ is bounded.

**Boundedness of $T$.** To derive boundedness of $T$ we must work a bit harder, essentially because the relevant matrix elements $\langle \hat{h}_m | \hat{h}_n \rangle$ are difficult to compute in closed form. Firstly, let us define

$$d_n(z) \equiv (-1)^{n+1} \left[ \hat{h}_n(z) - \hat{p}_n(z) \right] . \tag{B.22}$$

We claim based on many examples that this quantity and its derivative have definite sign:

$$d_n(z) \geq 0 , \qquad \partial_z d_n(z) \geq 0 \qquad \text{for} \quad z > 1 . \tag{B.23}$$

We also claim that, for large $n$ and fixed $z$,

$$d_n(z) \underset{n \to \infty}{\sim} \frac{(-1)^n A_n}{\hat{\Delta}_n(\hat{\Delta}_n - d)\nu_n \mu(z)z^{1+\tilde{\alpha}}} \sim \frac{\sqrt{2}}{\Gamma(\tilde{\alpha})^2} n^{2\tilde{\alpha} - \frac{3}{2}} \frac{1}{\mu(z)z^{1+\tilde{\alpha}}} . \tag{B.24}$$

Indeed, while $\hat{p}_n$ satisfies the homogeneous Casimir equation (B.18), we know $\hat{h}_n$ solves an inhomogeneous Casimir equation quoted in (A.8), which in the present normalisation for $\hat{h}_n$ gives:

$$\left[ C_z - \lambda_{\hat{\Delta}_n} \right] \hat{h}_n(z) = \nu_n^{-1} \frac{A_n}{\mu(z)z^{1+\tilde{\alpha}}} , \tag{B.25}$$

with

$$\nu_n^{-1} A_n \underset{n \to \infty}{\sim} \frac{(-1)^n 4\sqrt{2}}{\Gamma(\tilde{\alpha})^2} n^{2\tilde{\alpha} + \frac{1}{2}} . \tag{B.26}$$

At large $n$ and fixed $z$, one can easily check that $d_n$ as given in (B.24) correctly reproduces the RHS of (B.25), up to corrections that become important when $z \sim n^2$. Such corrections must of course be present, since the exact $\hat{h}_n(z)$ decay as $z^{-\tilde{\alpha}}$ for large $z$, cf. the second asymptotic expression A.18

To estimate the matrix elements $\langle \hat{h}_n | \hat{h}_m \rangle$ for $-\frac{1}{2} < \tilde{\alpha} < \frac{1}{2}$, we will first assume $\tilde{\alpha} > 0$ and then analytically continue to $\tilde{\alpha} < 0$. Using relation (A.8) and integrating by parts, the

off-diagonal matrix elements are given by

$$\langle \hat{h}_n | \hat{h}_m \rangle = \frac{\nu_n^{-1} A_n \int_1^\infty \frac{\mathrm{d}z}{z^{1+\tilde{\alpha}}} \hat{h}_m(z) - \nu_m^{-1} A_m \int_1^\infty \frac{\mathrm{d}z}{z^{1+\tilde{\alpha}}} \hat{h}_n(z)}{(\hat{\Delta}_m - \hat{\Delta}_n)(\hat{\Delta}_m + \hat{\Delta}_n - d)} \ , \qquad n \neq m \ . \tag{B.27}$$

For $\tilde{\alpha} > 0$ there are no boundary terms (unlike (B.19) above). The integrals in the numerator can actually be done exactly:

$$\int_1^\infty \frac{\mathrm{d}z}{z^{1+\tilde{\alpha}}} \hat{h}_n(z) = -\frac{A_n}{4\nu_n} \Gamma(2 - \tfrac{d}{2}) \Gamma(\tilde{\alpha})^2 \Gamma(2\tilde{\alpha}) \tag{B.28}$$

$$\times {}_4\tilde{F}_3 \left(1, 1 - n, 2\tilde{\alpha}, 2\tilde{\alpha} + n + 1 - \tfrac{d}{2}; \tilde{\alpha} + 1, \tilde{\alpha} + 1, 2\tilde{\alpha} + 2 - \tfrac{d}{2}; 1\right) \ .$$

Numerically, we find that for large $n$ this scales as $n^{\frac{1}{2} - 2\tilde{\alpha}}$. We confirm the same scaling by a different argument. First note that:

$$\int_1^\infty \frac{\mathrm{d}z}{z^{1+\tilde{\alpha}}} \hat{h}_n(z) = (-1)^{n+1} \int_1^\infty \frac{\mathrm{d}z}{z^{1+\tilde{\alpha}}} d_n(z) \tag{B.29}$$

since we claim that

$$\int_1^\infty \frac{\mathrm{d}z}{z^{1+\tilde{\alpha}}} \hat{p}_n(z) = 0 \qquad \text{for all } n \geq 1 \ . \tag{B.30}$$

This can be seen by (i) noting that this integral converges for $\tilde{\alpha} > 0$, (ii) deforming the contour to wrap the left cut $(-\infty, 0]$, and (iii) noting that $z^{-1-\tilde{\alpha}} g_{\hat{\Delta}_n}(z) = g_{\hat{\Delta}_n}^{(\tilde{\alpha})}(z)$ has no left cut (see equation (2.35)).[24]

Using the approximation (B.24) for $d_n(z)$ we have

$$\int_1^\infty \frac{\mathrm{d}z}{z^{1+\tilde{\alpha}}} d_n(z) \sim \frac{\sqrt{2}}{\Gamma(\tilde{\alpha})^2} n^{2\tilde{\alpha} - \frac{3}{2}} \int_1^\infty \frac{\mathrm{d}z}{z^{2\tilde{\alpha}}} \left(\tfrac{z-1}{z}\right)^{1-d/2} \tag{B.31}$$

This is divergent in the large $z$ region, but since the approximation breaks down for $z \sim n^2$ we expect that the integral gets regulated at that scale, yielding

$$\int_1^\infty \frac{\mathrm{d}z}{z^{1+\tilde{\alpha}}} d_n(z) \propto n^{2\tilde{\alpha} - \frac{3}{2}} \times (n^{2(1-2\tilde{\alpha})} + \mathcal{O}(1)) = n^{\frac{1}{2} - 2\tilde{\alpha}} + \mathcal{O}(n^{2\tilde{\alpha} - \frac{3}{2}}) \tag{B.32}$$

---

[24]Eq. (B.30) may also be understood as the action of the first GFF functional with $\tilde{\alpha} \to \tilde{\alpha} - 1$ on $\hat{p}_n$. One is able to consider such functionals as long as the integral (B.30) converges. By duality, it must then vanish.

Overall we get

$$\langle \hat{h}_n | \hat{h}_m \rangle \underset{n,m\to\infty}{=} c\,(-1)^{n+m}\sqrt{nm}\,\frac{\left[\left(\frac{n}{m}\right)^{2\tilde{\alpha}} - \left(\frac{m}{n}\right)^{2\tilde{\alpha}}\right]}{m^2 - n^2}\,, \qquad n \neq m \tag{B.33}$$

where $c$ is some numerical constant dependent on $\tilde{\alpha}$.

We must now concern ourselves with the case $n = m$. First let us write

$$\langle \hat{h}_n | \hat{h}_m \rangle = \langle d_n | d_m \rangle - (-1)^n \langle \hat{p}_m | d_n \rangle - (-1)^n \langle \hat{p}_n | d_m \rangle + \langle \hat{p}_n | \hat{p}_m \rangle \tag{B.34}$$

Combining previous results we find that $\langle d_n | d_m \rangle$ is suppressed for large $n, m$. Importantly this cannot change for $n = m$ because of the monotonicity properties (B.23).[25] Therefore we have

$$\langle \hat{h}_n | \hat{h}_n \rangle^d = \langle d_n | d_n \rangle^d - 2(-1)^n \langle \hat{p}_n | d_n \rangle^d + \langle \hat{h}_n | \hat{h}_n \rangle^d = 1 + O(1/n) \tag{B.35}$$

We have checked that both (B.33) and (B.35) hold up to numerical scrutiny. Thus we have

$$\langle \hat{h}_m | \hat{h}_n \rangle = \begin{cases} 1 + \mathcal{O}(\frac{1}{n}) & , \quad n = m \\[2ex] c\,\sqrt{nm}\,\frac{\left[\left(\frac{n}{m}\right)^{2\tilde{\alpha}} - \left(\frac{m}{n}\right)^{2\tilde{\alpha}}\right]}{m^2 - n^2} & , \quad n \neq m \end{cases} \tag{B.36}$$

We now analytically continue this result to negative value $\tilde{\alpha} < 0$ (and remark that it passes numerical checks also in that case).

Let us apply Schur's test with the ansatz $\beta_n = \alpha_n = n^\delta$. The sums $\sum_m |\langle \hat{h}_m | \hat{h}_n \rangle|\, m^\delta$ and $\sum_n |\langle \hat{h}_m | \hat{h}_n \rangle|\, n^\delta$ are finite only when

$$-3/2 + 2|\tilde{\alpha}| < \delta < \tfrac{1}{2} - 2|\tilde{\alpha}|\,. \tag{B.37}$$

Approximating $\sum_m \to n \int_{1/n}^\infty dx$ where $m = xn$, we get

$$\sum_m |\langle \hat{h}_m | \hat{h}_n \rangle|\, m^\delta \sim n^\delta \int_{1/n}^\infty dx\,\frac{x^{1/2+\alpha}(x^{2\tilde{\alpha}} - x^{-2\tilde{\alpha}})}{1 - x^2} \sim n^\delta\,, \tag{B.38}$$

---

[25]This is also consistent with the estimate

$$\langle d_n | d_m \rangle \propto n^{2\tilde{\alpha} - \frac{3}{2}} m^{2\tilde{\alpha} - \frac{3}{2}} \int_1^{nm} \frac{dz}{z^{2\tilde{\alpha}}}\left(\frac{z-1}{z}\right)^{1-d/2} \propto \frac{1}{\sqrt{nm}}(1 + \mathcal{O}(n^{2\tilde{\alpha}-1} m^{2\tilde{\alpha}-1}))\,.$$

and Schur's test is passed (by symmetry between $m, n$):

$$\sum_m |\langle \hat{h}_m | \hat{h}_n \rangle| \, m^\delta \lesssim n^\delta \,, \qquad\qquad \sum_n |\langle \hat{h}_m | \hat{h}_n \rangle| \, n^\delta \lesssim m^\delta \,. \tag{B.39}$$

An appropriate value of $\delta$ satisfying (B.37) can be chosen if and only if $|\tilde{\alpha}| < \frac{1}{2}$.

We conclude that both $T, S$ are bounded and by our previous argument $T$ is invertible, thus establishing that $\{\hat{p}_n\}$, $\{\hat{h}_n\}$ are biorthogonal Riesz bases for $L^2$.

## B.2 Interacting Riesz bases

Now let us turn to general sparse spectra $\Delta_n = 2\tilde{\alpha} + 2n + \gamma_n$ and prove that $p_n \propto \mathcal{I}_z \, g_{\Delta_n}$ form a Riesz basis for $L^2$ as long as $|\tilde{\alpha}| < \frac{1}{2}$.

We recall that the GFF spectrum and corresponding basis are denoted $\hat{\Delta}_n = 2\tilde{\alpha} + 2n$ and $\hat{p}_n$. It suffices to demonstrate that the map,

$$M : \quad \hat{p}_n \mapsto p_n \,, \tag{B.40}$$

defines a bounded, invertible operator from $L^2 \to L^2$, as it may then be composed with $T$ of the previous subsection to obtain a new bounded invertible $MT$ that maps $e_n \mapsto p_n$, thus establishing $p_n$ is Riesz. Below we will study the operator

$$K = 1 - M : \quad \hat{p}_n \mapsto \hat{p}_n - p_n \,. \tag{B.41}$$

We will begin by proving that $K$ is a compact operator and hence bounded, implying that $M$ is also bounded. Then we will prove that $M$ is invertible.

**Compactness of $K$.** We claim that the operator $K$ is *compact* (meaning it is a bounded operator and $\overline{K(U)}$ is compact for any bounded set $U \subset L^2$). The anomalous dimensions have been assumed to decay as a power for large $n$,

$$|\gamma_n| \overset{n \to \infty}{\lesssim} n^{-\varepsilon} \,, \tag{B.42}$$

but they can be large for finite values of $n$. For compactness, it is sufficient to show that $K$ is a limit of finite-rank operators in the operator-norm. Defining the finite-rank operators

$$K_N(p_n^0) = \Theta(n \leq N) \, (\hat{p}_n - p_n) \,, \tag{B.43}$$

(with $\Theta(x)$ the Heaviside function for the predicate $x$), we will show that $\|K_N - K\| \to 0$ as $N \to \infty$.

We will apply Schur's test to the matrix elements of $(K_N - K)^*(K_N - K)$,

$$\Delta K_{mn}^2 := \Theta(m, n > N) \langle p_m - \hat{p}_m | p_n - \hat{p}_n \rangle .$$ 
(B.44)

Since $\gamma_n$ is assumed to decay then $p_n - \hat{p}_n \sim \partial_n \hat{p}_n + \ldots$. Note that here it is understood

$$\partial_n \hat{p}_n \equiv \partial_n \left[ \nu_n^{-1} \mathcal{I}_z g_\Delta \big|_{\Delta = 2\tilde{\alpha} + 2n} \right] .$$ 
(B.45)

Hence

$$\Delta K_{mn}^2 \sim \Theta(m, n > N) \gamma_n \gamma_m \partial_m \partial_n \langle \hat{p}_m | \hat{p}_n \rangle .$$ 
(B.46)

Differentiating the matrix elements (B.20) (which hold even for $m, n$ non-integer) and using the bound (B.42), we find

$$|\Delta K_{mn}^2| \lesssim \Theta(m, n > N) (mn)^{-\varepsilon} \langle \hat{p}_m | \hat{p}_n \rangle$$ 
(B.47)

Schur's test for sequences $m^\delta, n^\delta$ gives

$$\sum_{m > N} |\Delta K_{nm}^2| m^\delta \sim n^{\delta - 2\varepsilon} \int_{N/n}^\infty \mathrm{d}x \frac{x^{\frac{1}{2} - \varepsilon + \delta} \log(x)}{1 - x^2} \lesssim n^\delta N^{-2\varepsilon}$$ 
(B.48)

for e.g. $\delta = \varepsilon$. This implies that $(K_N - K)^*(K_N - K)$ is bounded with norm decaying like $(N^{-2\varepsilon})$, and hence

$$\|K_N - K\| = \sqrt{\|(K_N - K)^*(K_N - K)\|} \lesssim N^{-\varepsilon} \to 0 ,$$ 
(B.49)

as required. Hence $K$ is compact and therefore bounded, and trivially $M = 1 - K$ is also bounded.

**Invertibility of $M$.** To prove that $M$ is invertible we will argue by contradiction. Suppose that $M$ is not invertible, so that it is either not injective or not surjective. Since $K$ is compact then $M$ is what is known as a *Fredholm operator*, which are injective if and only if surjective. Hence $M$ must be neither injective and nor surjective.

In particular, since $M$ is not injective, there is a non-zero element $\sum_n c_n \hat{p}_n$ such that

$$M \left( \sum_n c_n \hat{p}_n \right) = \sum_n c_n p_n = 0 .$$ 
(B.50)

Let us define :

$$\tilde{g}_\Delta := \sum_n \frac{c_n}{\lambda_{\Delta_n} - \lambda_\Delta} p_n \qquad \text{for } \Delta \neq \Delta_n \qquad \left(\lambda_\Delta = \Delta(\Delta - d)\right) . \tag{B.51}$$

This sum clearly converges in $L^2$ since the coefficients are more suppressed than $c_n$. It also converges pointwise on $(1, \infty)$ since

$$\sum_n |c_n|^2 < \infty \implies |c_n| \lesssim n^{-1/2} , \qquad \lambda_n \sim n^2 , \tag{B.52}$$

and $|p_n(z)| < \text{const}$ for fixed $z$ (by the estimate (A.16)). Differentiating term-by-term, we see that (B.51) satisfies the same Casimir equation as $\mathcal{I}_z g_\Delta(z)$:

$$(C_2 - \lambda_\Delta) \, \tilde{g}_\Delta = \sum_n c_n \, p_n = 0 . \tag{B.53}$$

Moreover, both $\mathcal{I}_z g_\Delta(z)$ and $\tilde{g}_\Delta$ are invariant under $\Delta \to d - \Delta$. There is only one such eigenfunction (up to normalisation), so we must have

$$\tilde{g}_\Delta(z) = h_\Delta \mathcal{I}_z \, g_\Delta(z) , \qquad h_\Delta := \sum_n \frac{c_n}{\lambda_{\Delta_n} - \lambda_\Delta} . \tag{B.54}$$

The function $h_\Delta$ is meromorphic in $\Delta$ with a countable set of zeros, which hence have empty interior (i.e. their complement is dense). For $\Delta$ away from those zeros,

$$\mathcal{I}_z \, g_\Delta = h_\Delta^{-1} \, g_\Delta = h_\Delta^{-1} \, M \Big( \sum_n \frac{c_n}{\lambda_{\Delta_n} - \lambda_\Delta} p_n \Big) \in \text{Im}(M) \tag{B.55}$$

Hence we have $\mathcal{I}_z \, g_\Delta \in \overline{\text{Im}(M)}$ for all $\Delta$. In particular all $\hat{p}_n = \mathcal{I}_z \, g_{\hat{\Delta}_n}$ are in $\overline{\text{Im}(M)}$ and since these are a Riesz basis, $\overline{\text{Im}(M)} = L^2$. Since $M$ is Fredholm then its image is closed, and so $\text{Im}(M) = L^2$. Thus $M$ is surjective, but since it is Fredholm, this implies it is also injective, giving the desired contradiction.

# C   Analyticity and bounds on interacting functionals

The interacting functional kernels $h_n$ were defined in Section 3.2.2 as $L^2$ functions. In this appendix, we shall prove that, under our assumptions on the sparse spectrum, they are actually $h_n = i^{(\tilde{\alpha})}(f_n)$ for some $f_n \in \mathcal{A}$. In other words they are real-analytic and sufficiently polynomially bounded near 1 and $\infty$.

Our strategy will be to use the following expansion:

$$h_m = \sum_n \theta_m(\hat{\Delta}_n)\, \hat{h}_n\,, \qquad\qquad \theta_m(\Delta) = \langle h_m \,|\, \mathcal{I}_z g_\Delta \rangle\,. \qquad\qquad \text{(C.1)}$$

This is just the canonical $L^2$-convergent decomposition of the $L^2$ elements $h_m$ with respect to the Riesz basis of free functional kernels $\hat{h}_n$. In this sum the $\hat{h}_n$ are analytic — in fact they are essentially polynomials. The basic problem is that this representation of $h_m$ is too 'coarse': in particular the sum above cannot even be commuted with differentiation. We must therefore produce with a different representation that will make manifest the analyticity of $h_m$. The key will be to replace the sum over integer $n$ with a complex integral.

We will proceed as follows:

- In Section C.1, we will show that the coefficients $\theta_m(\hat{\Delta}_n)$ admit an analytic extension with respect to $n$ that is polynomially bounded in a certain complex domain. This is non-trivial as $\theta_m(\Delta)$ actually grows exponentially for complex $\Delta$.

- In Section C.2, we will use this fact to rewrite the sum in (C.1) as a integral over complex $n$ à la Sommerfeld-Watson. After deforming the contour, this representation establishes analyticity of $h_m$ in a finite domain containing $z \in (1,\infty)$.

- Finally in Sections C.3 and C.4, we will prove that this representation also establishes that the kernels $h_m$ are sufficiently bounded at $z = \infty$ and $z = 1$ so that indeed $h_m = i^{(\tilde{\alpha})}(f_n)$.

## C.1   Functional actions and analyticity

Here we will show that the coefficients $\theta_m(\hat{\Delta}_n)$ in (C.1) admit an analytic extension in $n$ of the form

$$\theta_m(\hat{\Delta}_n) = a(n) + (-1)^n c(n) \qquad \text{for} \quad n > \Lambda \qquad\qquad \text{(C.2)}$$

with $a(n), c(n)$ analytic and polynomially bounded in a 'wedge'-shaped region extending to infinity around the positive-real axis (see Figure 3),

$$W_{\Lambda,\delta} = \left\{ n \in \mathbb{C} \;\middle|\; |\arg(n - \Lambda)| < \delta \right\}\,. \qquad\qquad \text{(C.3)}$$

As we will see in Section C.2, this will be sufficient for establishing analyticity of $h_m$. We will need to use our assumption (3.2) that the sparse spectrum $\{\Delta_n\}$ admits precisely this type of analytic extension on such a wedge. Equivalently, for the Casimir eigenvalues $\lambda_n := \lambda_{\Delta_n}$, by assumption there are two analytic functions $\lambda^{(+)}(n), \lambda^{(-)}(n)$ on the same region such

that

$$\lambda_n = \left(\frac{1 + (-1)^n}{2}\right)\lambda^{(+)}(n) + \left(\frac{1 - (-1)^n}{2}\right)\lambda^{(-)}(n), \qquad n > \Lambda .$$

(C.4)

We note that this is definitely true for the GFF spectrum,

$$\hat{\lambda}_n = \hat{\lambda}(n), \qquad \hat{\lambda}(n) := \hat{\Delta}_n(\hat{\Delta}_n - d) .$$

(C.5)

To prove (C.2), one needs to use the analyticity of the interacting spectrum $\Delta_n$ to learn about the analyticity of the interacting functional actions. The explicit formula (4.19) is the perfect tool to do this: it expresses the interacting functionals actions in terms of the dual interacting spectrum:

$$\theta_m(\Delta) = \prod_{\substack{n=1 \\ n \neq m}}^{\infty} \left(\frac{\lambda_\Delta - \lambda_n}{\lambda_m - \lambda_n}\right) , \qquad \lambda_\Delta = \Delta(\Delta - d), \quad \lambda_n \equiv \lambda_{\Delta_n}$$

(C.6)

Note that validity of this formula does not require that the kernels $h_m$ be analytic.

Furthermore, we will need to use the assumption (3.2) that the anomalous dimensions $\Delta_n - \hat{\Delta}_n$ decay like a power on this wedge. We will write this as

$$
\begin{aligned}
\lambda^{(\pm)}(n) \underset{|n|\to\infty}{\sim}\; & \hat{\lambda}(n) + c^{\pm}\hat{\lambda}(n)^{\frac{1 - \eta_\pm}{2}} + \dots \\
= & 4n^2 + 4(2\tilde{\alpha} - \tfrac{d}{2})n + c^{\pm}(2n)^{1-\eta_\pm} + \dots
\end{aligned}
$$

(C.7)

for constants $c_\pm$ and $\eta_\pm > 0$. These statements imply that there exists $N$ such that for integer $n$,

$$\hat{\lambda}_n < \lambda^{(+)}(n) < \hat{\lambda}_{n+1} \quad \forall n \geq N \qquad \text{or} \qquad \hat{\lambda}_{n-1} < \lambda^{(+)}(n) < \hat{\lambda}_n \quad \forall n \geq N ,$$

(C.8)

and similarly for $\lambda^{(-)}(n)$. That is, the eigenvalues are eventually always above or always below the GFF ones. This property will help us control the overall sign of the functional action.

Let us therefore set $n > N > \max(p, \Lambda)$ and write the functional action in the following form:

$$\theta_p(\hat{\Delta}_n) = \Pi_{\text{poly}}(n)\, \Pi^{(+)}(n)\, \Pi^{(-)}(n) ,$$

(C.9)

with

$$\Pi_{\text{poly}}(n) = \prod_{\substack{m=1 \\ m \neq p}}^{N-1} \left( \frac{\hat{\lambda}_n - \lambda_m}{\lambda_p - \lambda_m} \right) , \qquad \Pi^{(\pm)}(n) = \prod_{\substack{m=N \\ m \text{ even/odd}}}^{\infty} \left( \frac{\hat{\lambda}_n - \lambda^{(\pm)}(m)}{\lambda_p - \lambda^{(\pm)}(m)} \right) . \qquad \text{(C.10)}$$

Since $\Pi_{\text{poly}}(n)$ is polynomial in $n$ we may focus on analysing the infinite products. The analysis is similar for $(\pm)$, so for definiteness we consider here just the $(+)$ case. We write

$$\Pi^{(+)}(n) = S^{(+)}(n) \exp \left\{ \sum_{\substack{m=N \\ m \text{ even}}}^{\infty} \log \left| \frac{\hat{\lambda}_n - \lambda^{(+)}(m)}{\lambda_p - \lambda^{(+)}(m)} \right| \right\} \qquad \text{(C.11)}$$

with $S^{(+)}(n)$ an overall sign to account for the absolute values, as we will discuss below. The argument of the exponential can be written as:

$$-\frac{1}{2} \int_{N-\delta}^{\infty} dm \, \text{Im} \left[ \cot(\tfrac{\pi}{2}m) \right] \, \text{Re} \left[ \log \left( \frac{\hat{\lambda}_n - \lambda^{(+)}(m)}{\lambda_p - \lambda^{(+)}(m)} \right) \right] . \qquad \text{(C.12)}$$

This can be massaged into:

$$-\frac{1}{2} \int_{N-\delta}^{\infty} dm \, \text{Im} \left[ \cot(\tfrac{\pi}{2}m) \log \left( \frac{\hat{\lambda}_n - \lambda^{(+)}(m)}{\lambda_p - \lambda^{(+)}(m)} \right) \right] + \frac{\pi}{2} \int_{N-\delta}^{\hat{n}} dm \, \text{Re} \left[ \cot(\tfrac{\pi}{2}m) \right]$$
$$= -\frac{1}{2} \oint_{<} \frac{dm}{2i} \, \cot(\tfrac{\pi}{2}m) \log \left( \frac{\hat{\lambda}_n - \lambda^{(+)}(m)}{\lambda_p - \lambda^{(+)}(m)} \right) + \log \left( \frac{\sin[\tfrac{\pi}{2}\hat{n}]}{\sin[\tfrac{\pi}{2}(N-\delta)]} \right) . \qquad \text{(C.13)}$$

Here we define $\hat{n}^{(+)}$ implicitly by the equation

$$\lambda^{(+)}(\hat{n}^{(+)}) = \hat{\lambda}_n , \qquad n \text{ even} . \qquad \text{(C.14)}$$

In fact, in deriving (C.13), we had to assume that $\partial_m \lambda^{(+)} > 0$ throughout the integration range. Thanks to (C.7) this may be assured by taking $N$ large enough. It also implies that $n \to \lambda(n)$ is invertible, so (C.14) gives a valid definition of $\hat{n}$. As for the symbol $<$, it means that the integration contour has been deformed to a wedge in the complex $m$ plane with apex at $N - \delta$. That this can be done follows from the assumed analyticity properties of $\lambda_m^{(+)}$.

Performing a similar analysis for $\Pi^{(-)}$ we find:

$$\theta_p(\hat{\Delta}_n) = (-1)^{N+1} S^{(+)}(n) S^{(-)}(n) \, \sin \left[ \tfrac{\pi}{2}\hat{n}^{(+)} \right] \cos \left[ \tfrac{\pi}{2}\hat{n}^{(-)} \right] \frac{\Pi_{\text{poly}}(n) E^{(+)}(n) E^{(-)}(n)}{\tfrac{1}{2} \sin(\pi\delta)} \qquad \text{(C.15)}$$

with

$$E^{(+)}(n) = -\frac{1}{2} \int_< \frac{\mathrm{d}m}{2i} \, \cot(\tfrac{\pi}{2}m) \, \log\left(\frac{\hat{\lambda}(n) - \lambda^{(+)}(m)}{\lambda_p - \lambda^{(+)}(m)}\right)$$

$$E^{(-)}(n) = \frac{1}{2} \int_< \frac{\mathrm{d}m}{2i} \, \tan(\tfrac{\pi}{2}m) \, \log\left(\frac{\hat{\lambda}(n) - \lambda^{(-)}(m)}{\lambda_p - \lambda^{(-)}(m)}\right)$$

(C.16)

Note that we have written $\hat{\lambda}_n \to \hat{\lambda}(n)$. This representation makes clear that both $E^{(\pm)}(n)$ are now also analytically defined in a certain domain in complex $n$ space containing the real line. This domain contains a wedge of slope $\delta$, i.e. the same slope appearing in (C.3) since for large $n$ we have $\hat{\lambda}(n) \sim \lambda^{(\pm)}(n)$.

We will now show that both these factors grow at most logarithmically with $\hat{\lambda}(n)$ (and hence with $n$). To see this we first split the integration region between large $|m|$ and $|m| \sim N$, with the split performed at a scale $M$ such that $\lambda^{\pm}(m)$ is well approximated by its asymptotic form (C.7). Clearly in the small $|m|$ region both integrals are bounded by the logarithm of $\hat{\lambda}(n)$. In the large $|m|$ region we have analytic control over the asymptotics of $\lambda^{\pm}(m)$ and we find that region contributes

$$\sim \frac{1}{4} \int_{-\infty}^{-4M^2} \mathrm{d}\lambda \, \mathrm{Re}\left(\lambda^{-\frac{1}{2}} + c_\pm \lambda^{-\frac{1+\eta}{2}} + \dots\right) \log\left(1 - \frac{\hat{\lambda}(n)}{\lambda}\right)$$

$$= -\frac{1}{4} \sin(\tfrac{\pi}{2}\eta) c_\pm \int_{-\infty}^{-4M^2} \mathrm{d}\lambda \, (-\lambda)^{-\frac{1+\eta}{2}} \, \log\left(1 - \frac{\hat{\lambda}(n)}{\lambda}\right)$$

$$= O(\log[\hat{\lambda}(n)])$$

(C.17)

Thus we conclude that

$$P(n) := \frac{\Pi_{\mathrm{poly}}(n) E^{(+)}(n) E^{(-)}(n)}{\sin(\pi\delta)}$$

(C.18)

is analytic and polynomially bounded in a complex wedge in $n$ space.

Finally, we need to analyse the various prefactors. Define

$$s_\pm = \mathrm{sign}\, c_\pm .$$

(C.19)

We then have

$$(-1)^{N+1} S^{(+)}(n) S^{(-)}(n) = \begin{cases} -s_+, & \text{for } n \text{ even} \\ +s_-, & \text{for } n \text{ odd} \end{cases}$$

(C.20)

As for the remaining factors, note that setting $\delta\hat{n}^{(\pm)} := \hat{n}^{(\pm)} - n$, we have

$$
\sin\left[\tfrac{\pi}{2}\hat{n}^{(+)}\right]\cos\left[\tfrac{\pi}{2}\hat{n}^{(-)}\right] =
\begin{cases}
\sin\left[\tfrac{\pi}{2}\delta\hat{n}^{(+)}\right]\cos\left[\tfrac{\pi}{2}\delta\hat{n}^{(-)}\right] & \text{for } n \text{ even}, \\[2mm]
\cos\left[\tfrac{\pi}{2}\delta\hat{n}^{(+)}\right]\sin\left[\tfrac{\pi}{2}\delta\hat{n}^{(-)}\right] & \text{for } n \text{ odd},
\end{cases}
\tag{C.21}
$$

and for large $n$,

$$
\delta\hat{n}^{\pm} \sim -\frac{c_{\pm}}{n^{1+\eta_{\pm}}} ,
\tag{C.22}
$$

which implies that these prefactors are also analytic and polynomially bounded on a wedge in complex $n$ space, after splitting between even and odd. Hence we conclude:

$$
\theta_p(\hat{\Delta}_n) =
\begin{cases}
-2s_+ \sin\left[\tfrac{\pi}{2}\delta\hat{n}^{(+)}\right]\cos\left[\tfrac{\pi}{2}\delta\hat{n}^{(-)}\right] P(n) & \text{for } n \text{ even}, \\[2mm]
+2s_- \sin\left[\tfrac{\pi}{2}\delta\hat{n}^{(-)}\right]\cos\left[\tfrac{\pi}{2}\delta\hat{n}^{(+)}\right] P(n) & \text{for } n \text{ odd},
\end{cases}
\tag{C.23}
$$

thus establishing the desired result.

## C.2 Analyticity of functional kernels

Each of the interacting functionals can be expanded as an $L^2$-convergent sum of the GFF functionals $\hat{h}_n$ (see eq. (3.28))

$$
h(z) = \sum_n b_n \, \hat{h}_n(z) ,
\tag{C.24}
$$

We showed in Section C.1 that our assumptions on the sparse spectrum $\Delta_n$ imply that these coefficients satisfy

$$
b_n = a(n) + (-1)^n c(n) , \qquad n \in \mathbb{Z}
\tag{C.25}
$$

where $a(n)$ and $c(n)$ are polynomially bounded, analytic functions on some 'wedge' domain (C.3) near infinity. In this section we will show that this means that that $h(z)$ is then real-analytic on $(1, \infty)$. The strategy will be to convert the infinite sum into a contour integral using a Sommerfeld-Watson type trick.

Firstly, let us set

$$
\tilde{b}_n := b_n \times (\hat{\Delta}_n - \tfrac{d}{2}),
\tag{C.26}
$$

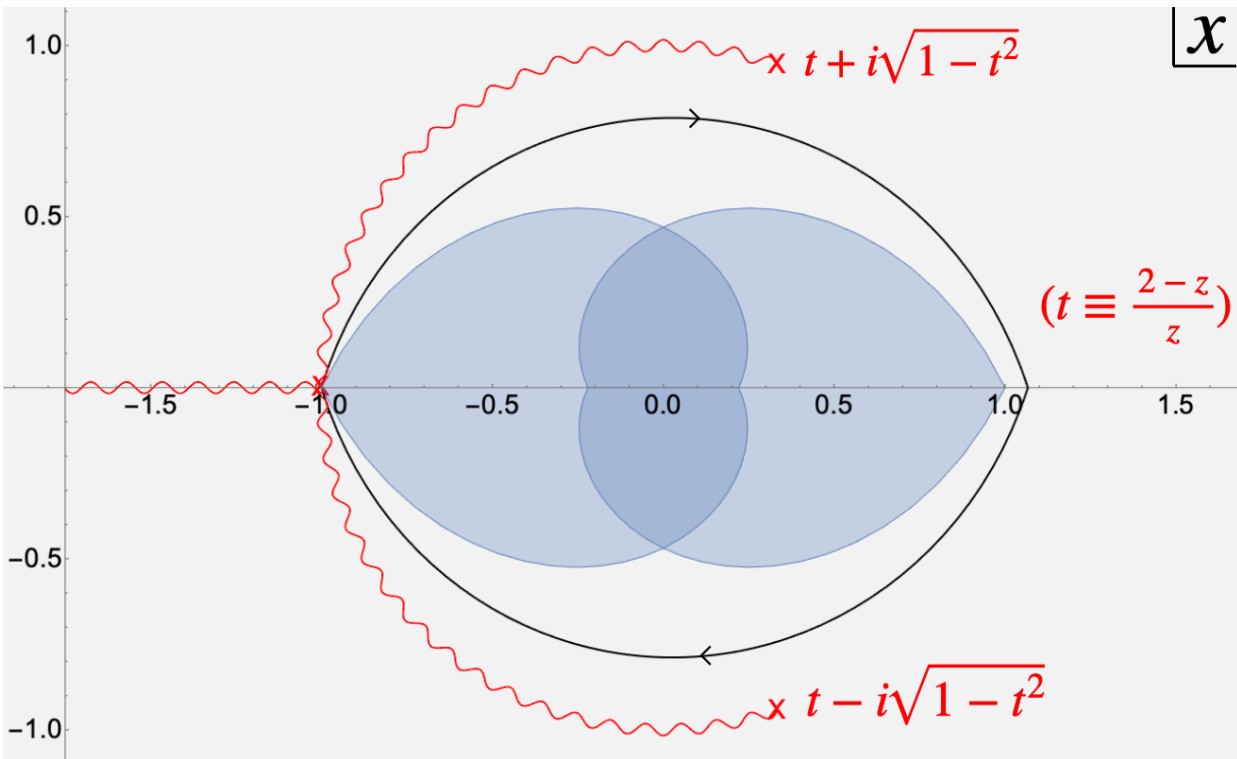

**Figure 7:** The analytic structure of the integral representation (C.30) for interacting functional kernels in the auxiliary $x$-plane. $K(x,z)$ is analytic in $x$ away from the red branch cuts (whose positions depend on $z$, shown here for $z = \frac{3}{2}$). $g(x)$ is analytic outside the 'two-leaf' domain in blue (shown here for $\delta = \frac{\pi}{7}$). The eventual integration contour used in (C.40) is shown here in black.

and introduce a generating function for the $\hat{h}_n$:

$$
\begin{aligned}
K(x,z) &:= \sum_{n=1}^{\infty} x^{n-1} \frac{\hat{h}_n(z)}{\hat{\Delta}_n - \frac{d}{2}} \\
&= z^{-\tilde{\alpha}}\left(\tfrac{z}{z-1}\right)^{\frac{d}{2}-1}(1+x)^{-2-2\tilde{\alpha}+\frac{d}{2}}\,\Gamma(2+2\tilde{\alpha}-\tfrac{d}{2}) \\
&\qquad \times\, {}_3\tilde{F}_2\big(1, 1+\tilde{\alpha}-\tfrac{d}{4}, \tfrac{3}{2}+\tilde{\alpha}-\tfrac{d}{4}; 1+\tilde{\alpha}, 1+\tilde{\alpha}; \tfrac{4x}{(1+x)^2 z}\big) \,.
\end{aligned}
\tag{C.27}
$$

For any $z \in (1,\infty)$, this is an analytic function of $x$ with branch points at

$$
x = -1 \,, \qquad x = t + i\sqrt{1-t^2} \,, \qquad x = t - i\sqrt{1-t^2} \qquad (t \equiv \tfrac{2-z}{z}) \,. \tag{C.28}
$$

The three branch cuts may be chosen to all meet at 1, running along the negative real axis $(-\infty, -1)$ and along the unit circle from 1 to $t + i\sqrt{1-t^2}$ and to $t - i\sqrt{1-t^2}$ (in red in Figure 7). If any $z \in (1,\infty)$ gains a small enough imaginary part, the branch points $t \pm i\sqrt{1-t^2}$ move slightly off the unit circle and $K(x,z)$ remains an analytic function of $x$ away from the cuts.

We may now write the expansion (C.24) as

$$h(z) = \sum_{n=1}^{\infty} \tilde{b}_n \oint_0 \frac{dx}{2\pi i} \frac{1}{x^n} K(x, z) = \sum_{n=1}^{\infty} \tilde{b}_n \oint_\Gamma \frac{dx}{2\pi i} \frac{1}{x^n} K(x, z) \ . \tag{C.29}$$

In the first expression the contour tightly encloses $x = 0$, while in the second it is deformed towards the edge of the unit disk. We claim that we can exchange the order of summation and integration in (C.29) to obtain

$$h(z) = \oint_\Gamma \frac{dx}{2\pi i} g(x) K(x, z) \ , \qquad g(x) := \sum_{n=1}^{\infty} \tilde{b}_n x^{-n} \ . \tag{C.30}$$

First, we shall establish the domain of analyticity of $g$. Clearly this series converges absolutely to an analytic function for $|x| > 1$, since $\tilde{b}_n$ is polynomially bounded. We will show that it admits an analytic extension to a strictly larger region: the non-analyticities are confined to a smaller 'two-leaf' region (in blue in Figure 7), which only touches the unit disc at $\pm 1$. It is sufficient to treat the tail of the sum, for which the analytic expression (C.25) for $\tilde{b}_n$ applies:

$$\tilde{g}(x) := g(x) - \sum_{n \leq \Lambda} \tilde{b}_n x^{-n} = \sum_{n > \Lambda} \left( a(n) + (-1)^n c(n) \right) x^{-n} \ . \tag{C.31}$$

Following the trick of Sommerfeld and Watson, we may recast the sum over $n$ as an integral:

$$\tilde{g}(x) = \oint_{\Gamma_{[\Lambda, \infty)}} \frac{dn}{2\pi i} \frac{a(n)(-x)^{-n} + c(n) x^{-n}}{\sin \pi n} \ . \tag{C.32}$$

To begin with, the integration contour wraps the segment $n > \Lambda$ of the real axis, but we will now deform it away from the real axis. To check that the integral will converge on such a contour, let us estimate the integrand for large complex $n$ with $\text{Re}(n) > 0$: focusing first on the second term in (C.32), we note that

$$\left| \frac{x^{-n}}{\sin \pi n} \right| \overset{|n| \to \infty}{\sim} \exp \left( -\pi |\text{Im}(n)| - \arg(x) \, \text{Im}(n) - \log |x| \, \text{Re}(n) \right) \tag{C.33}$$

For $|x| > 1$, the final term in the exponent is negative, and the first two terms combine to be non-positive. The same analysis applies (with $x \to -x$) to the other term. Hence the integrand of (C.32) is exponentially suppressed for $n$ large with $\text{Re}(n) > 0$. One may then deform the contour to the wedge-shaped one (denoted $<$) inside the domain of analyticity $W_{\Lambda, \delta}$ for $a(n), c(n)$ (see Figure 8)

$$\tilde{g}(x) = \oint_< \frac{dn}{2\pi i} \frac{a(n)(-x)^n + c(n) x^{-n}}{\sin \pi n} \ . \tag{C.34}$$

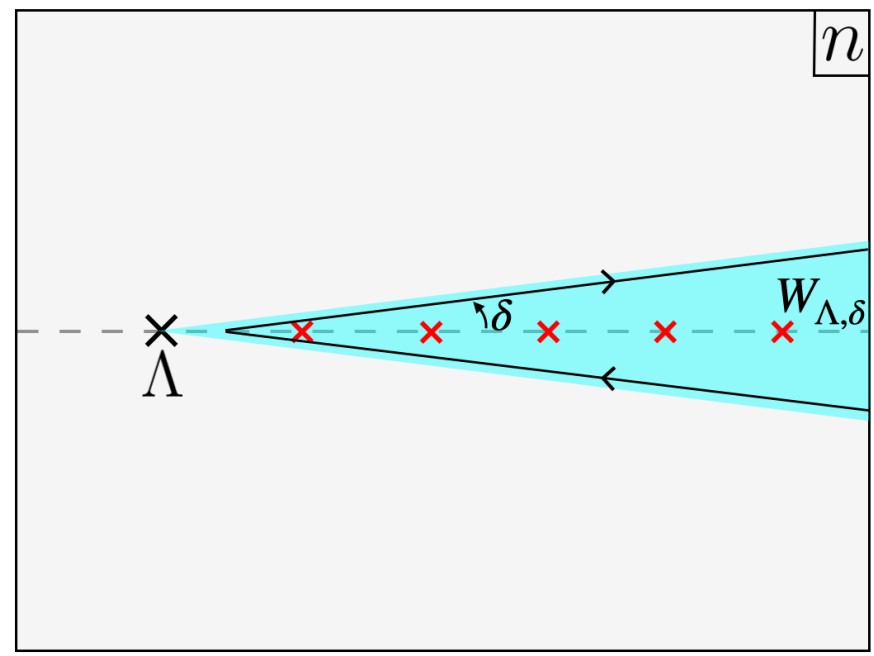

**Figure 8:** The integration contour denoted $<$ in the $n$-plane in the expression (C.34). The domain of analyticity $W_{\Lambda,\delta}$ of the integrand is shown in blue, with its poles shown in red.

This expression is now manifestly analytic for a larger region in $x$: using the same estimate (C.33) with $\arg n \sim \delta$, the second term of the integrand is exponentially suppressed outside of a leaf-shaped region,

$$|x| < e^{-\tan \delta \, (\pi - |\arg x|)} \, , \qquad |\arg(x)| \leq \pi \, . \tag{C.35}$$

Replacing $x \to -x$, we see that the first term of the integrand is exponentially suppressed outside a mirrored leaf-shaped region, and the whole integrand is exponentially suppressed outside of a two-leaf region (the blue region in Figure 7),

$$R_{\text{two-leaf}} := \left\{ |x| < e^{-\tan \delta \, |\arg x|} \right\} \cup \left\{ |x| < e^{-\tan \delta \, (\pi - |\arg x|)} \right\} \, . \tag{C.36}$$

(Note that, if the angle $\delta$ of the wedge on which $a(n), c(n)$ are analytic tends to zero, the two-leaf region becomes the unit disc.) Hence it follows that the integral (C.34) converges locally uniformly over $x \in \mathbb{C} \setminus R_{\text{two-leaf}}$, so $\tilde{g}(x)$, and hence $g(x)$, is analytic on this region.

Now let us return to the question of interchanging the sum and integral in (C.29) to obtain (C.30). The integration contour $\Gamma$ is pinched at $x = -1$ between the branch cut singularities of $K(x, z)$ and the singularities of $g$ in $R_{\text{two-leaf}}$, as shown in Figure 7. We will prove at the same time that the integral in (C.30) converges, and that the sum and integral in (C.29) swap to give (C.30). First, these statements are clearly true away from $x = -1$ since the integrand is analytic and the sum converges locally uniformly.

Near the problematic point $x = -1$, we shall argue by dominated convergence. One can

check that

$$K(x,z) \underset{x \to -1}{\sim} \underbrace{O(1)}_{\text{analytic}} + \underbrace{O\left(|x+1|^{-2\tilde{\alpha}+d/2}\right)}_{\text{non-analytic}} . \tag{C.37}$$

Meanwhile, using the estimate (5.21) to get the leading behaviour

$$\tilde{b}_n \sim -\left(\hat{\Delta}_n - \frac{d}{2}\right) \gamma_n \hat{\theta}'(\hat{\Delta}_n) = O(n^{d/2-2-2\tilde{\alpha}-\epsilon}) , \tag{C.38}$$

one finds as $x$ approaches the unit circle that

$$g(x) = \sum_{n=1}^{\infty} \tilde{b}_n x^{-n} \underset{|x| \to 1}{\sim} \underbrace{O(1)}_{\text{analytic}} + \underbrace{O\left[(|x|-1)^{1-d/2+2\tilde{\alpha}+\epsilon}\right]}_{\text{non-analytic}} . \tag{C.39}$$

For terms in the integral (C.30) involving the analytic part of either $K$ or $g$, the contour can be deformed respectively to the left or the right and we are done. The term involving only the non-analytic pieces of $K$ and $g$, which really has to pass through $x = -1$, is dominated by the integrable function $|x+1|$.

We conclude that $h(z)$ may represented as

$$h(z) = \oint_{\Gamma_{R_{\text{two-leaf}}}} \frac{\mathrm{d}x}{2\pi i} g(x) K(x,z) . \tag{C.40}$$

with the contour wrapping the region $R_{\text{two-leaf}}$ as in Figure 7.

Note now that any $z \in (1, \infty)$ has a complex neighbourhood over which the integrand is analytic in $z$ and the integral over $x$ converges uniformly. The latter is true because the cuts along the unit circle exit the point $-1$ with arguments $\pm\pi - \frac{1}{2} \arg z$ (for example, see Figure 9a). For any $\delta > 0$, the two-leaf region departs from $x = -1$ at an angle strictly bounded away from the unit circle. Hence $z$ can always be given a finite imaginary part without the cuts and the two-leaf region overlapping, meaning the integration contour does not get pinched anywhere other than $x = -1$.

Hence $h(z)$ is real-analytic on $(1, \infty)$. Note that the radius of convergence of $h(z)$ does generically shrink to zero at the endpoints $z = 1$ and $z = \infty$, with the branch points $x = t \pm \sqrt{1 - t^2}$ colliding and pinching the integration contour at $x = 1$ and $x = -1$ respectively (see Figure 9). As a result, we typically expect the interacting functionals $h(z)$ to have a branch cut running along $(-\infty, 1)$. In the next sections we will verify that $h(z)$ satisfies the correct properties at $z = 1$ and $z = \infty$, namely appropriate polynomial boundedness, so that it lies in the image of $\mathcal{A}$ inside $L^2$.

## C.3 Polynomial bound at $z = \infty$

Using the integral representation (C.40) for an interacting kernel $h = h_n$, we shall prove the polynomial bound

$$|h_n(z)| \underset{z \to \infty}{\lesssim} |z|^{-\tilde{\alpha}} . \tag{C.41}$$

This is the limit when the branch points at $x = t \pm i\sqrt{1 - t^2}$ pinch the contour at $x = -1$, as shown in Figure 9a. First, we note that the parts of the contour away from $x = -1$ indeed scale as $K(x, z) \sim z^{-\tilde{\alpha}}$ in this limit. The same applies for the terms involving the analytic part of $g(x)$ in (C.39) or the analytic part of $K(x, z)$ in (C.37), for which the contour can be deformed away from $x = -1$.

Turning to the part of the contour near $x = -1$ involving only the non-analytic parts, we find the bound

$$\left| \int_{x \sim -1} \mathrm{d}x \, g(x) \, K(x, z) \right| \underset{z \to \infty}{\supset} |z|^{-\tilde{\alpha}} \int_{x \sim -1} |\mathrm{d}x| \, |x + 1|^{-1+\epsilon} \left| {}_3\tilde{F}_2(y/z) \right| , \quad y \equiv \frac{4x}{(1 + x)^2} , \tag{C.42}$$

where ${}_3\tilde{F}_2$ function is the one appearing in (C.27). Changing the integration variable $x \to y$, properly accounting for the Jacobian factor, and expanding at large $y$, one obtains the bound

$$\left| \int_{x \sim -1} \mathrm{d}x \, g(x) \, K(x, z) \right| \underset{z \to \infty}{\supset} |z|^{-\tilde{\alpha}} \int_z^\infty \mathrm{d}y \, |y|^{-1-\epsilon/2} \left| {}_3\tilde{F}_2(y/z) \right| \sim |z|^{-\tilde{\alpha} - \frac{\epsilon}{2}} , \tag{C.43}$$

giving the desired bound (C.41).

## C.4 Polynomial bound at $z = 1$

Finally, let us study the behaviour at $z = 1$. In order for the interacting kernel $h \in L^2$ to belong to the subspace $i^{(\tilde{\alpha})}(\mathcal{A})$ the definition (2.17) requires a bound on its behaviour as $z \to 1$:

$$(z - 1)^{d/2-1} \, h_n(z) \underset{z \to 1}{\sim} (\text{analytic}) + O(|z - 1|^{\tilde{\alpha}}) . \tag{C.44}$$

We will prove this using the integral representation (C.40). As $z \to 1$, the cuts along the unit circle pinch the contour at $x = 1$ as shown in Figure 9b. The parts of the contour away from $x = 1$ will lead to purely analytic contributions to $(z - 1)^{d/2-1} \, h(z)$ (with the prefactor coming from (C.27)). The same applies for the analytic part of $g(x)$ in (C.39), for which the integration contour can be deformed inwards, away from $x = 1$.

The non-analytic terms that we need to bound in (C.44) correspond to the part of the contour passing through $x = 1$ containing only the non-analytic part of $g(x)$. This

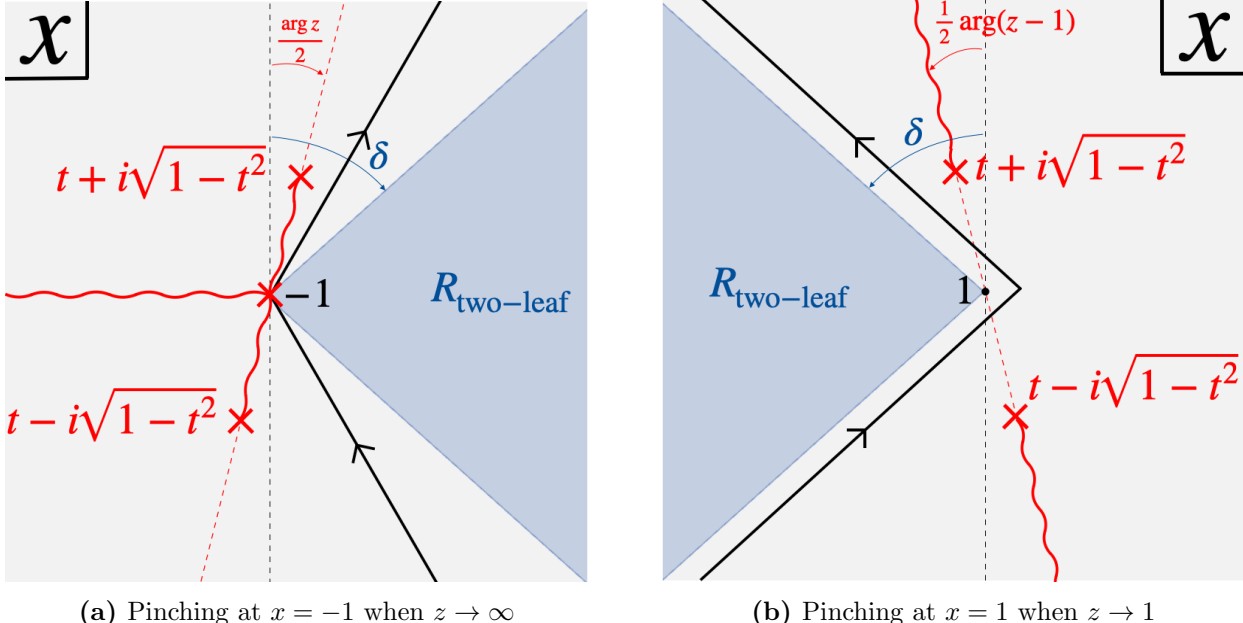

**(a)** Pinching at $x = -1$ when $z \to \infty$      **(b)** Pinching at $x = 1$ when $z \to 1$

**Figure 9:** Close-ups of the branch cuts (red) pinching the integration contour (black) against the 'two-leaf' region (blue) at $x = -1$ and $x = 1$ in the respective limits $z \to \infty$ and $z \to 1$. Here, as above, we denote $t \equiv \frac{2-z}{z}$.

contribution is bounded by

$$\left| (z-1)^{d/2-1} \int_{x \sim 1} \mathrm{d}x \, g(x) \, K(x,z) \right| \underset{z \to 1}{\lesssim} \int_{|x-1| \sim |z-1|^{1/2}} |\mathrm{d}x| \, |x-1|^{1-d/2+2\tilde{\alpha}+\epsilon} \left| {}_3\tilde{F}_2(y/z) \right| , \quad (\text{C.45})$$

where $y$ and ${}_3\tilde{F}_2(y)$ are defined as in (C.42). Again changing integration variable $x \to y$ and including the relevant Jacobian, we obtain the bound

$$\left| (z-1)^{d/2-1} \int_{x \sim 1} \mathrm{d}x \, g(x) \, K(x,z) \right| \underset{z \to 1}{\lesssim} \int_{|y-1| \sim |z-1|} |\mathrm{d}y| \, |y-1|^{-d/4+\tilde{\alpha}+\epsilon/2} \left( 1 + |y-1|^{(d-3)/2} \right)$$

$$\lesssim |z-1|^{\tilde{\alpha}+(4-d)/4+\epsilon/2} + |z-1|^{\tilde{\alpha}+(d-2)/4+\epsilon/2} \quad (\text{C.46})$$

Since $d \leq 4$ then the first term is bounded by $|z-1|^{\tilde{\alpha}}$ as required. The same applies for the second term as long as $d \geq 2$. For the remaining case $d = 1$, it is necessary to make a slightly stronger boundedness assumption on the sparse spectrum $\Delta_n = 2\tilde{\alpha} + 2n + \gamma_n$:

$$\gamma_n = \alpha(n) + (-1)^n \beta(n) \qquad \text{with} \qquad \alpha(n) \sim n^{-\epsilon} , \qquad \beta(n) \sim n^{-\frac{1}{2}-\epsilon} ,$$

instead of $\beta(n) \sim n^{-\epsilon}$ in (3.2).

# D  On general values of $\tilde{\alpha}$ and subtractions

In Section 3, we used an intermediate $L^2$ space to generate complete Schauder bases of analytic functionals $f_n \in \mathcal{A}$, dual to particular sets of blocks $g_{\Delta_n}$. We stated there that such a basis exists dual to any sparse set, $\Delta_n = 2\tilde{\alpha} + 2n + \gamma_n$, under assumptions of large-$n$ analyticity and power-law decay of $\gamma_n$. We gave an argument for this fact in the main text in the special case $\tilde{\alpha} \in (-\frac{1}{2}, \frac{1}{2})$, and this appendix will treat the case of general values of $\tilde{\alpha}$.

## D.1  Almost all values of $\tilde{\alpha}$

First let us consider the case $\tilde{\alpha} \notin \frac{1}{2} + \mathbb{N}$, i.e. all values related to $(-\frac{1}{2}, \frac{1}{2})$ by an integer shift (these are almost all values of $\tilde{\alpha}$).

In Section 3.3, we explained the extra subtleties that arise when $\tilde{\alpha} > \frac{1}{2}$. Riesz bases for $L^2$ can still be used to produce Schauder bases for $\mathcal{A}$ and $\mathcal{HF}$, but the embedding of $L^2$ becomes more non-trivial, and the bases are subject to a finite subtraction procedure reviewed in Section 3.3. There, we derived a set of sufficient conditions (a)–(c) in $L^2$ language to produce a Schauder basis $f_n \in \mathcal{A}$. In this section, we will show that these conditions are satisfied by any sparse spectrum satisfying our assumptions.

**(a) The set of discontinuities of blocks $\{p_n = \mathcal{I}_z\, g_{\Delta_n}\}$ can be completed to a Riesz basis by adding finitely many blocks.**
As explained in eq. (3.32), if $\tilde{\alpha} = \beta + M$ for $\beta \in (-\frac{1}{2}, \frac{1}{2})$ and $M \in \mathbb{N}$, then one can just add $M$ arbitrary blocks to the sparse spectrum $\Delta_n = \tilde{\alpha} + 2n + \gamma_n$ to obtain an extended spectrum $\Delta_n^\beta \sim 2\beta + 2n + \gamma_n$. Then the analysis of Section 3.2 applies, so the corresponding blocks $\{p_n = \mathcal{I}_z\, g_{\Delta_n^\beta}\}$ form a Riesz basis.

**(b) Finite combinations of $M$ of the dual Riesz basis $\{h_n\}$ take the form $i^{(\tilde{\alpha})}(f_n)$ for some $f_n \in \mathcal{A}$.**
We already know from Appendix C that each $h_n$ is analytic on $(1, \infty)$ and satisfies the same bounds (2.17) as the GFF functionals at $z = 1$ and $z = \infty$. At $z = 1$, this is immediately sufficient. At $z = \infty$, we encounter the need for additional subtractions for $\tilde{\alpha} > \frac{1}{2}$ — even for the GFF functionals — since $h_n$ behave as a $z^{-\beta}$, which is $M$ powers less suppressed than the desired functionals $i^{(\tilde{\alpha})}(f_n) \sim z^{-\alpha}$.

Therefore one should take the combinations of $M$ of the $h_n$ that are sufficiently bounded at $z = \infty$. More precisely, the combinations of interacting functionals $h_n$ that are in the

image of $i^{(\tilde\alpha)}$ are:

$$h_n - \sum_{k=1}^{M} c_{nk}^{\tilde\alpha}\, h_k \quad \in i^{(\tilde\alpha)}(\mathcal{A})\,, \qquad n > M \tag{D.1}$$

$$c_{nk}^{\tilde\alpha} := \left( \prod_{\substack{l\neq k \\ 0\leq l\leq M}} \frac{\lambda_{\Delta_n^\beta} - \lambda_{\Delta_l^\beta}}{\lambda_{\Delta_k^\beta} - \lambda_{\Delta_l^\beta}} \right) \frac{\theta_n(\Delta_0^\beta)}{\theta_k(\Delta_0^\beta)}\,, \tag{D.2}$$

where $\theta_n(\Delta) := \langle h_n | \mathcal{I}_z\, g_\Delta \rangle$. This expression appears to depend on another (arbitrary) variable $\Delta_0^\beta$, but in fact it is independent of all the arbitrary parameters $\Delta_0^\beta, \ldots, \Delta_M^\beta$. This is because any dependence on these arbitrary parameters would lead to multiple subtracted functionals (D.1) dual to the same subtracted blocks $g_{\Delta_n}^{\tilde\alpha}$, which in turn form a Schauder basis of $\mathcal{HF}$. Hence the difference between them would be orthogonal to all basis elements, so must vanish.

To see that the combination (D.1) is suppressed by $z^{-M}$ at $\infty$ relative to $h_n$, let us consider its action against blocks. Using the identity (D.15) derived below, one finds

$$\langle h_n - \sum_{k=1}^{M} c_{nk}^{\tilde\alpha}\, h_k \,|\, \mathcal{I}_z\, g_\Delta \rangle = \left( \prod_{1\leq k\leq M} \frac{\lambda_{\Delta_n^\beta} - \lambda_{\Delta_k^\beta}}{\lambda_\Delta - \lambda_{\Delta_k^\beta}} \right) \theta_n(\Delta)\,, \qquad n > M\,, \tag{D.3}$$

i.e. the functional action defined by this combination is suppressed by $(\lambda_\Delta)^{-M} \sim \Delta^{-2M}$ relative to $\theta_n(\Delta)$. The large $\Delta$ behaviour of the functional action is tied to the large $z$ behaviour of the kernel with the identification $z \sim \Delta^2$, signaling the desired $z^{-M}$ suppression.

### (c) Decomposition of Cauchy kernel.

As explained in Section 3.3, the loss of $M$ functionals is related by duality to the appearance of $M$ relations (3.36) between the subtracted blocks in the space of hyperfunctions. Let us explain the origin of these relations.

We know that the blocks $p_n = \mathcal{I}_z\, g_{\Delta_n^\beta}$ form a Riesz basis for $L^2$ so, for example, any block can be uniquely decomposed into them:

$$\mathcal{I}_z\, g_\Delta = \sum_{n=1}^{\infty} \theta_n(\Delta)\, \mathcal{I}_z\, g_{\Delta_n^\beta} \in L^2\,. \tag{D.4}$$

Since the embedding $j^{(\tilde\alpha)} : L^2 \to \mathcal{HF}$ is continuous, then we obtain a convergent decomposition in $\mathcal{HF}$:

$$g_\Delta^{(\tilde\alpha)} = \sum_{n=1}^{\infty} \theta_n(\Delta)\, g_{\Delta_n^\beta}^{(\tilde\alpha)} \in \mathcal{HF}\,. \tag{D.5}$$

However, we claim that, when $\tilde{\alpha} > \frac{1}{2}$, the blocks on the RHS are **not independent** as elements of $\mathcal{HF}$. Rather, there are relations among them — morally because they have been 'oversubtracted' by $\tilde{\alpha}$, which is $M$ units greater than $\beta$. (In other words, for $\tilde{\alpha} > \frac{1}{2}$, the map $j^{(\tilde{\alpha})}$ does not have a continuous inverse.)

One can eliminate $M$ blocks by applying the Casimir operator to the decomposition (D.5). We define the 'subtracted Casimir', acting on hyperfunctions, by

$$C_z^{(\tilde{\alpha})} H := z^{-\tilde{\alpha}-1} C_z(z^{\tilde{\alpha}+1} H) , \qquad C_z := 4(1-z)^{1-d/2} z^{1+d/2} \partial_z[(1-z)^{d/2} z^{1-d/2} \partial_z] , \quad \text{(D.6)}$$

defined equivalently by either acting on the analytic function $H$ away from the cuts, or on its discontinuity $\mathcal{I}_z H$ itself. Note that in general the result $C_z^{(\tilde{\alpha})} H$ may not be a member of the same space $\mathcal{HF}$ since its behaviour at $\infty$ may be worse, but this is not a concern for individual blocks since they are eigenfunctions of the subtracted Casimir:

$$\left(C_z^{(\tilde{\alpha})} - \lambda_\Delta\right) g_\Delta^{(\tilde{\alpha})} = 0 . \tag{D.7}$$

One now needs to analyse whether the action of the Casimir preserves the convergence of the sum (D.5) in $\mathcal{HF}$. In Section D.2.1 below, we show that it can be applied precisely $M$ times. Hence, denoting $\Delta \equiv \Delta_0$ in (D.5) and applying the combination $\prod_{\substack{l \neq k \\ 0 \leq l \leq M}} (C_z^{(\tilde{\alpha})} - \lambda_{\Delta_l^\beta})$ for $0 < k \leq M$, we obtain the desired relations expressing the lowest $M$ blocks in terms of the higher ones:

$$g_{\Delta_k^\beta}^{(\tilde{\alpha})} = -\sum_{n > M} c_{nk}^{\tilde{\alpha}} \, g_{\Delta_n^\beta}^{(\tilde{\alpha})} , \qquad k = 1, \ldots, M , \tag{D.8}$$

where $c_{nk}^{\tilde{\alpha}}$ are the same coefficients defined in (D.2).

As a result, we may eliminate all of the 'additional' blocks and, as argued in Section 3.3, one obtains the following decomposition of the Cauchy kernel into the physical blocks $\Delta_n$:

$$\frac{1}{z-w} = \sum_{n=1}^{\infty} g_{\Delta_n}^{(\tilde{\alpha})}(w) \, f_n(z) , \tag{D.9}$$

converging as a hyperfunction in $w$. We further need to prove that this expression converges as an analytic function of $z$. This essentially follows from the fact the dimensions $\Delta_n$ decay to the GFF ones at large $n$. One may argue similarly to (3.30),(3.31) in the unsubtracted case that this sum converges in the same way as in the GFF case:

$$g_{\Delta_n}^{(\tilde{\alpha})}(w) \, f_n(z) \sim g_{\hat{\Delta}_n}^{(\tilde{\alpha})}(w) \, \hat{f}_n(z) \left(1 + O(n^{-\epsilon})\right) . \tag{D.10}$$

The subtracted GFF functionals $\hat{f}_n$ coincide with the analytic continuation of the GFF functionals (A.7) to $\tilde{\alpha} > \frac{1}{2}$ (see discussion in Appendix D.3 below). Hence they satisfy the estimate (2.33), so the sum (D.10) converges absolutely, as required, for $z$ in a neighbourhood

of the cut $[1, \infty]$ and $w$ away from the cut.

## D.2 An identity for interacting functionals

### D.2.1 How many times can the Casimir be applied?

We will derive an identity for interacting functional actions by applying the Casimir to the infinite sum (D.5). First we need understand whether it will preserve the sum's convergence, and how many times it can be applied.

To begin with, we note that the sum (D.4) in $L^2$ converges with the bounds $O(\log z)$ and $[O((z-1)^{1-d/2}) + O(1)]$ at the respective endpoints. Hence it follows that the sum (D.5) in $\mathcal{HF}$ converges with behaviour at least as good as $O(z^{-\tilde{\alpha}-1-\epsilon})$ and $[O((z-1)^{1-d/2}) + O(1)]$ (in the sense discussed in footnote 8).

Applying the subtracted Casimir (D.6) to hyperfunctions worsens their $z \to \infty$ and $z \to 1$ behaviour by one power, so applying it $P$ times will give a sum converging with behaviour at least as good as $O(z^{P-\tilde{\alpha}-1-\epsilon})$ and $[O((z-1)^{-P+1-d/2}) + O((z-1)^{-P})]$. In order to have convergence in the space $\mathcal{HF}$, we require that these exponents satisfy:

$$P - \tilde{\alpha} - 1 - \epsilon < 0 \qquad \text{and} \qquad \max\left(P, P-1+\tfrac{d}{2}\right) < \tilde{\alpha}+1 \ . \tag{D.11}$$

In the worse case $d = 3$, this requirement becomes

$$P < \tilde{\alpha} + \tfrac{1}{2} \ . \tag{D.12}$$

By definition, the largest integer less than $\tilde{\alpha} + \frac{1}{2}$ is $M$. Hence the Casimir can be applied to (D.5) at most $M$ times.

### D.2.2 Deriving the identity

Let us derive an identity for the interacting functionals, which shows that the subtracted functionals (D.1) have an action (D.3) on blocks that is more suppressed for large $\Delta$.

On one hand, setting $\Delta \equiv \Delta_0$ and applying the combination $\prod_{\substack{l \neq k \\ 0 \leq l \leq M}} (C_z^{(\tilde{\alpha})} - \lambda_{\Delta_l^\beta})$ to (D.5) leads to the relation (3.36) between Riesz basis elements. Substituting this back into (D.5) gives the Schauder basis decomposition in $\mathcal{HF}$,

$$g_\Delta^{(\tilde{\alpha})} = \sum_{n>M} \left[ \theta_n(\Delta) - \sum_{0<k\leq M} \Big[ \prod_{\substack{l \neq k \\ 0 \leq l \leq M}} \frac{\lambda_{\Delta_n^\beta} - \lambda_{\Delta_l^\beta}}{\lambda_{\Delta_k^\beta} - \lambda_{\Delta_l^\beta}} \Big] \frac{\theta_n(\Delta_0^\beta)}{\theta_k(\Delta_0^\beta)} \theta_k(\Delta) \right] g_{\Delta_n^\beta}^{(\tilde{\alpha})} \tag{D.13}$$

On the other hand, applying the combination $\prod_{1 \leq l \leq M}(C_z^{(\tilde{\alpha})} - \lambda_{\Delta_l^\beta})$ and setting $\Delta = \Delta_k$

gives another decomposition of $g_\Delta^{(\tilde\alpha)}$ into the same basis,

$$g_\Delta^{(\tilde\alpha)} = \sum_{n>M} \Big[ \prod_{1\le k\le M} \frac{\lambda_{\Delta_n^\beta} - \lambda_{\Delta_k^\beta}}{\lambda_\Delta - \lambda_{\Delta_k^\beta}} \Big] \theta_n(\Delta)\, g_{\Delta_n^\beta}^{(\tilde\alpha)} \tag{D.14}$$

Equating the two decompositions (D.13) and (D.14) and noting, e.g. by applying the functionals, that their coefficients must coincide, we obtain the non-trivial identity (for $n > M$):

$$\theta_n(\Delta) - \sum_{\substack{0<k\le M}} \Big[ \prod_{\substack{l\neq k \\ 0\le l\le M}} \frac{\lambda_{\Delta_n^\beta} - \lambda_{\Delta_l^\beta}}{\lambda_{\Delta_k^\beta} - \lambda_{\Delta_l^\beta}} \Big] \frac{\theta_n(\Delta_0^\beta)}{\theta_k(\Delta_0^\beta)} \theta_k(\Delta) = \Big[ \prod_{0<k\le M} \frac{\lambda_{\Delta_n^\beta} - \lambda_{\Delta_k^\beta}}{\lambda_\Delta - \lambda_{\Delta_k^\beta}} \Big] \theta_n(\Delta)\, . \tag{D.15}$$

One can check that the explicit product formula (4.19) for $\theta_n(\Delta)$ indeed solves this identity: this provides a non-trivial cross-check of that formula and the results of Section 4.

## D.3  All values of $\tilde\alpha$

For all values $\tilde\alpha \notin \frac{1}{2} + \mathbb{Z}$, we have constructed complete Schauder bases of functionals $\{f_n^{\tilde\alpha}\}$ for the spaces $\mathcal{A}^{\tilde\alpha}$, dual to the spectra $\Delta_n^{\tilde\alpha} := 2\tilde\alpha + 2n + \gamma_n$. (In this section, we shall indicate explicitly the dependence on $\tilde\alpha$.)

Let us now define functionals for the final case $\tilde\alpha \in \frac{1}{2} + \mathbb{Z}$ by simply taking a pointwise limit of the functional kernels we have already constructed:

$$f_n^{1/2+M} := \lim_{\tilde\alpha\to(1/2+M)^+} f_n^{\tilde\alpha}\, , \qquad M \in \mathbb{Z}\, , \tag{D.16}$$

In the rest of this section, we will argue that (D.16) leads to well-defined functionals $f_n^{1/2+M}$ that are: (i) elements of $\mathcal{A}^{1/2+M}$, (ii) complete in $\mathcal{A}^{1/2+M}$ and (iii) dual to the spectrum $\Delta_n^{1/2+M}$.

For clarity of presentation, we will set $M = 0$ and treat here the $\tilde\alpha = \frac{1}{2}$ case, but the others work in the same way.

**Elements of $\mathcal{A}^{1/2}$.**  First, it is easy to argue that $f_n^{1/2}(z)$ satisfy the appropriate bounds at the endpoints $z = \infty$ and $z = 1$ by starting from the polynomial bounds (C.41) and (C.44) derived above for $\tilde\alpha \notin \frac{1}{2} + \mathbb{Z}$. The subleading terms depend on $\tilde\alpha$ in a way that is uniform as $\tilde\alpha \to \frac{1}{2}^+$: in other words, none of the subleading contributions blow up and spoil the estimates, so they still apply in this limit. To show that $f_n^{1/2} \in \mathcal{A}^{1/2}$, all that remains is then to show real-analyticity.

To do this, let us take $\alpha, \beta \notin \frac{1}{2} + \mathbb{Z}$ with $\alpha > \beta$. Since $\mathcal{A}^\alpha \subset \mathcal{A}^\beta$, and $\{f_n^\beta\}$ is a complete

basis for $\mathcal{A}^\beta$, then $f_n^\alpha \in \mathcal{A}^\alpha$ may be decomposed with respect to this basis:

$$f_n^\alpha = \sum_m (f_n^\alpha, g_{\Delta_m^\beta}^{(\beta)}) \, f_m^\beta \; . \tag{D.17}$$

The coefficients $(f_n^\alpha, g_{\Delta_m^\beta}^{(\beta)})$ appearing here are not simple in terms of our functional actions $\theta_n^{\tilde{\alpha}}(\Delta) \equiv (f_n^\alpha, g_\Delta^{(\alpha)})$. Instead, let us consider decomposing the elements $z^{\beta-\alpha} f_n^\alpha$:

$$z^{\beta-\alpha} f_n^\alpha = \sum_m (z^{\beta-\alpha} f_n^\alpha, g_{\Delta_m^\beta}^{(\beta)}) \, f_m^\beta = \sum_m \theta_n^\alpha(\Delta_m^\beta) \, f_m^\beta \; . \tag{D.18}$$

We claim that this sum converges in $\mathcal{A}^\beta$ (with the weak topology), uniformly over $\alpha$ in a neighbourhood of $\frac{1}{2}$. To show this, it is sufficient to note that the large-$m$ behaviour of the expansion coefficients $\theta_n^\alpha(\Delta_m^\beta)$ is uniform over $\tilde{\alpha}$.

Hence, we may take the limit $\alpha \to \frac{1}{2}^+$ in (D.18) with $\beta < \frac{1}{2}$ to obtain

$$z^{\beta-1/2} f_n^{1/2} = \sum_m \theta_n^{1/2}(\Delta_m^\beta) \, f_m^\beta \qquad \text{in } \mathcal{A}^\beta \; . \tag{D.19}$$

Since the space $\mathcal{A}^\beta$ is complete, then $z^{\beta-1/2} f_n^{1/2}$ is an element and is hence real-analytic. But this means $f_n^{1/2}$ is also real-analytic on $(1, \infty)$. We have therefore shown that the pointwise limit (D.16) converges and is an element of $\mathcal{A}^{1/2}$.

**Completeness.** On the other hand, to show that $\{f_m^{1/2}\}$ are complete in $\mathcal{A}^{1/2}$, we would instead like to take the limit $\beta \to \frac{1}{2}^+$ in (D.18). For this, we note that (D.18) can be derived from the Cauchy kernel decomposition (3.38), which we repeat here:

$$\frac{1}{z-w} = \sum_m g_{\Delta_m}^{(\beta)}(w) \, f_m^\beta(z) \; . \tag{D.20}$$

This converges as an analytic function of $z$ and a hyperfunction of $w$. Moreover, this convergence, controlled by the estimate (2.33), is completely uniform over $\beta$ near $\frac{1}{2}$. In particular, it converges in the weak topology for the space $\mathcal{A}^\beta$ with respect to the $z$ variable, uniformly over $\beta$. Acting on this hyperfunction of $w$ with the functional $z^{\beta-\alpha} f_n^\alpha$ gives the decomposition (D.18), and tells us that its weak-topology convergence is uniform over $\beta$.

We may therefore take the limit $\beta \to \frac{1}{2}^+$ to obtain

$$z^{1/2-\alpha} f_n^\alpha = \sum_m \theta_n^\alpha(\Delta_m^{1/2}) \, f_m^{1/2} \qquad \text{in } \mathcal{A}^{1/2} \; . \tag{D.21}$$

Since $\{f_n^\alpha\}$ are complete in $\mathcal{A}^\alpha$, the objects on the LHS are complete in a space $z^{1/2-\alpha} \mathcal{A}^\alpha$ with slightly softer behaviour at $z = \infty$. Thus we learn that the space generated by $\{f_m^{1/2}\}$

includes at least that one. Taking $\alpha$ arbitrarily close to $\frac{1}{2}$, that space becomes $\mathcal{A}^{1/2}$ and we learn that $\{f_m^{1/2}\}$ are complete.

**Duality.** Now that we have established that $\{f_n^{1/2}\}$ are a complete set in $\mathcal{A}$, their duality to the desired spectrum is immediate. For example, starting with $\alpha > \frac{1}{2}$, we have

$$\delta_{mn} = (f_m^{\tilde{\alpha}}, g_{\Delta_n^{\tilde{\alpha}}}^{(\tilde{\alpha})}) \equiv \oint_\Gamma \frac{\mathrm{d}z}{2\pi i}\, f_m^{\tilde{\alpha}}(z)\, g_{\Delta_n^{\tilde{\alpha}}}^{(\tilde{\alpha})}(z) \ . \tag{D.22}$$

This integral clearly converges uniformly over $\tilde{\alpha}$, so taking the limit $\tilde{\alpha} \to \frac{1}{2}^+$ through the integral gives

$$(f_m^{1/2}, g_{\Delta_n^{1/2}}^{(1/2)}) = \delta_{mn} \ . \tag{D.23}$$

**Continuity across thresholds.** Finally, we note that this whole arugment could have been made instead with the lower limit $\tilde{\alpha} \to (\frac{1}{2} + M)^-$ in (D.16). The resulting functionals would be identical, since they are dual to the same spectrum.[26] Therefore, while the relation to the $L^2$ space is subject to a subtraction procedure across the values $\tilde{\alpha} \in \frac{1}{2} + \mathbb{Z}$, everything physical (i.e. involving the physical spaces $\mathcal{A}$ and $\mathcal{HF}$) is perfectly continuous across these thresholds — with the upper and lower limits coinciding.

# E  Perturbative computation of interacting bases for special values of $\Delta_\phi$

This short appendix contains technical details for the analytic computation of interacting functionals in the particular example (5.25) with $\Delta_\phi = 1$. We begin by setting

$$\delta h_n(z) := h_n(z) - \hat{h}_n(z) = \frac{f_n(z) - \hat{f}_n(z)}{\pi \mu(z) z^{1+\Delta_\phi}} = -\sum_{m=1}^{\infty} \gamma_m\, \hat{\theta}_n'(\hat{\Delta}_m)\, \hat{h}_m(z)\,, \tag{E.1}$$

and recall that $\gamma_n = \frac{2\gamma_0}{\hat{\Delta}_n(\hat{\Delta}_n - 1)}$. Acting with the Casimir operator on $\delta h_n$ and using (A.8) the above relation leads to two new sums. One of them is the same as the original but with $\gamma_n$ set to a constant, and is thus proportional to $\partial_{\tilde{\alpha}} h_n|_{\tilde{\alpha} = \Delta_\phi}$. The second sum can be performed explicitly, as the $z$-dependence of the summands factors out. We thus get

$$C_z\, \delta h_n(z) = \frac{\gamma_0}{2}\, \partial_{\tilde{\alpha}} \hat{h}_n|_{\tilde{\alpha} = \Delta_\phi} - \frac{B_n}{\mu(z)\, z^{1+\Delta_\phi}} \ , \tag{E.2}$$

---

[26]Explicitly, the difference between the upper and lower limits would be a functional that vanishes on all blocks $g_{\Delta_n^{1/2+M}}^{(1/2+M)}$. These blocks will be dense in $\mathcal{HF}^{1/2+M}$ (by an argument similar to that for $\mathcal{A}$ above), forcing a functional vanishing at all of them to be the zero functional.

where

$$B_n = \sum_{m=0}^{\infty} \gamma_m \, A_m \, \theta'_n(\hat{\Delta}_m) \; . \tag{E.3}$$

We can now solve for $\delta h_n(z)$ by making an ansatz:

$$\delta h_n(z) = a_1 \, \partial_{\tilde{\alpha}} \hat{h}_n|_{\tilde{\alpha}=\Delta_\phi} + a_2 \, \hat{h}_n + a_3 \, g_{\Delta=0}(z) + a_4 \, g_{\Delta=1}(z) + a_5 \, m(z) + a_6 \, n(z) \; . \tag{E.4}$$

Here $m(z)$ and $n(z)$ satisfy

$$C_z \, m(z) = \frac{1}{\mu(z) z^{1+\Delta_\phi}}, \qquad C_z \, n(z) = \frac{1}{\mu(z) z^{1+\Delta_\phi}} \log(z) \; . \tag{E.5}$$

and can be solved for explicitly for $\Delta_\phi = 1$. Imposing (E.2), as well as $\delta h_n$ being sufficiently well behaved for large $z$, uniquely determines all $a_i$.

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
