# Peer review of "Bootstrapping bulk locality. Part II: Interacting functionals"

_SciPost Physics_

## Round 2 · Referee Report · Anonymous (Referee 1) · 2025-3-19

Report

The paper contains a discussion of analytic functionals for the locality of the bulk form-factors. The main result is the construction of families of analytic functionals dual to certain "interacting" types of boundary spectra.

This result is certainly very interesting, and without doubt easily meets the criteria for publication.

However, I found that the paper itself is suprisingly challenging to read due to its style. Specifically, while the paper is quite mathematical in nature, relying on some non-trivial functional analysis and other techniques, the reader is rarely given a chance to follow the details and precise statements are often substituted with informal ones. This leaves it up to the reader to distill the clean mathematical results from the text, which is quite long at 84 pages.

I think the readability and usefulness of the paper to the readers would improve significantly if some precise theorems, propositions and definitions were formulated. While not standard in physics literature, I do believe they are called for, given the amount of functional analysis in the paper.

For instance, the space $\mathcal{A}$ is introduced in (2.17) but the first mention of any topology on $\mathcal{A}$ is in footnote 13. This topology is declared to be the weak topology induced by $\mathcal{HF}$. If we take this as the topology in (2.16), then (2.16) is true by a standard argument. However, choosing the weak topology on $\mathcal{A}$ only guarantees that $(\cdot,\cdot)$ is continuous on $\mathcal{A}$. Therefore, and contrary to what is claimed, the swapping property (2.20) is not automatic, as it requires continuity in a particular topology on $\mathcal{HF}$. Instead, one needs to prove a version of the theorem in footnote 10 for the actual defintions of $\mathcal{HF}$ and $\mathcal{A}$ (the quoted but not referenced in footnote 10 duality is only stated for a compact real domain).

This is not done, and yet the informal nature of discussion makes it appear as if (2.20) follows. Related details are hidden in footnote 8.

Based on the technical side of the above objection alone, I would recommend a revision that would add an argument or a reference for the claimed continuity of the pairing. It is possible that I have misunderstood the logic and that there is no technical problem. However, this possibility only highlights the need to revise the exposition of at least the key points, stating clear definitions and theorems.

For this reason, my recommendation is to publish the paper after a major revision to address the above concerns. I want to stress that they are not confined to the vicinity of equation (2.20), and the above is only meant as an illustration.

Recommendation

Ask for major revision

---

## Round 2 · Referee Report · Anonymous (Referee 2) · 2025-3-24

Report

The paper considers the problem of the existence of local operators in AdS compatible with given boundary CFT data. Its focus is the analysis of completeness of sets of sum rules on the bulk-to-boundary OPE coefficients.

From the abstract, a reader learns: "The sum rules trivialise the reconstruction of bulk operators in strongly interacting QFTs in AdS space and allow us to write down explicit, exact, interacting solutions to the locality problem."

In my opinion, this sentence vastly oversells the achievements of the article. In particular, the article only deals with the bulk locality constraint arising from a single three-point function of the bulk operator with two fixed boundary operators. In reality, the bulk locality problem necessarily involves imposing this constraint for all pairs of boundary operators. In particular, while the constraints studied in the present article are necessary, it is not clear if their solutions are relevant for the full locality problem.

Relatedly, the paper also stresses the role of so-called extremal solutions to four-point crossing, whose spectra are dual to the sets of zeros of the basis functionals. A feature of the extremal spectra is their relative sparseness at high energy. On the other hand, it is expected that the spectra exchanged in OPEs in physical CFTs are much more dense at high energy. With the exception of generalized free fields, the extremal solutions are thus likely an unphysical artifact of focusing on a single crossing equation. In that light, it is not clear what is the physical relevance of the interacting bases considered in this paper.

The actual question on which the article focuses is the following. The implication of bulk locality for a single three-point function is that a certain discontinuity vanishes. This in turn implies sum rules on the bulk-to-boundary coefficients of the AdS local operator. The paper discusses completeness of various sets of sum rules. In other words, it aims to identify sets of sum rules whose validity implies the vanishing of the discontinuity for a fixed pair of boundary operators.

Although this question is only a small component of the full bulk locality problem, it is nontrivial and interesting. The paper attacks it using a variety of tools from functional analysis. However, the arguments are not particularly clear or easy to follow. In particular, it is not clear where exactly one can find the proof of the main claimed result of the paper (that certain sets of sum rules imply vanishing discontinuity).

Given these comments, I recommend that the authors rewrite the main results using a clear mathematical language, and that their temper the rather overly grand statements in the abstract and introduction.

Recommendation

Ask for major revision

---

## Editorial Decision

awaiting_resubmission